# Aging-associated decline of phosphatidylcholine synthesis is a malleable trigger of natural mitochondrial aging

Tetiana Poliezhaieva[1,5], Yuting Li [1,2], Prerana Shrikant Chaudhari[1], Ulas Isildak[1], Pol Alonso-Pernas[1], Isabela Santos Valentim [1], Fengting Su [3], Lilia Espada [1,5], Melike Bayar[1], Li Fu [2], Andreas Koeberle [3], Handan Melike Dönertaş [1,4] & Maria A. Ermolaeva [1,4] ✉

Mitochondrial dysfunction is a prominent hallmark of aging contributing to the decline of metabolic plasticity in late life. While genetic distortions of mitochondrial integrity elicit premature aging, the mechanisms leading to "natural" aging of mitochondria are less clear. Here we use proteomics, lipidomics, genetics and functional tests in wild type *Caenorhabditis elegans* and long-lived *clk-1(qm30)* and *isp-1(qm150)* mitochondrial mutants to identify molecular pathways that support longevity amid persistent mitochondrial inefficiency. These tests and subsequent transcriptomics and metabolomics analyses in humans reveal aging-associated decline of phosphatidylcholine synthesis as a trigger of mitochondrial network disruption, which contributes to mitochondrial dysfunction during normal aging. Moreover, ectopic boosting of phosphatidylcholine levels via diet restores late life mitochondrial integrity in vivo in nematodes and reinstates metabolic resilience in human cell culture tests. We thus describe a previously unrecognized natural driver of mitochondrial decline in aging that is malleable by dietary interventions.

Aging is characterized by the progressive deterioration of multiple cellular and organismal functions[1], and one of the key challenges in countering aging-associated functional decline is to identify and mitigate core mechanistic drivers of the aging process. Mitochondrial dysfunction is clearly one of the best recognized hallmarks of aging[1,2], and in addition to causing cellular deterioration in late life, it blunts the efficacy of anti-aging interventions that rely on metabolic plasticity such as dietary restriction (DR) and DR mimetics[3,4]. Despite extensive studies linking genetic impairments of mitochondrial homeostasis (e.g., altered fidelity of mtDNA synthesis, mitochondrial unfolded protein response (UPR), oxidative phosphorylation (OXPHOS) and others) to diseases and accelerated aging[2,5,6], it is less clear what

endogenous processes instigate mitochondrial decline during normal aging. The identification of such "natural" drivers of mitochondrial aging is however crucial because they likely comprise suitable intervention targets towards restoration of mitochondrial integrity and organismal health in late life. Focusing on late-life events is especially important to identify interventions that are truly appropriate for older individuals, rather than extrapolating early-life changes to therapies applied in old age, when these effects may no longer be relevant or reversible. In this work, we combined omics, genetics and functional analyses in *C. elegans*, with transcriptomics and metabolomics analysis in humans, and metabolic resilience tests in cell culture models to discover previously unrecognized interventions that improve

[1]Leibniz Institute on Aging – Fritz Lipmann Institute (FLI), Beutenbergstrasse 11, Jena, Germany. [2]Department of Pharmacology and International Cancer Center, Shenzhen University Medical School, Shenzhen, Guangdong, China. [3]Institute of Pharmaceutical Sciences and Field of Excellence BioHealth, NAWI Graz, University of Graz, Graz, Austria. [4]Cluster of Excellence Balance of the Microverse, Friedrich Schiller University Jena, Jena, Germany. [5]Present address: TP - Philipps-Universität Marburg, Institute for Anatomy and Cell Biology, Robert-Koch-Straße 8, 35037 Marburg; LE - ISAR Bioscience, Semmelweisstraße 5, Planegg, Germany. ✉e-mail: maria.ermolaeva@leibniz-fli.de

mitochondrial health and metabolic plasticity during advanced aging. Our studies revealed a decline in phosphatidylcholine (PC) synthesis as a previously unappreciated, conserved driver of natural mitochondrial aging, which can be overcome by dietary supplements.

## Results

### Proteomics analysis of wild type *C. elegans* and long-lived mitochondrial mutants reveals metabolic decline as a late event during aging

In our search for malleable processes that can alleviate mitochondrial deterioration with age, we decided to analyze genetic backgrounds that maintain normal or even extended longevity amid persistent mitochondrial dysfunction. While in most cases congenital mitochondrial impairments lead to cell dysfunction or accelerated aging[2], there are two extensively characterized examples in *C. elegans*–the hypomorphic mutants *clk-1(qm30)* and *isp-1(qm150)*, which exhibit extended longevity despite carrying impaired mitochondria for their entire lifetime[7,8]. It is thus feasible that these mutants possess intrinsic molecular adaptations, which support functionality of mildly impaired mitochondria. The previous omics studies comparing the mutants to WT counterparts were performed exclusively at young age[9–12], identifying early life molecular differences without offering an aging perspective. To uncover potential adaptations with a life-long resolution, we performed label-free unbiased proteomics analysis in young (adulthood day 1, AD1), middle aged (adulthood day 5, AD5) and old (adulthood day 10, AD10) WT and mutant nematodes (Fig. 1a) searching for protein expression changes, which occur in the aging WT but are alleviated in both mito-mutants. In this experiment, we measured a total number of 5339 proteins across all conditions (Supplementary Data 1), and our proof of concept data analysis successfully detected previously described molecular features of the mito-mutants such as reduced ribosomal content in young *clk-1(qm30)* animals[13] (Supplementary Fig. 1a and Supplementary Data 2–3) and increased expression of the mitochondrial UPR components HSP-6 and HSP-60 in young *isp-1(qm150)* mutants[14] (Supplementary Fig. 1b and Supplementary Data 4). We next assessed protein changes occurring during early and advanced WT aging by analyzing proteins differentially regulated (Log2FC ≥ 0.58 and Q ≤ 0.05) between young (AD1) and post-reproductive (AD5), and young (AD1) and old (AD10) WT animals. The purpose of this experiment was to discover at what point mitochondria begin to fail during normal aging. We found a higher number of proteins (1835) to be differentially regulated in the AD10/AD1 comparison compared to the AD5/AD1 condition (1187) (Fig. 1b and Supplementary Data 5–7) in line with the expected stronger effect of aging in AD10 nematodes. We next extracted regulated proteins, which were either specific to early (AD5/AD1) or advanced (AD10/AD1) aging or overlapped between the two conditions (Supplementary Data 7, 8), and analyzed these by using the WormCat pathway analysis tool[15]. The analysis identified alterations in mRNA splicing as the strongest enriched specific feature of early aging (Fig. 1b, Supplementary Data 9 and 46). Middle age was characterized by changes in stress responses, and old age was marked by chromatin-related shifts, while ribosomal changes were common to both middle and advanced age (Fig. 1b, Supplementary Data 10, 11). Finally, advanced aging showed outstanding enrichment of metabolic pathways, including those related to lipid and mitochondrial metabolism (Fig. 1b, Supplementary Data 11). This data suggested that alterations of mRNA processing, proteostasis and stress responses occur relatively early during aging in line with previous reports[16,17], while metabolic and mitochondrial aging occur at a later point, consistent with our previous findings demonstrating functional loss of metabolic plasticity in nematodes on adulthood day 10[3]. Because our interest in studying mitochondrial mutants was mainly directed at the reversal of metabolic aging, we next compared protein sets differentially regulated between AD10 and AD1 in WT animals and the two mito-mutants. Initially, we observed an inverse correlation between the number of age-affected proteins and lifespan across the three strains compared: the WT strain, which had the shortest lifespan, exhibited the highest number of affected proteins (1835) (Fig. 1c, d and Supplementary Data 6) while the lowest number (1323) was found in the longest lived *isp-1(qm150)* mutant, with *clk-1(qm30)* nematodes residing in the middle both in terms of the number of affected proteins (1625) and the reported longevity[11] (Fig. 1c–e; Supplementary Data 12, 13, 17 and 18). This observation gave us additional reassurance of our data being representative of the aging processes in the strains. By combining WormCat analysis with the same data extraction strategy used in Fig. 1b, we found that several metabolic terms exhibited overlapping age-associated patterns in WT and both mitochondrial mutants (Fig. 1f, g and Supplementary Data 13–21) despite the perceived metabolic resilience of the mutants. At the same time, we found distinct mitochondrial processes to be specifically enriched in the aged mito-mutants in line with (i) the mitochondrial origin of their differential longevity and (ii) their functionally distinct mitochondrial impairments (Fig. 1c, d; and Supplementary Data 16 and 21). Finally, when the two mitochondrial mutants were compared, only a limited portion of the overlapping age-related proteomic changes was associated with mitochondrial proteins (Fig. 1e, h; and Supplementary Data 22–26). This suggests that the shared adaptation to lifelong mitochondrial dysfunction may largely involve non-mitochondrial processes. Of note, the terms highlighted by individual enrichment in the mutants such as chromatin and histone modifications and mitochondrial ribosome (Fig. 1e, and Supplementary Data 25 and 26), were previously linked to both longevity extension and moderate mitochondrial dysfunction[18,19], confirming that the chosen data analysis approach delivers physiologically relevant outputs.

### The S-adenosylmethionine synthetase SAMS-1 facilitates longevity of mitochondrial mutants and is downregulated during normal aging

To further explore pathways that mediate adaptive responses to mitochondrial dysfunction, we next searched for individual proteins whose expression is strongly and progressively altered during advanced aging in WT animals, but whose changes are attenuated in the aging mito-mutants. We started by the unbiased filtering of all regulated proteins in the WT AD10/AD1 data set to select the strongest changes with AVG Log2FC ≥ 2,5 followed by their Q value ranking in the ascending order, placing most significant changes on top of the list (Supplementary Data 27). We next prioritized for downregulated proteins, which might indicate loss of function with aging, and this analysis returned S-adenosylmethionine synthetase SAMS-1 as a clear top candidate (Supplementary Data 27). Notably, the downregulation of SAMS-1 protein in aged nematodes has also been reported in previous studies[20–22], thereby validating our findings. Further analysis showed that the reduction of SAMS-1 protein expression in WT animals was indeed progressive between AD5 and AD10 age points (Fig. 2a, b, and Supplementary Data 4), and this change was alleviated in both mito-mutant backgrounds and at both tested aging stages (Fig. 2a, b, and Supplementary Data 4). Moreover, one carbon metabolism (1CC)–the pathway, which involves *sams-1*[23], was indeed among the top pathway enrichment outputs, which overlapped between the two aging mito-mutants in the proteomics analysis (Supplementary Data 24). Collectively, these data indicated that sustained SAMS-1 expression may support organismal function in the presence of diverse mitochondrial impairments. To experimentally test this hypothesis, we next performed the RNAi-mediated knock down (KD) of the *sams-1* gene in WT and mito-mutant strains (Supplementary Fig. 1c). We found that *sams-1* KD elicits lifespan extension in WT animals (Fig. 2c) in line with previous reports[24,25], whereas the lifespan of both long-lived mito-mutants was significantly shortened upon *sams-1* KD (Fig. 2d, e). These original findings indicate that the impact of

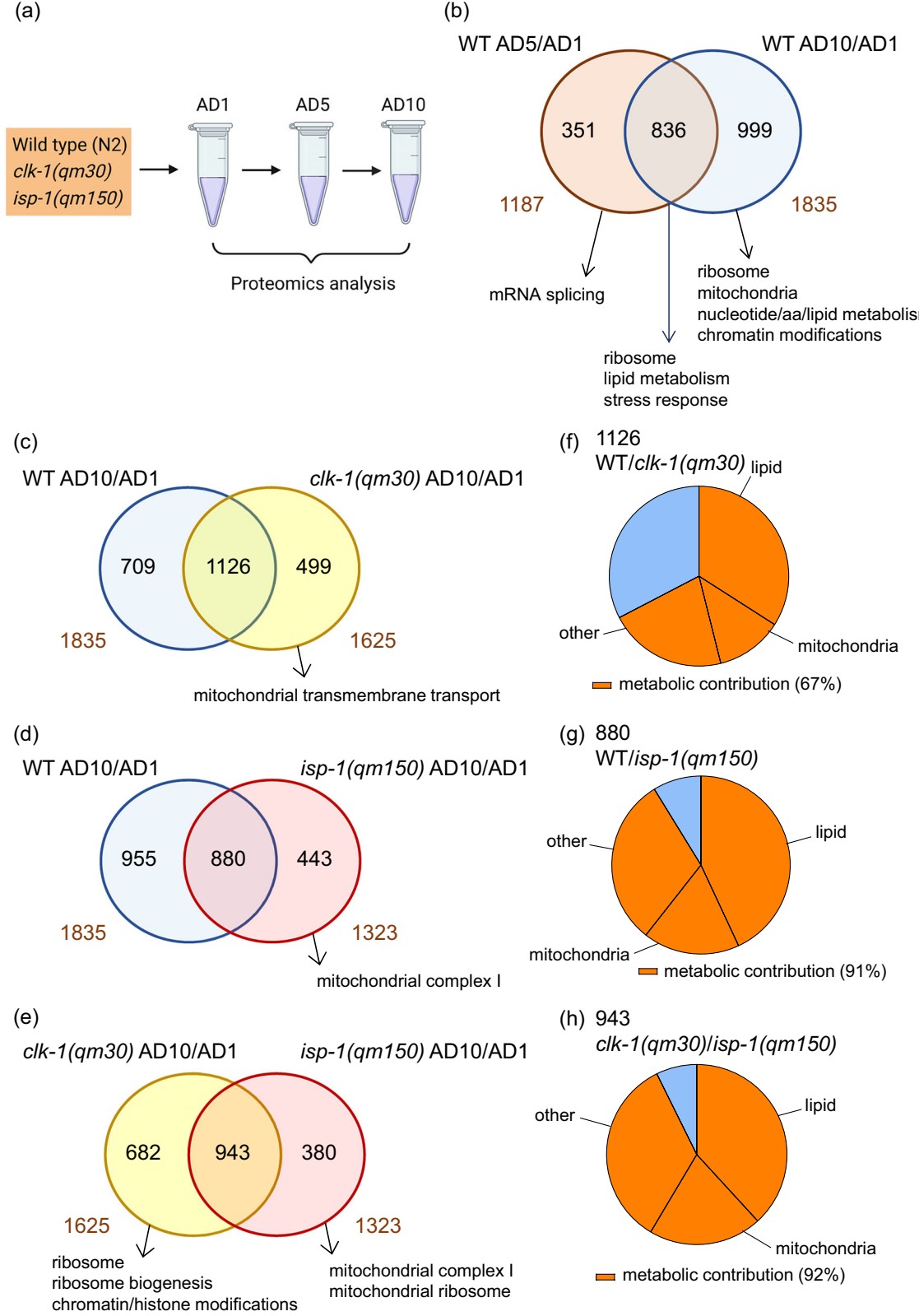

sams-1 on longevity depends on mitochondrial health. Specifically, in the presence of young, functional mitochondria, such as in young wild-type (WT) animals, *sams-1* deficiency exerts a positive longevity effect. In contrast, the presence of mitochondrial impairments, as seen in mito-mutants or aged WT animals[3], offsets the longevity benefits of *sams-1* deficiency. This suggests that *sams-1* may have a role in mitochondrial homeostasis.

## *sams-1* gene inactivation disrupts mitochondrial network integrity and triggers mitochondrial unfolded protein response

We next tested if *sams-1* gene inactivation has a direct impact on mitochondrial integrity and dynamics by subjecting the reporter strain harboring GFP targeted to mitochondria in the body wall muscle[26,27] to *sams-1* RNAi. Interestingly, *sams-1* KD resulted in the severe disruption of the mitochondrial network as seen by loss of tubular mitochondria

**Fig. 1 | Metabolic alterations represent a late event during aging. a** Wild type (WT, N2 strain), *isp-1(qm150)* and *clk-1(qm30)* nematodes were age-synchronized, and proteomics samples (*n* ≥ 500 animals per sample) were collected at young (adulthood day 1, AD1), post-reproductive (AD5) and old (AD10) age. The image is created with BioRender (Ermolaeva, M. (2026) https://BioRender.com/8k33rq6). **b** Differentially expressed proteins (DEPs, absolute Log2FC ≥ 0.58 and Q ≤ 0.05) in post-reproductive (AD5 vs AD1, WT AD5/AD1) and old (AD10 vs AD1, WT AD10/AD1) wild type animals were compared, revealing overlapping and non-overlapping subsets. Each of these subsets was analyzed using WormCat pathway analysis tool. Venn diagram and respective WormCat analysis highlights are shown. Numbers outside the diagram depict total DEPs in each condition, numbers inside the diagram represent respective overlapping and non-overlapping DEPs. Complete lists of DEPs used as an input, Venn diagram sub-setting and WormCat outputs for (**b**) can be found in Supplementary Data 5–11. **c** Proteins differentially regulated in aged wild-type animals (AD10 vs. AD1, WT AD10/AD1) and aged *clk-1(qm30)* mutants

(AD10 vs. AD1, *clk-1(qm30)* AD10/AD1) were compared, analyzed, and presented as in panel (**b**) with the relevant WormCat output shown in the caption. Respective protein lists and WormCat highlights, assembled as for panel (**b**) are shown in Supplementary Data 12–16. **d** Proteins differentially regulated in old wild-type animals (WT AD10/AD1) and old *isp-1(qm150)* mutants (*isp-1(qm150)* AD10/AD1) were compared, analyzed and presented as in (**b**). Accessory lists and analyses are shown in Supplementary Data 17–21. **e** Proteins differentially expressed in old *clk-1(qm30)* (*clk-1(qm30)* AD10/AD1) and *isp-1(qm150)* (*isp-1(qm150)* AD10/AD1) mutants were compared, analyzed and presented as in (**b**). Accessory lists and analyses are shown in Supplementary Data 22–26. **f**–**h** the overlapping DEPs from (**c**–**e**) respectively were analyzed using WormCat as shown in Supplementary Data 15, 20 and 24, and partitioning of respective Category 3 data into metabolic and non-metabolic pathways is presented, indicating the proportion (%) of Category 3 RGS (Regulated Gene Score) belonging to metabolic pathways. All native WormCat outputs including graphics are provided in Supplementary Data 46.

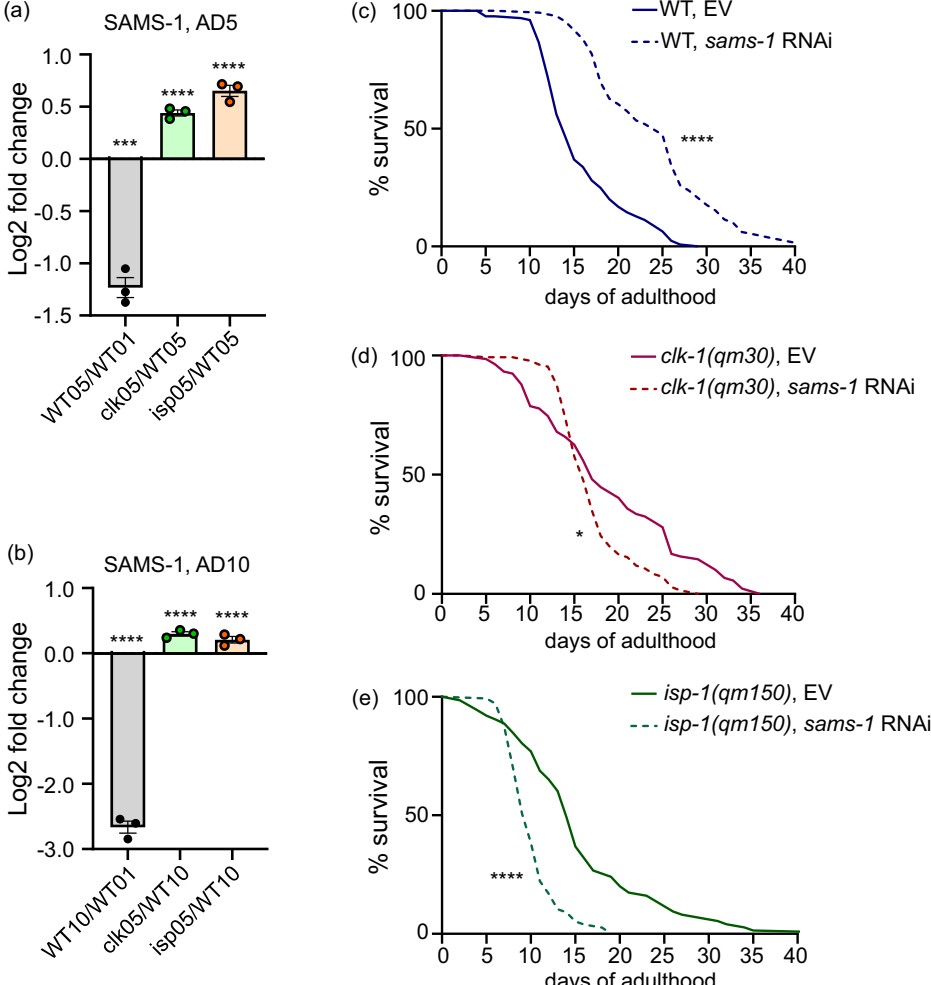

**Fig. 2 | The longevity of mitochondrial mutants is compromised in the absence of SAMS-1.** Proteomics samples were collected as described in Fig. 1a, and DEPs with strongest downregulation during wild type aging were identified as shown in Supplementary Data 27. **a** Relative SAMS-1 protein levels (log₂ fold change) in post-reproductive vs. young WT *C. elegans* (AD5 vs AD1, labeled as WT05/WT01), and in post-reproductive *clk-1(qm30)* and *isp-1(qm150)* mutants compared to age-matched WT controls (labeled as clk05/WT05 and isp05/WT05 respectively); *n* = 3, each dot represents an independent replicate of ≥500 animals. (**b**) The same analysis as in (**a**) was performed in old (AD10) animals, and relative SAMS-1 expression values are presented; *n* = 3, each dot represents an independent replicate of ≥500 animals. In **a**–**b** mean and SEM values are presented. **c**–**e** WT and mito-mutant animals were

age synchronized and exposed to *sams-1* RNA or empty vector (EV) control RNAi from the L1 larval stage, survival was scored daily, *n* = 140 in each experimental condition. The data shown in **c**–**e** belongs to the same experiment with all strains analyzed in parallel, and WT serving as control for *clk-1(qm30)* and *isp-1(qm150)* mutants. This experiment was repeated 3 times with one representative result shown in **c**–**e**. The accessory proteomics data for **a**–**b** can be found in Supplementary Data 4. In (**a**–**b**) statistics was calculated using unpaired t-test and Tukey's multiple comparison test, in (**c**–**e**) Mantel-Cox test was used; two-tailed *p* values were calculated in all cases. *-*p* < 0.05; ***-*p* < 0.001 and ****-*p* < 0.0001. Exact *p* values and *n* numbers can be found in the Source Data file.

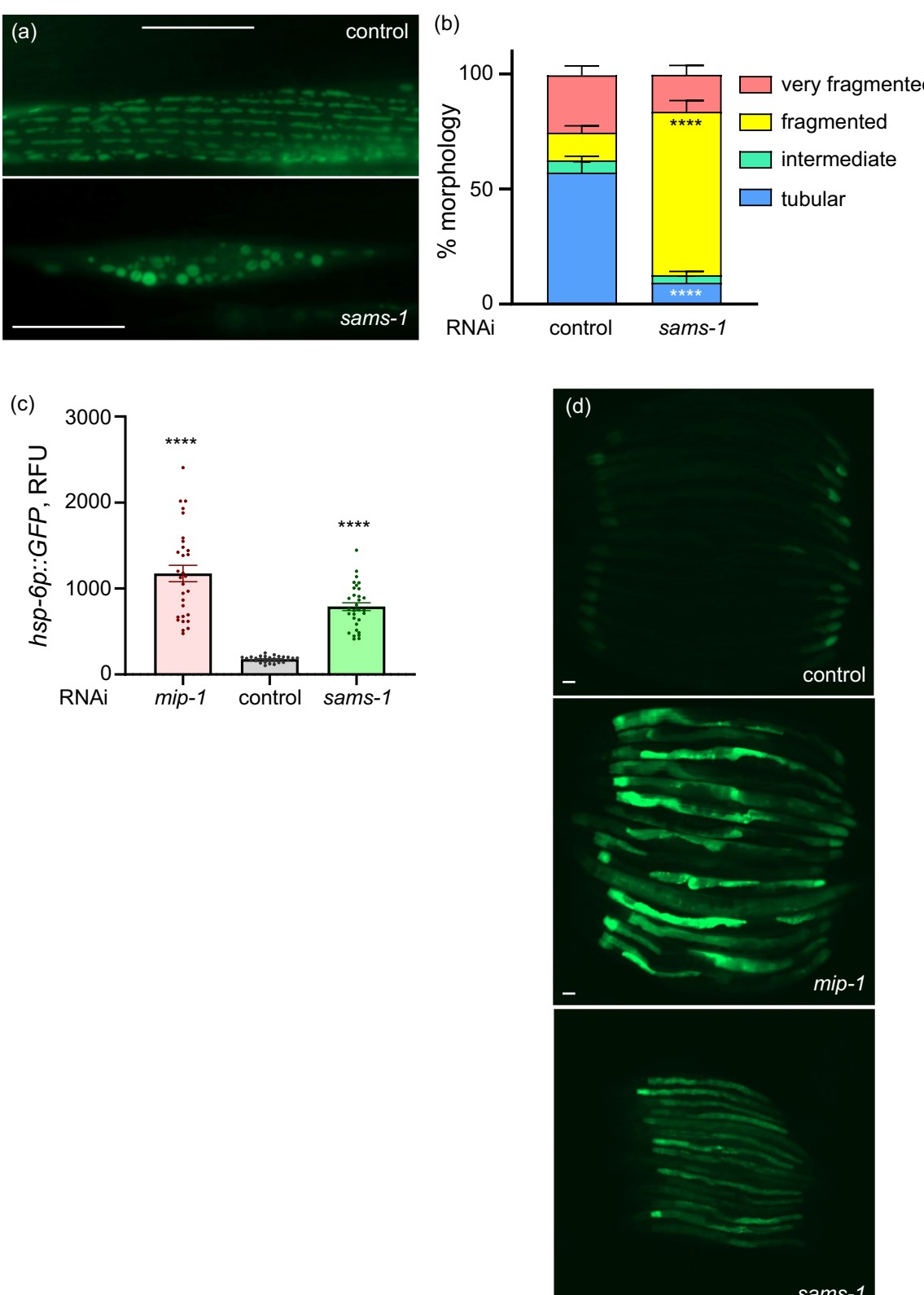

paralleled by an increase of fragmented and very fragmented organelles in the *sams-1* inactivated animals (Fig. 3a, b). Because mitochondrial fusion is implicated in supporting mitochondrial efficacy and homeostatic integrity[28,29], we next tested whether *sams-1* knockdown affects other aspects of mitochondrial health, such as proteostasis by using the *C. elegans* strain expressing the mitochondrial unfolded protein response (UPR^mt) reporter *hsp-6p::GFP*[14] and using RNAi against the

mitochondrial intermediate peptidase *mip-1(Y67H2A.7)* as a positive control for the disruption of mitochondrial proteostasis[30]. We found elevated levels of GFP expression in the reporter animals exposed to *sams-1* RNAi (Fig. 3c, d) consistent with previous reports[31] and the requirement of *sams-1* for the stress-free mitochondrial homeostasis. Collectively, our data demonstrated a prominent role of *sams-1* in safeguarding mitochondrial network integrity and homeostasis.

**Fig. 3 | sams-1 inactivation disrupts mitochondrial network integrity.**
**a, b** Transgenic animals expressing GFP-tagged mitochondria in the body wall muscle (*myo-3p::gfpmit*) were age-synchronized and exposed to *sams-1* RNAi or empty vector control from the L1 stage. Representative images of control and *sams-1* knock down (KD) mitochondria are shown in (**a**), scale bar 10 μm. (**b**) The % of tubular, intermediate, fragmented, and very fragmented mitochondrial morphologies in *sams-1* KD and control nematodes were scored in young, actively reproducing adults (day 2 of adulthood, AD2), *n* = 60 in each condition. **c, d** Transgenic nematodes expressing GFP under control of the *hsp-6* mitochondrial chaperone gene promoter (*hsp-6p::GFP*) were age synchronized and exposed to control and

*sams-1* RNAi from the L1 stage with *mip-1(Y67H2A.7)* RNAi serving as a positive control for the UPR MT induction. **c** GFP expression was assessed microscopically and quantified on AD2, *n* ≥ 30 with exact n number for each condition provided in the Source Data file; each dot corresponds to one animal, and representative images are shown in (**d**), scale bar 100 μm, RFU−relative fluorescence units. Statistics in (**b**, **c**) were assessed using unpaired t-test with Welch's correction, mean and SEM values are presented; two-tailed *p* values were calculated. ****-*p < 0.0001*. Each experiment was repeated at least 3 times, and one representative result is shown in each case. Exact *p* values can be found in the Source Data file.

### sams-1 and S-adenosylmethionine influence mitochondria through their role in the synthesis of phosphatidylcholine

The *sams-1* gene encodes the *C. elegans* orthologue of the S-adenosylmethionine synthetase, which is responsible for the production of the key cellular methyl donor S-adenosylmethionine (SAM)[23] (Fig. 4a). We next asked if SAM depletion was causal of mitochondrial distortions induced by the *sams-1* KD (Fig. 3). In line with this hypothesis, the dietary SAM supplementation alleviated mitochondrial fragmentation and reduced mitochondrial UPR activity in *sams-1* RNAi exposed nematodes (Fig. 4b, c and Supplementary Fig. 2a, b), confirming the causal role of SAM synthesis in mitochondrial effects of *sams-1* deficiency. As indicated previously, SAM is the key methyl donor for the variety of cellular processes including DNA and histone methylation as well as synthesis of the membrane phospholipid PC[32] (Fig. 4d). Because the lipid composition of mitochondrial membranes influences mitochondrial fission and fusion[33], and PC is the most abundant lipid in both outer and inner membranes of mitochondria in the majority of eukaryotic species[34–36] contributing to the membrane fluidity required for fusion[37], we next asked if mitochondrial phenotypes seen in young *sams-1* KD animals were driven by the disruption of SAM-dependent PC synthesis. For that we focused on the best-characterized methylation-dependent pathway for phosphatidylcholine synthesis in *C. elegans*: the plant-like Phosphobase Methylation Pathway[38]. In this route, SAM is consumed as a co-substrate by the phosphoethanolamine N-methyltransferases PMT-1 and PMT-2 in a three-step methylation process - one SAM molecular per step - to generate phosphocholine, a precursor for PC synthesis via the Kennedy pathway[38,39] (Fig. 4d and Supplementary Fig. 3a). We next examined mitochondrial morphology and UPR[mt] induction in nematodes exposed to *pmt-1* RNAi (Supplementary Fig. 3b) and found that *pmt-1* inactivation at young age phenocopied the effect of *sams-1* KD on mitochondrial integrity (Fig. 4e and Supplementary Fig. 3c), while only a small effect of *pmt-1* RNAi on the *hsp-6p::GFP* expression was detected (Fig. 4f and Supplementary Fig. 3d). This suggests that (i) mitochondrial morphology changes do not consistently correlate with the activation of the mitochondrial unfolded protein response (UPR MT), and (ii) the influence of *sams-1* inactivation on UPR MT is possibly independent of its effects on mitochondrial morphology. To this end, previous studies demonstrated the dependency of mitochondrial UPR activity on the extent of histone methylation[18], which uses the product of *sams-1* activity SAM as a key donor[40], providing a possible alternative link. We next demonstrated that *pmt-1* deficiency extended lifespan of WT animals but reduced the lifespan of both analyzed mitochondrial mutants similar to the KD of *sams-1* (Fig. 4g) and the same effect was seen with KD of the second phosphoethanolamine N-methyltransferase *pmt-2* (Fig. 4d, g). These findings conclusively demonstrated a role for SAM-dependent PC synthesis in the regulation of longevity through the preservation of mitochondrial function.

We next tested the effect of the dietary PC supplementation on the mitochondrial and organismal phenotypes elicited by *sams-1* and *pmt-1* KDs. Interestingly, PC supplementation failed to dampen the *hsp-6p::GFP* expression in *sams-1* KD animals (Supplementary Fig. 4a and b), confirming that UPR MT induction in *sams-1* KD worms is independent of reduced PC synthesis. Concurrently, PC

supplementation mitigated the excessive mitochondrial fragmentation (Fig. 5a, b; and Supplementary Fig. 5a and b) and the reduced body size (Fig. 5c) in both *pmt-1* and *sams-1* KD worms. This suggests that disrupted PC synthesis underpins mitochondrial network disruption and contributes to at least some of the organismal defects observed in these KDs. The body size alteration (tested in *pmt-1* and *sams-1* KD worms) and mitochondrial morphology defects (tested in *sams-1* deficient animals) could also be rescued by choline (Cho) supplementation (Supplementary Fig. 6a and 6b). Choline is converted to phosphatidylcholine via the CDP-choline pathway, which involves three enzymatic reactions, with first step catalyzed by choline kinase. This pathway is active in most tissues and converges with the phosphobase methylation pathway at the level of the common intermediate phosphocholine (Supplementary Fig. 3a)[41,42],. Notably, the ability of choline to functionally restore PC levels in the nematode has been demonstrated in previous studies[43]. Because PC is chemically unstable and must be dissolved in mildly toxic organic solvents to be delivered to worms, we chose choline−which is water-soluble and more stable− as a safer and more practical way to increase PC levels in some tests. Notably the effects of choline and SAM on the mitochondrial morphology were independent of their metabolism by the RNAi producing bacteria as seen by tests with heat-killed bacteria (Supplementary Fig. 6b). Moreover, choline supplementation alleviated the strong mitochondrial network and body size defects of the animals lacking *pmt-2* (Supplementary Fig. 7a−c), aligning with the previous report in the case of the body size data[39]. The ability of choline supplementation to rescue the mitochondrial morphology and body size alterations of *sams-1*, *pmt-1* and *pmt-2* KD worms is thus supportive of the link between PC synthesis and organismal and mitochondrial fitness. Of note, reduction of body size was previously linked to lowered PC synthesis in nematodes[44] but no connection to mitochondrial integrity was previously made. To investigate how the alterations in mitochondrial morphology observed in *sams-1*, *pmt-1* and *pmt-2* RNAi-treated animals affect mitochondrial function, we performed whole-animal respirometry assays using the Seahorse platform. Our analysis revealed a significant decline in mitochondrial oxygen consumption rate (OCR) across the three knockdowns (Fig. 5d and Supplementary Fig. 7d). Moreover, mitochondrial OCR of *pmt-1* KD worms could be rescued by the supplementation of PC (Fig. 5e), whereas vehicle control animals (EV + EtOH condition) consistently showed impaired mitochondrial function due to ethanol exposure across experiments. Altogether, our findings demonstrate that PC-dependent mitochondrial fusion is crucial for maintaining both mitochondrial integrity and function.

Finally, we employed ultra-performance liquid chromatography coupled with mass spectrometry (UPLC-MS/MS) to validate the distinct lipid alterations in animals deficient in *sams-1* and *pmt-1*. Specifically, we measured levels of phosphatidylcholine (PC) and phosphatidylethanolamine (PE)−a related phospholipid synthesized independently of the enzymes involved in the methylation-dependent PC synthesis pathway (Supplementary Fig. 3a). We also quantified lysophosphatidylcholine (LPC) and lysophosphatidylethanolamine (LPE), two biologically relevant derivatives of PC and PE, respectively. All measurements were performed in control, *sams-1* and *pmt-1*

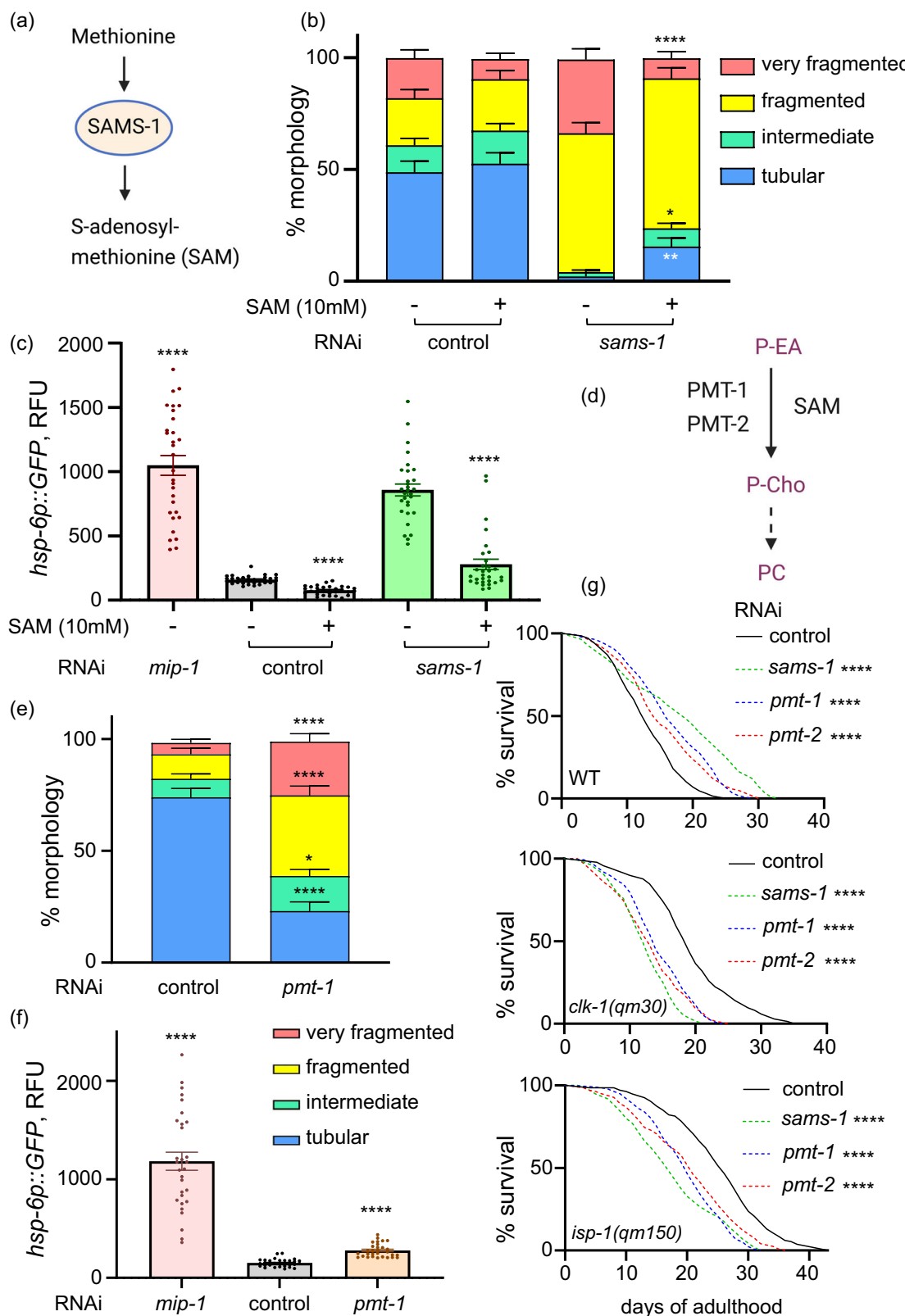

deficient animals with and without choline supplementation. LPC was selected for the measurement because: (i) it can originate from PC[45] and, therefore, contributes to the outcomes of PC abundance; (ii) in previous work, we found its levels to decrease with age, similar to those of PC[3]; and (iii) LPC is known to mediate membrane curvature and fusion[46]. In contrast, LPE was measured as a control lysophospholipid, as its levels are not directly influenced by the extent of

methylation-dependent PC synthesis. We observed a significant reduction in the PC/PE and LPC/LPE ratios in *sams-1* and *pmt-1* knockdowns (Fig. 6a, b, and Supplementary Data 28), consistent with a substantial decline in PC synthesis. Notably, choline − whose conversion to PC bypasses the methylation-dependent pathway (Supplementary Fig. 3a)−prevented the reduction of both ratios in these knockdowns. Overall, our findings demonstrate that disruption of the

**Fig. 4 | SAM-dependent phosphatidylcholine synthesis links *sams-1* to modulation of mitochondrial morphology. a** Schematic depicts the role of SAMS-1 in converting methionine to S-adenosylmethionine (SAM). Created in BioRender (Ermolaeva, M. https://BioRender.com/8k33rq6). **b** Transgenic nematodes expressing GFP-tagged mitochondria in the body wall muscle (*myo-3p::gfpmit*) were age-synchronized and exposed to *sams-1* RNAi or empty vector control from the L1 stage, 10 mM SAM was supplemented from the L4 stage. Mitochondrial morphologies were scored on AD2 as in Fig. 3b, *n* = 60 in each experimental group, and representative images can be found in Supplementary Fig. 2a. **c** Transgenic nematodes expressing GFP under control of the *hsp-6* promoter (*hsp-6p::GFP*) were age synchronized and exposed to control and *sams-1* RNAi from the L1 stage, 10 mM SAM was given from the L4 stage; *mip-1(Y67H2A.7)* RNAi was used as positive control for the UPR MT induction. GFP expression was assessed microscopically and quantified on AD2, each dot corresponds to one animal, *n* = 30 in each condition and representative images are shown in Supplementary Fig. 2b. **d** Schematic depicts the pathway of converting phosphoethanolamine (P-EA) to phosphocholine (P-Cho) in a SAM-dependent manner by phosphoethanolamine methyltransferases 1 and 2 (PMT-1 and 2); P-Cho is subsequently converted to phosphatidylcholine (PC) by the Kennedy pathway schematized in Supplementary

Fig. 3a. Created in BioRender (Ermolaeva, M. https://BioRender.com/8k33rq6). **e** Transgenic nematodes expressing GFP-tagged mitochondria in the body wall muscle (*myo-3p::gfpmit*) were age-synchronized and exposed to *pmt-1* RNAi or empty vector control from the L1 stage. Mitochondrial morphologies were scored on AD2 as in Fig. 3b, *n* = 60 in each experimental group. The representative images are shown in Supplementary Fig. 3c. **f** Transgenic nematodes expressing GFP under control of the *hsp-6* promoter (*hsp-6p::GFP*) were age synchronized and exposed to control and *pmt-1* RNAi from the L1 stage; *mip-1(Y67H2A.7)* RNAi was used as positive control for the UPR MT induction. GFP expression was assessed microscopically and quantified on AD2, each dot corresponds to one animal, *n* = 30 in each condition and representative images are shown in Supplementary Fig. 3d. **g** WT and mito-mutant animals were age synchronized and exposed to control, *sams-1*, *pmt-1* and *pmt-2* RNAi from L1 larval stage, survival was scored daily, *n* = 140 in each experimental condition. Statistics in **b**, **c**, **e** and **f** were assessed using unpaired t-test with Welch's correction, mean and SEM values are presented; two-tailed *p* values were calculated in all cases. For **g** statistical analysis was performed using Mantel-Cox test with two-tailed *p* values. *-*p < 0.05*; **-*p < 0.01* and ****-*p < 0.0001*. Exact *p* values can be found in the Source Data file. Each experiment was repeated at least 3 times, and one representative result is shown in each case.

methylation-dependent PC synthesis pathway results in a systemic reduction of PC levels, consistent with prior observations for *sams-1* deficiency[40]. We show that this biosynthetic perturbation causes mitochondrial network fragmentation and a corresponding decrease in mitochondrial functional capacity.

### Ectopic PC supplementation and boosting of PC synthesis alleviate- mitochondrial aging and restores metabolic resilience

We next asked if the reduction of methylation-dependent PC synthesis was a relevant phenomenon in the context of normal aging possibly contributing to aging-associated mitochondrial and cellular dysfunction[3]. We first re-examined the most outstanding protein changes occurring in aging WT nematodes and notably found PMT-1, PMT-2 and SAMS-1 to be the top 3 strongest downregulated proteins at the advanced age (AD10) (Supplementary Data 27). Moreover, similar to changes shown for SAMS-1 in Figs. 2a, b the downregulation of PMT-1 and PMT-2 expression was discovered to be progressive with age (Fig. 6c and Supplementary Fig. 8a, Supplementary Data 4). These findings uncovered methylation-dependent PC synthesis as one of the strongest aging-inhibited pathways in *C. elegans*. Our follow up mass spectrometry analysis revealed that the levels of PC were lower in post-reproductive animals, than the levels of PE (Supplementary Fig. 8b and c, Supplementary Data 28), while PC but not PE levels were boosted by choline supplementation (Supplementary Fig. 8b and c, Supplementary Data 28). Accordingly, PC/PE and LPC/LPE ratios were elevated in post-reproductive worms supplemented with choline throughout adulthood (Fig. 6d, Supplementary Data 28). This result is in line with our previous data showing reduction of PC and LPC levels with age[3], and demonstrates that the synthesis of PC, and particularly its methylation-dependent component, declines with aging. Furthermore, it shows that choline supplementation can effectively enhance in vivo PC levels in the context of aging. Although components of the Kennedy pathway involved in the conversion of choline to PC show a trend toward age-related decline (Supplementary Fig. 8d and Supplementary Data 29), neither the cumulative change nor the expression changes of individual enzymes are comparable in magnitude to the coordinated downregulation of SAMS-1, PMT-1, and PMT-2 observed with age (Supplementary Data 27). This difference may explain the continued efficacy of choline supplementation in the aging context.

A reduction of SAM-dependent PC synthesis was previously linked to pathological elevation of triglyceride (TAG) storage in the rat liver[47], and increased abundance/size of TAG containing lipid droplets was seen in both *sams-1* and *pmt-1* deficient nematodes[40,44,48]. In earlier work, we showed that vitellogenin expression closely correlates with the levels of lipid droplet triglycerides (TAGs) and other lipid droplet

components, such as short-chain dehydrogenases[3], and indeed vitellogenin levels progressively increased with age in WT *C. elegans* (Supplementary Fig. 9a and Supplementary Data 30). This finding coincided with aging-triggered decline of SAMS-1, PMT-1 and PMT-2 protein levels (Fig. 2a, b; Fig. 6c and Supplementary Fig. 8a), suggesting that reduced PC synthesis may contribute to the de-regulation of organismal lipid storage also in the context of aging. However, vitellogenins are also major components of yolk granules in nematodes and can accumulate in the pseudocoelomic cavity of aging worms, so these observations should be interpreted with caution.

Previous tests in young nematodes implicated increased UPR[ER] activity in connecting lowered PC levels to elevated lipid storage, demonstrating increased expression of ER chaperones HSP-3 and HSP-4 in *pmt-2* deficient *C. elegans*[48]. However, we found that HSP-3 and HSP-4 levels decline with age in WT nematodes (Supplementary Fig. 9b, c; Supplementary Data 4), suggesting that the UPR ER does not mediate the link between reduced PC synthesis and the age-associated increase in lipid droplet abundance, the latter inferred from vitellogenin expression and previous reports. On the other hand, our earlier work implicated aging-associated mitochondrial dysfunction in driving the TAG storage abnormalities in late life[3]. We thus hypothesized that aging-triggered reduction of PC synthesis may cause alterations of cellular lipid turnover via interference with mitochondrial integrity. Notably, the increased mitochondrial fragmentation and reduced network integrity seen in *sams-1*, *pmt-1* and *pmt-2* KDs (Fig. 3a, b; Fig. 4e and Supplementary Fig. 3c, and Supplementary Fig. 7a, b) are well known features of aged mitochondria[3,49]. In the next experiment we on the one hand validated the previously described increase of mitochondrial fragmentation during normal aging[3,49], and on the other hand determined that dietary choline supplementation indeed alleviates this aging-triggered alteration (Fig. 6e and Supplementary Fig. 10a). Consistently, choline supplementation improved mitochondrial OCR in post-reproductive WT animals as seen by a Seahorse test (Fig. 6f). Baseline OCR was lower in these animals compared to young worms (Fig. 5d, e) similar to our previous report[50], although the measurements were performed independently and under distinct bacterial diet conditions in this case. Notably, the rescue of mitochondrial morphology by choline supplementation during aging was less pronounced than that achieved with PC in *sams-1* and *pmt-1* knockdown worms (Fig. 5a, b). We believe this is because, in the knockdowns, the defect is explicitly caused by a decline in PC synthesis, whereas during aging, the PC decline is only a strong contributing factor, with additional impairments present that cannot be rescued by choline or PC provision. Collectively, these data show that a reduction of methylation-dependent PC synthesis contributes to loss of

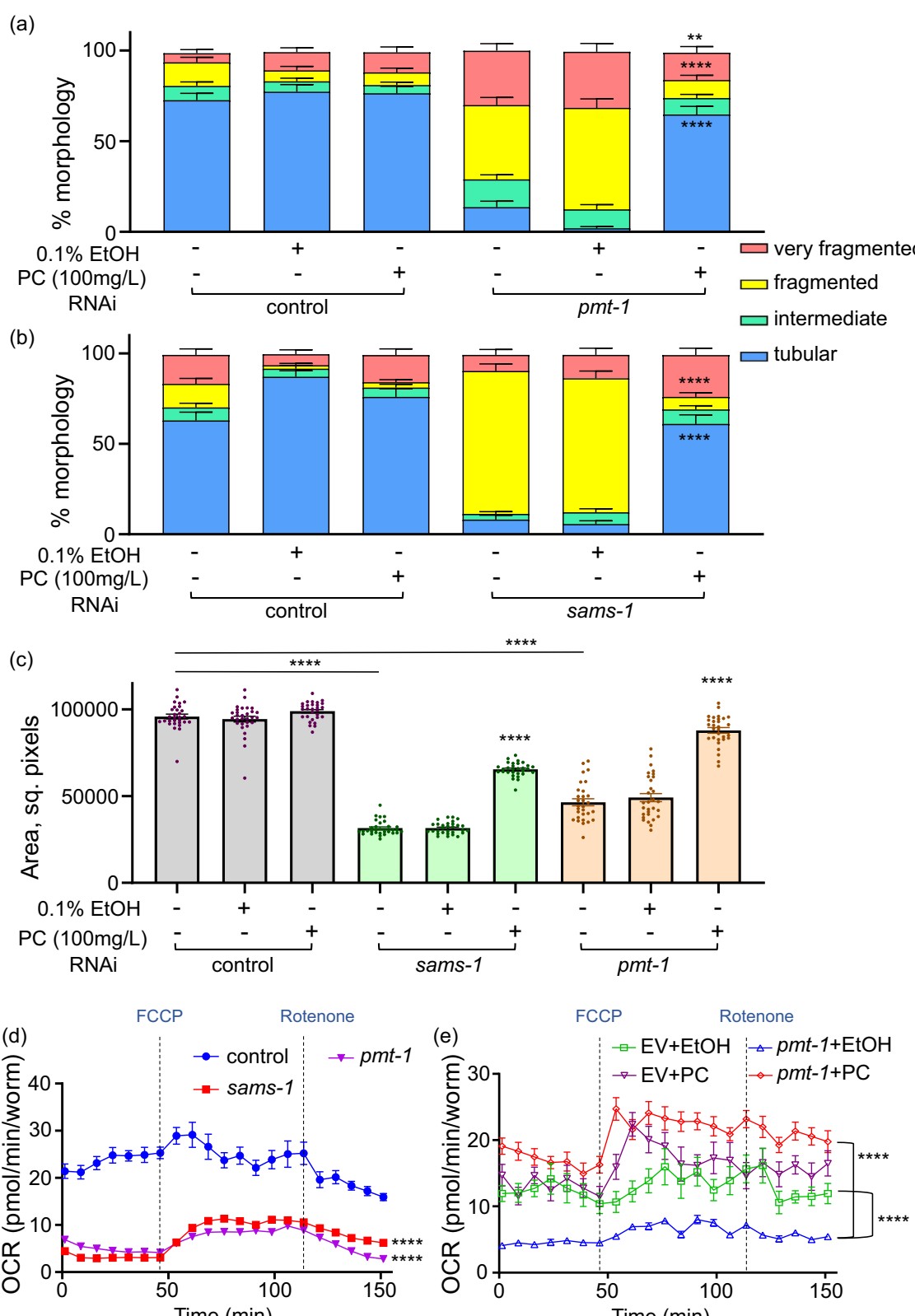

mitochondrial integrity and function during normal aging with possible broader effects on cellular lipid storage. We also demonstrate that the aging-linked alteration of mitochondrial fusion can be alleviated by dietary boosting of choline/PC levels.

Mitochondrial integrity and specifically mitochondrial fusion are important regulators of metabolic plasticity that are known to decline with age affecting cellular health and responses to drugs like metformin[2,3]. In particular, a reduction of mito-fusion contributes to the toxicity of mitochondrial inhibitor metformin in old organisms as seen in our previous work[3]. To probe the conservation of the links between PC levels and mitochondrial resilience, we next tested the effect of choline supplementation in the human cell culture model of metformin toxicity[3] in comparison or combination with the electron transport chain Complex II substrate succinate. The boosting of

**Fig. 5 | Phosphatidylcholine feeding rescues morphological and functional defects of mitochondria in *sams-1* and *pmt-1* deficient nematodes.**
**a, b** Transgenic nematodes expressing GFP-tagged mitochondria in the body wall muscle (*myo-3p::gfpmit*) were age-synchronized and exposed to *pmt-1* RNAi (**a**) or *sams-1* RNAi (**b**) with empty vector RNAi as control from the L1 stage, 100 mg/L phosphatidylcholine (PC) was provided from L4 stage and 0.1% ethanol (EtOH) was used as a vehicle control for PC. Mitochondrial morphologies were scored on AD2 as in Fig. 3b, *n* = 60 in all experimental groups. The representative images are shown in Supplementary Fig. 5a, b. **c** Age-synchronized *hsp-6p::GFP* nematodes were exposed to RNAi and compound treatment as in (**a, b**) and body size (area) was measured on AD2, n ≥ 29 with exact n number for each condition provided in the Source Data file; each dot corresponds to one animal. **d,e** Age-synchronized WT

nematodes were exposed to EV or *sams-1* or *pmt-1* RNAi from the L1 stage and mitochondrial oxygen consumption rate (OCR) was measured on AD2 by using a Seahorse platform. For **e** nematodes were exposed to 0.1% EtOH (vehicle) or PC (100 mg/L) from the L4 stage and analyzed on AD2. n ≥ 180 nematodes were used with exact n number for each condition provided in the Source Data file. Statistics in (**a**–**c**) was assessed using unpaired t-test with Welch's correction; In **d**–**e** the raw values were normalized to the number of animals measured. Area under the curve values were used to perform statistical analyses by unpaired *t*-test. Two-tailed *p* values are presented. n.s. - not significant; **-*p* < 0.01; ****-*p* < 0.0001. Each experiment was repeated at least 3 times, and one representative result is shown in each case. Exact *p* values, *n* numbers, mean and SEM values can be found in the Source Data file.

Complex II activity was chosen as it is expected to compensate for the inhibition of Complex I by metformin[51]. We found that choline protected metformin-exposed cells from cell death and loss of mitochondrial membrane potential, with most potent rescues delivered by a combination of both supplements (Fig. 6g and Supplementary Fig. 10b). These data demonstrate that PC repletion is protective against aging-induced mitochondrial fragmentation as shown in the nematodes and has a conserved restorative impact on metabolic plasticity.

Notably, unlike choline and PC, SAM supplementation had a variable effect across the different assays we carried out. For example, SAM supplementation was able to improve the mitochondrial network integrity in young *sams-1* deficient worms and old WT animals (Fig. 4b, and Supplementary Fig. 2a, 11e), while respirometry and body size improvements could not be seen (Supplementary Fig. 11c, d) and no measurable effect on PC and LPC content could be observed by lipidomics in *sams-1 KD* and *pmt-1* KD animals (Supplementary Fig. 11a and 11b). This was not due to SAM being metabolized and degraded by intestinal bacteria because the positive effect of the compound on mitochondrial morphology could be detected in the presence of both live and heat killed *E. coli* (Supplementary Fig. 6b), and we rather attribute this variability to the known poor stability and in vivo bioavailability of SAM[52]. To support this notion, the strongest effects of SAM could be seen in a low throughput assay that entailed the longest exposure of the animals to the compound, e.g., the tests of mitochondrial morphology during WT aging (Supplementary Fig. 11e) – 4 and 6 days of exposure, versus 2 days of exposure in Supplementary Fig. 11a–d. However, and most importantly, the mitochondrial morphology, PC content and body size defects of *sams-1* KD animals could be rescued by supplementation of PC or choline in all cases, clearly linking these defects to the alterations of PC synthesis.

## A decline in the phosphatidylcholine biosynthetic pathway correlates with parameters indicative of mitochondrial dysfunction during human aging

We next tested if the expression of PEMT - a functional analog of PMT-1/2 in humans[53] (Supplementary Fig. 3a), also declines during aging by analyzing the transcriptomic data of the Genotype-Tissue Expression (GTEx) Project[54,55]. Interestingly, a trend of PEMT down-regulation with age was observed across several human organs (Supplementary Fig. 12 and Supplementary Data 31, 32) with highest PEMT expressing tissues (top 25%) showing the most notable reduction (Fig. 7a, Supplementary Fig. 12, Supplementary Data 31 and 32). Of note, the especially affected cohort included organs with high lipid content such as subcutaneous and visceral adipose tissues (Fig. 7b and Supplementary Fig. 13a; Supplementary Data 33 and 34) and the ovary (Supplementary Fig. 13b and Supplementary Data 35). These observations suggested that the decline of methylation-dependent PC synthesis is occurring also in human aging at least in a subset of relevant tissues.

We next investigated whether phosphatidylcholine levels decline with age in humans by analyzing NMR metabolomics data from EDTA

plasma samples in the UK Biobank cohort[56,57] (Supplementary Fig. 14a and Supplementary Data 36). We found that total PC levels are gradually lowered in men at an advanced age (Fig. 7c, upper panel and Supplementary Data 37). Concurrently, relative PC levels (normalized to total fatty acid, TFA content) strongly declined in women after the approximate age of the menopause (Fig. 7c, lower panel and Supplementary Data 37), amid an overall increase in TFA levels suggesting increased adiposity (Supplementary Fig. 14b and Supplementary Data 38). Notably, menopause is known to be associated with the decline of energy levels and mitochondrial function in the females[58]. Meanwhile, upregulation of serum fatty acid (TFA) content is reminiscent of the enhanced abundance of lipid droplets observed in old *C. elegans* affected by a relative decline of PC synthesis[3] (Supplementary Fig. 9a). Notably, a decline of relative PC levels in aged females was paralleled by a reduction in relative polyunsaturated fatty acids (PUFAs) levels, while relative levels of monounsaturated and saturated fatty acids (MUFA and SFA respectively) increased (Supplementary Fig. 14b and Supplementary Data 38). Interestingly, similar to PC and LPC, PUFAs increase membrane curvature (a pre-requisite for fusion) while long-chain SFAs and in part MUFAs enhance membrane rigidity[37,59]. Overall, the in silico analysis of the NMR metabolomics data indicates that aging remodels the lipidome, likely leading to decreased membrane fluidity and curvature, with reduced PC levels being a contributor to these changes, and post-menopausal women affected the strongest.

We next asked if aging-relevant changes of PC, PUFAs, MUFAs and SFAs can be correlated with molecular or phenotypic indicators of mitochondrial impairment. For instance, mitochondrial dysfunction is compensated by increased glycolysis, resulting in elevated systemic lactate levels[60]. Consistently, we observed that high levels of phosphatidylcholine (PC) and polyunsaturated fatty acids (PUFAs) correlate with low lactate levels (Fig. 7d, e, Supplementary Fig. 15a, b and Supplementary Data 39, 40), whereas high levels of monounsaturated fatty acids (MUFA) and saturated fatty acids (SFA) are associated with elevated lactate levels (Supplementary Fig. 15a, b and Supplementary Data 40). These findings suggest that pro-curvature and pro-fluidity lipids may play a role in supporting mitochondrial function in vivo in humans. Similar results were obtained also for obesity and diabetes – the medical conditions known to be linked to mitochondrial and metabolic impairments[61], and to older age. We found that total and relative levels of PC and PUFAs were higher in lean and non-diabetic individuals (Fig. 7f, Supplementary Fig. 15b, 15c, and Supplementary Data 41, 42, 43), while obese and diabetic individuals exhibited higher relative levels of MUFAs and lower levels of PUFAs and PC (Fig. 7f, Supplementary Fig. 15b, 15c and Supplementary Data 41, 42, 43). We next correlated PC, MUFA and PUFA levels with indicators of healthy aging such as metabolic rate, co-morbidity index, walking speed and memory strength in humans. For instance, metabolic rate rises with mitochondrial uncoupling[62] and predicts mortality, particularly in aging[63]. High PC and PUFA levels negatively correlated with metabolic rate, while high MUFA levels showed a positive correlation (Fig. 7h and

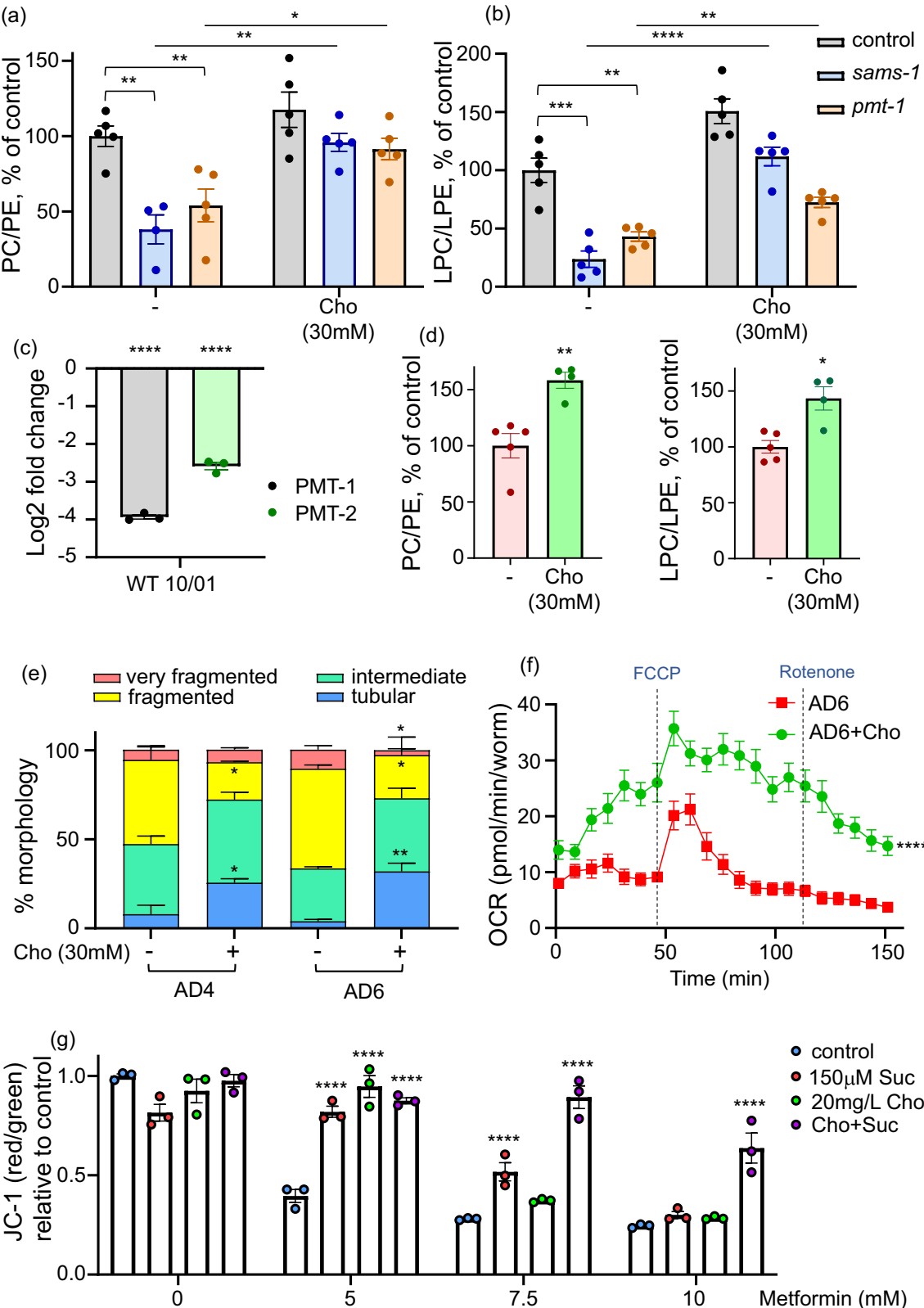

Supplementary Data 44). This pattern was mirrored in the comorbidity index (CCI), a mortality predictor[64], while walking speed and digit memory test results exhibited the opposite dynamics, e.g., positive correlation with PC and PUFA levels and negative—with MUFA levels (Fig. 7h and Supplementary Data 44). Collectively, our analysis of human data, though correlative and descriptive, indicates that phosphatidylcholine synthesis declines with aging, particularly in adipose-rich organs, and this decline is associated with lower serum PC levels, especially in postmenopausal women who are more susceptible to mitochondrial dysfunction. PC likely supports mitochondrial network integrity by contributing to membrane fluidity and curvature, as similar expression patterns and functional associations are observed for other lipid classes, such as PUFAs, MUFAs, and SFAs, which are also known to influence membrane fluidity properties.

**Fig. 6 | Declining phosphatidylcholine synthesis drives mitochondrial dysfunction during aging. a** WT animals were exposed to control (EV), *sams-1* and *pmt-1* RNAi from the L1 stage, with choline (Cho, 30 mM) provision initiated at the L4 stage, and absolute intensities of phosphatidylcholine (PC) and phosphatidylethanolamine (PE) were measured by ultra-performance liquid chromatography-tandem mass spectrometry on AD2 and normalized to the internal standard and the number of worms. PC/PE ratios are shown, normalized to EV, Cho- control. Data are expressed as mean ± SEM, *n* = 4 for the *sams-1* RNAi choline-free condition and *n* = 5 for all others with each sample containing 800 worms and each dot representing one sample. **b** Levels of lysophosphatidylcholine (LPC) and lysophosphatidylethanolamine (LPE) were measured in the same samples as in (**a**) and data were analyzed and presented as in (**a**); *n* = 5 in all cases with each sample containing 800 worms and each dot representing one sample. **c** Proteomics samples were collected as described in Fig. 1a and relative levels (expressed as Log2 fold change) of PMT-1 and PMT-2 proteins in old (AD10 vs AD1) WT animals are shown. Each dot represents an independent replicate of ≥500 worms (*n* = 3). All relevant calculations can be found in Supplementary Data 4. **d** WT animals were aged until AD6 with or without 30 mM choline (Cho) supplementation from the L4 stage; PC, PE, LPC and LPE levels were measured as in (**a**, **b**) and data were analyzed and presented as in (**a**, **b**); *n* = 5 for the choline-free condition and *n* = 4 for choline supplemented samples with each sample containing 800 worms and each dot representing one sample. Raw data and calculations for (**a**, **b** and **d**) can be found in Supplementary

Data 28. **e** Transgenic nematodes expressing GFP-tagged mitochondria in the body wall muscle (*myo-3p::gfpmit*) were age-synchronized and grown until AD4 or AD6 with or without 30 mM choline (Cho) supplementation from the L4 stage. Mitochondrial morphologies were scored on AD4 and AD6 as described in Fig. 3b, *n* ≥ 54 with exact n number for each condition provided in the Source Data file, representative images are shown in Supplementary Fig. 10a. **f** Age-synchronized WT nematodes were grown on OP50 *E. coli* and exposed to 30 mM choline (Cho) from the L4 stage and until AD6. Mitochondrial OCR was measured on AD6. *n* ≥ 190 with exact n number for each condition provided in the Source Data file. Raw values were normalized to the number of worms measured. **g** BJ human skin fibroblasts were treated with indicated concentrations of metformin with or without 150 μM succinate, 20 mg/L choline or a mix of choline and succinate. The mitochondrial membrane potential (JC-1 assay) was measured after 24 h; *n* = 3 in each condition. Significance was assessed using unpaired t-test with Welch's correction in (**a**, **b** and **d**) and by unpaired t-test in (**c**, **e**). For **f** area under the curve values were used to perform statistical analyses by unpaired t-test. For **g** multiple comparison t-test was used. Two-tailed *p* values were computed in all cases. *-*p* < 0.05; **-*p* < 0.01; ***-*p* < 0.001; ****-*p* < 0.0001. In **e**–**g** each experiment was repeated at least 3 times, and one representative result is shown; in **a**, **b** and **d** 4–5 independent biological replicas were measured for each condition; in **c** 3 independent replicas were measured. Exact *p* values, *n* numbers, mean and SEM values are presented in the Source Data file.

## Discussion

In this study, we explored natural mechanisms that contribute to the deterioration of mitochondria during normal aging in order to identify targets mendable by pharmacological or dietary interventions. While congenital failures of mitochondria are widely known to cause premature aging[2,5,6], the "natural causes" of mitochondrial aging remain poorly understood. We initially used longitudinal proteomics analysis in WT *C. elegans* and long-lived mitochondrial mutants, which was coupled to RNAi-mediated gene inactivation and longevity testing, to demonstrate that S-adenosylmethionine synthetase SAMS-1 is required for longevity maintenance in the context of mitochondrial impairments. Specifically, we could confirm previous reports of increased lifespan of WT animals exposed to *sams-1* RNAi[24,25], while discovering that the same RNAi exposure was detrimental for the longevity of long-lived *clk-1(qm30)* and *isp-1(qm150)* strains carrying hypomorphic mitochondrial mutations. Concurrently, we found SAMS-1 to be among the strongest downregulated proteins in old WT nematodes in line with previous observations[20–22], and same strong and progressive downregulation with age was discovered for phosphoethanolamine N-methyltransferases PMT-1 and PMT-2, which utilize S-adenosylmethionine (SAM, the product of SAMS-1) in the nematode pathway of methylation-dependent PC synthesis. We next explored the mechanistic basis of the longevity link between *sams-1* gene inactivation and mitochondrial impairments, and discovered that knockdowns (KDs) of *sams-1, pmt-1* and *pmt-2* genes cause an early life increase of mitochondrial fragmentation and a decline of mitochondrial respiration that are comparable to structural and functional alterations of mitochondria observed during normal aging[3,49]. Although elevated mitochondrial fission has previously been linked to *sams-1* inactivation outside the aging context[65], our data demonstrate that *pmt-1* and *pmt-2* also affect mitochondrial network integrity, thereby connecting all three knockdowns to mitochondrial regulation through their shared role in phosphatidylcholine production. Importantly we could alleviate these defects by dietary provision of PC[66] or choline (is converted to PC by the CDP-choline pathway[42]) both in the KDs and also during WT aging. The impact of aging and the above gene knockdowns on the abundance of PC and its derivative LPC were validated by lipidomics, and the restorative effect of choline supplementation was also detected in this case. While choline can enter several metabolic pathways once inside the cell, the collective evidence presented in this study suggests that the effects of choline provision on mitochondrial function are largely mediated through its

conversion to PC. Interestingly, we discovered by using the GTEx dataset (v8)[54] and previously pre-processed gene expression data[55] that levels of the human PMT-1/2 analog PEMT decline with age in several tissues and especially in organs showing overall highest PEMT expression. We followed up by analyzing the NMR metabolomics data of the UK biobank cohort to discover that PC levels decline with age also in humans and especially in post-menopausal females known to be affected by mitochondrial insufficiency[58]. This data indicates that aging-associated decrease of methylation-dependent PC synthesis is evolutionary conserved and may contribute to "natural" aging of mitochondria across species while other factors − such as the age-related upregulation of phospholipase activity[67,68], may also play a role in reducing PC levels with age. Subsequently, we used a human cell culture model of metformin toxicity to demonstrate that choline/PC repletion restores metabolic plasticity in human cells affected by mitochondrial stress.

In this study focusing on mechanisms of aging, we did not scrutinize the exact biochemical links between PC levels and mitochondrial morphology. Phosphatidylcholine is abundant in mitochondrial membranes and crucial for their function[34,69]. It interconverts with lysophosphatidylcholine (LPC) via enzymes like phospholipase A2 and Lysophosphatidylcholine acyltransferase (LPCAT), playing key roles in lipid remodeling and membrane homeostasis[45]. Both PC and LPC are known to contribute to membrane fluidity and curvature that are in turn supportive of membrane fusion[70]. Interestingly, our in silico analysis of the human NMR metabolomics data revealed that a decline of total and relative PC levels during aging coincided with a relative decline of PUFAs and an increase of MUFAs and SFAs known to facilitate membrane fluidity and rigidity respectively[71,72]. Aging therefore appears to cause lipidome remodeling towards the composition that may restrict membrane and mitochondrial fusion. Notably, high levels of PC and PUFAs and low levels of MUFAs and SFAs correlated with physiological markers of effective mitochondria, healthy metabolism and healthy aging, in line with our hypothesis. Additionally, phosphatidic acid (PA) that can be generated from PC by phospholipase D[73] is known to promote mitochondrial fusion both at the level of mitochondrial membrane structure (by creating negative membrane curvature) and by interacting with the fission and fusion machinery (PA interferes with Drp1-mediated fission and stimulates the activity of mitofusins-1 and -2)[74,75]. These observations and earlier findings provide the appropriate mechanistic context for the hereby discovered role of PC synthesis in the decline of mitochondrial plasticity during

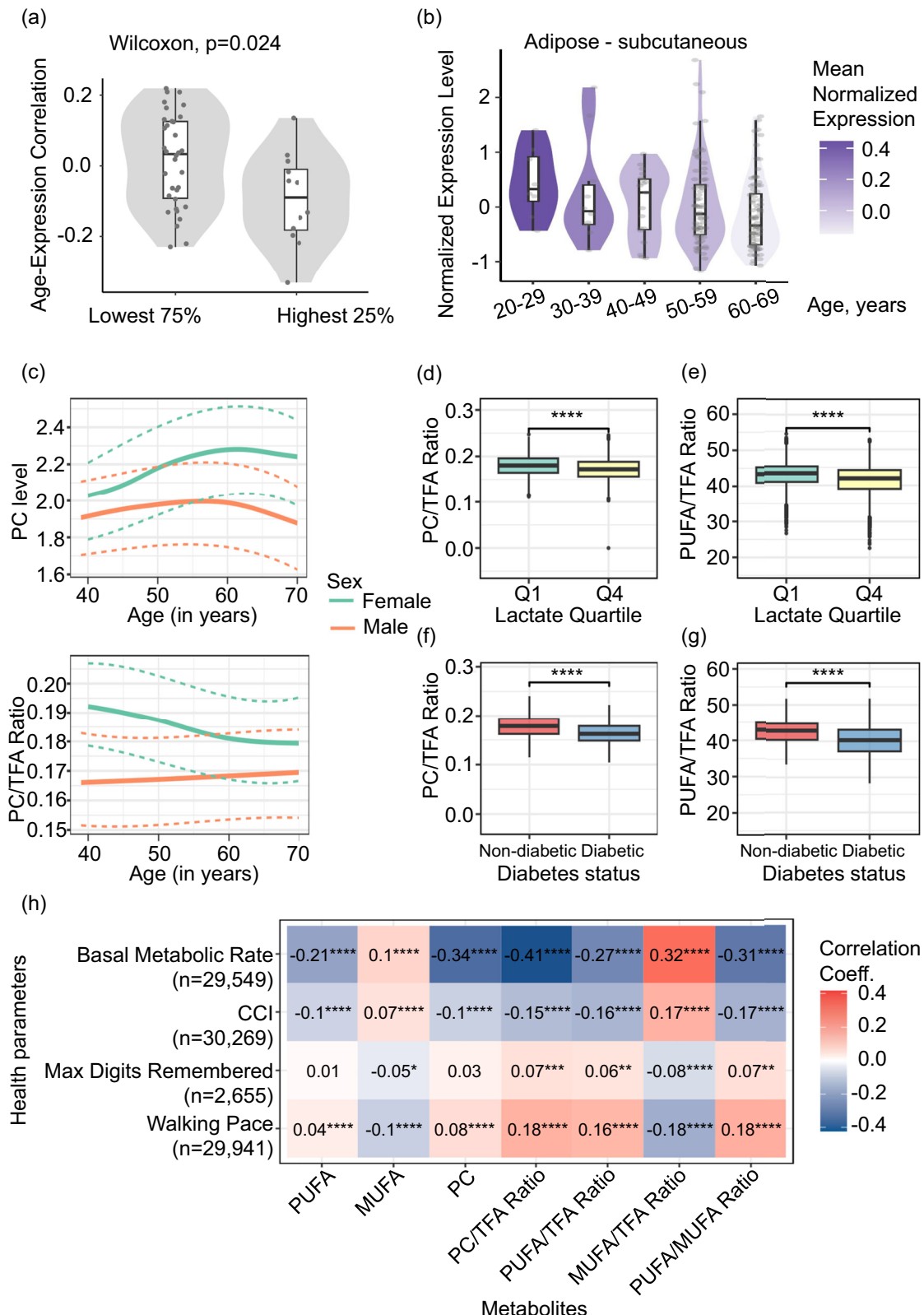

aging (Supplementary Fig. 16). In summary, we used a combination of omics and functional tests in two model systems (*C. elegans* and human) to identify aging-associated decline of methylation-dependent PC synthesis as a "natural" trigger of mitochondrial aging that is malleable by dietary choline and PC treatments. Notably, both choline and PC are applicable as dietary supplements in humans[76,77]. However, their metabolism by the human intestinal microbiome is likely more complex than in *C. elegans*, whose gut in laboratory settings contains only specific, single bacterial strains. In addition, PC can be converted to LPC and free fatty acids (FFA) during intestinal uptake and subsequently re-synthesized inside the cell[78]. For cases where any factors impede the ingestion or oral bioavailability of PC, alternative delivery methods—such as incorporation into lipid nanoparticle (LNP) formulations, may be considered.

**Fig. 7 | Lowered PC levels correlate with impaired metabolic health during human aging. a** The expression of PEMT gene across human tissues at different age was analyzed using the GTEx dataset (v8; *n* = 46 tissues). The selection of samples, details of data processing and exclusion criteria can be found in the "Methods section". Tissues with high (highest 25%) and low (lowest 75%) PEMT expression were determined as described in Supplementary Fig. 12, and age-related changes in each tissue were computed using Spearman's correlation between age and expression level across all available age points. Each dot represents one tissue, and significance was assessed by Wilcoxon test. The comparison of age-related changes between PEMT high- and low-expressing tissues is shown, and complete data can be found in Supplementary Data 31 and 32. **b** The expression of human PEMT gene in the subcutaneous adipose tissue at different age was tested using the GTEx dataset (v8; *n* = 192 samples) as described in (**a**). Normalized expression levels represent individual log2 transformed quantile normalized Transcripts Per Million (TPM) values corrected for sex and death circumstances using a linear model, each dot represents one sample; median normalized PEMT expression in each age group is color coded and complete data can be found in Supplementary Data 33. In **a**, **b** box edges represent the 25th and 75th percentiles, whiskers extend to 1.5× the interquartile range (IQR), whisker endpoints indicate the minima and maxima, and points beyond the whiskers denote outliers (**c**–**h**) NMR metabolomics and clinical data from the UK Biobank consortium were used for the respective analyses, as described in the "Methods section". **c** Age-associated changes in

phosphatidylcholine levels measured in mmol/l (PC, top panel) and as the ratio of phosphatidylcholine to total fatty acids (PC/TFA Ratio, bottom panel). The color coding reflects different sexes. The dashed lines show the first and fourth quartiles. The curves are created using the loess smoothing function. **d** Samples were sorted based on serum lactate levels, as shown in Supplementary Fig. 15a. Boxplots show the distribution of phosphatidylcholine to total fatty acids ratio (PC/TFA Ratio) for samples in the top 25% (i.e., Q1, *n* = 7556) and bottom 25% (i.e., Q4, *n* = 7559) lactate quartiles. The groups were compared using the Wilcoxon rank-sum test. **e** Similar to (**d**) except the y-axis shows the ratio of polyunsaturated fatty acids to total fatty acids (PUFA/TFA ratio). **f** Boxplots show the distribution of phosphatidylcholine to total fatty acids ratio (PC/TFA Ratio) for diabetic (*n* = 3100) and non-diabetic (*n* = 27,169) individuals. The groups were compared using the Wilcoxon rank-sum test. **g** Similar to (**f**) except the y-axis shows the ratio of polyunsaturated fatty acids to total fatty acids (PUFA/TFA Ratio). The center line in **d**–**g** represents the median. **h** Heatmap showing the Spearman's correlation between metabolites (x-axis) and health parameters (y-axis) where sample sizes are shown in parenthesis. The color coding reflects the correlation direction, and the color intensity reflects the correlation strengths, whereas the numbers on the heatmap show the exact value of Spearman's correlation coefficients; CCI - Charlson Comorbidity Index. The asterisk throughout the figure shows significance: *-$p < 0.05$; **-$p < 0.01$; ***-$p < 0.001$ and ****-$p < 0.0001$. All reported statistical tests are two-sided. Exact *p* values, *n* numbers, mean and SEM values are presented in the Source Data file.

Finally, the comparison of our data with the existing literature indicates that physiological stressors may be mitigated by diverse homeostatic pathways at different age. For instance, omics analyses of the same *C. elegans* strains as used in our proteomics study were already conducted earlier with the exclusive focus on young animals[9–12]. These earlier tests revealed increased activity of adaptive stress responses such as autophagy[12] and DAF-16/FOXO targets[11] as key longevity-relevant differences between WT nematodes and long-lived mito-mutants. The results of our longitudinal proteomics study agree with these findings in identifying a decline of proteostasis and stress responses as an early event during WT aging, and it is feasible that higher stress resilience confers a delayed onset of aging in the mito-mutants. However, as seen in our omics and functional data, when metabolic failures onset at the advanced age, their mitigation requires other protective mechanisms such as modulation of lipid synthesis and mito-morphology. Specifically, we hereby reveal PC synthesis as a suitable and malleable target of anti-aging treatments focused on post-middle-age patients.

## Methods

### *C. elegans* strains
The *C. elegans* strains were obtained from the Caenorhabditis Genetics Center (CGC) of the University of Minnesota, which is funded by NIH Office of Research Infrastructure Programs (P40OD010440), (www.cgc.umn.edu). Hermaphrodite nematodes were used in all tests, and maintained under standard conditions at 20ºC unless otherwise noted. We used the following strains: N2 Bristol (wild-type), MQ130 *clk-1(qm30)* III, MQ887 *isp-1(qm150)* IV, SJ4103: *zcIs14[myo3p::gfp(mito)]*, SJ4100: *zcIs13[hsp-6::gfp]*.

### Bacterial strains
Worms were fed with OP50 *E. coli* obtained from CGC. Unless otherwise explicitly stated, animals were cultured on Escherichia coli OP50 as a food source. Gene knockdowns were performed by feeding HT115 (DE3) L4440 *E.coli* obtained from ORF (*sams-1* RNAi) or Ahringer *C. elegans* RNAi collections (*mip-1(Y67H2A.7)* and *pmt-2* RNAi). *pmt-1* RNAi was kindly provided by Prof. O. Klotz (Friedrich Schiller University of Jena). All RNAi expressing plasmids were verified by sequencing. Growth conditions for both OP50 and RNAi bacteria were as described in (Espada et al., 2020).

### Preparation of heat killed bacteria
Single colonies of RNAi-expressing *E. coli* were grown in 30 mL LB with 90 μL of 100 mg/mL ampicillin at 37 °C, shaking for 12 h. Bacterial cultures were pre-induced by adding 15 mL LB and 90 μL of 1 M IPTG, followed by incubation at 37 °C for 25 min. Cells were pelleted at 3500 rpm for 10 min, and some supernatant (typically 30 mL) was removed. The bacterial pellet was resuspended in the remaining LB to OD600 = 3. The culture tube was sealed with film and plastic wrap, then fully submerged in an 80 °C water bath for 1 h, with shaking every 15 min. After heat treatment, cells were pelleted again at 3500 rpm for 10 min and resuspended in fresh LB. For RNAi feeding, 850 μL of the heat-killed bacterial suspension was seeded onto 10-cm IPTG plates (350 μL for 6-cm plates) and dried overnight at room temperature. Here and throughout, "room temperature" refers to conditions where temperature was maintained by the building's air conditioning rather than by an incubator. In these cases, room temperature typically fluctuated between 20 °C and 23 °C. Complete bacterial inactivation was confirmed by inoculating 10 μL of the heat-treated suspension into LB and monitoring OD after overnight incubation. This procedure was developed by the Ermolaeva lab through the integration of established knowledge and relevant insights from the literature.

### Treatment with drugs and supplements
For worms, SAM (Hanoju) was extracted from capsules, dissolved in Milli Q® water, filtered with 0.22 μm filter to remove microcellulose impurities contained in the capsules, and applied on top of the bacterial lawn to reach a final concertation of 10 mM, considering the 10 ml NGM agar volume. Phosphatidylcholine (PC) concentration and application was chosen according to a previous report (Kim et al.). Briefly, PC was dissolved in EMPROVE®exp ethanol (Merck) and applied to NGM plates at a final concentration of 100 mg/L PC in 0.1% ethanol. Higher ethanol concentrations were avoided, as they caused significant mitochondrial fragmentation in pilot experiments. PC and its vehicle control were added directly on top of the bacterial lawn. Both SAM and PC supplemented plates were kept under light-protective conditions. Choline (Sigma), was given to worms by addition to the NGM agar. In cell culture tests, 150 μM succinate (Oroboros, MitoKit-CII), 30 mM choline and 7.5 mM and 10 mM metformin (M2009 TCI) were added directly to culture media.

## Lifespan assays

Lifespan assay was performed as described in (Espada et al., 2020). In brief, age-synchronized worms were maintained on NGM agar plates seeded with *E. coli* OP50 at 20 °C for lifespan analysis. For RNAi conditions, plates contained 1.5 mM IPTG and 0.2 mg/mL ampicillin, and worms were fed *E. coli* HT115(DE3). Worms were transferred to fresh plates every 2 days during reproduction and every 4 days post-reproduction, and viability was assessed daily by monitoring movement.

## Assessment of the mitochondrial oxygen consumption rate (OCR)

Wild-type nematodes were age-synchronized and fed with empty vector or RNAi from L1 stage at 20 °C. All supplements were given from the L4 stage until the end of each experiment. Mitochondrial OCR measurements were performed on day 2 of adulthood (AD2) for gene knockdown comparisons as well as for SAM and PC supplementation tests. For the experiments with choline supplementation, mitochondrial OCR was measured on day 6 of adulthood (AD6). To maintain the nematodes until AD6 without contamination from the progeny, the choline treated and untreated worms were washed and transferred to new plates on AD2 and AD4. For OCR measurements, worms were rinsed off the plates and washed three times with 1X M9 buffer to remove traces of bacteria and progeny. Worms were then suspended in 1 ml μQ water supplemented with salts (60 mg $MgSO_4$, 60 mg $CaSO_4$ and 4 mg KCl per 1 L water) and transferred to wells of a 96 well plate (20-30 worms/well). FCCP (Sigma, Cat No 2920) and Rotenone (Sigma, Cat No R8875) and Antimycin A (Sigma, Cat No A8674) were dissolved in 2% DMSO (Sigma, Cat No F7524) prior to assessment and diluted in water to obtain the working concentration of 200 μM and 50 μM, respectively. The assay was performed at room temperature using Agilent 4 Seahorse XF Pro analyser. Data was normalized to the number of worms.

## Imaging and quantification

In testing effects of gene inactivation on mitochondrial morphology and stress levels, *zcIs14[myo3p::gfp(mito)]* and *zcIs13[hsp-6::gfp]* reporter strains were fed with respective RNAi or EV control bacteria from L1 stage and until imaging on adulthood day 2 (AD2). Drug and vehicle treatments were delivered from L4 stage and until imaging at AD2. For mitochondrial morphology quantification, *zcIs14[myo3p::gfp(mito)]* worms were immobilized in 12.5 mM levamisole on 3% agarose pads and imaged in vivo using an AXIO Imager M.2 microscope (ZEISS). Morphology was assessed in the body wall muscles of the tail area. x63 magnification, oil immersion and temperature-controlled environment (15–19 °C) were applied. For high-resolution imaging of mitochondrial morphology, worms were mounted on 1.5 mm agar pads and imaged using a Zeiss AxioImager Z2 microscope equipped with a 100× oil immersion lens and ApoTome function. Images were acquired in auto-exposure mode using ZEN software, with exposure times ranging from 200 ms to 800 ms. For mitochondrial stress imaging, *zcIs13 [hsp-6p::gfp]* transgenic worms were immobilized by placing plates on ice for 10 min, followed by stacking worms, and imaging with an AXIO Zoom.V16 microscope (ZEISS) equipped with an Axiocam 503 camera and REO Objective 5281-9150-000 at 100× total magnification (Zoom 4.2). Exposure times were 7 ms for brightfield and 90 ms for EGFP. Images were analyzed using ImageJ 1.52n to quantify average EGFP intensity per worm. Each individual worm perimeter was encircled, and mean fluorescence was measured followed by background subtraction in Fiji - ImageJ software. Area of each worm calculated using Fiji-Image J was used to compare the size of the worms when necessary. In testing the effect of choline supplementation on mitochondrial aging, *zcIs14[myo3p::gfp(mito)]* worms were grown under standard conditions from L1 stage and until adulthood day 1 (AD1) followed by choline or control exposure and imaging at indicated times, the imaging was

performed as described above. The % of tubular, intermediate, fragmented and very fragmented mitochondria per worm was calculated based on following criteria: continuous, elongated formations with minimal gaps were classified as tubular; cells containing small, scattered mitochondrial units without a defined network were categorized as fragmented; those exhibiting a mix of tubular and fragmented mitochondria were scored as intermediate; and cells showing fragmented mitochondria with a visibly reduced mitochondrial content were designated as very fragmented. The same distal muscle cells were examined in each specimen, with four cells assessed per worm. The scoring procedure was adapted from Regmi et al., 2014 with modifications. In all cases, at least 20 worms per condition were used in 3 replicas (individual plates) and all experiments were repeated at least 3 times.

## Quantitative PCR with reverse transcription

Total RNA was isolated from worms for cDNA synthesis, which was used for real time quantitative PCR, with data analysed using the ΔΔCt method (Livak & Schmittgen, 2001). Briefly, 60 worms at L4 stage were washed with M9 buffer, lysed in TRIzol ™ (Sigma-Aldrich), homogenized using ceramic beads (biostep GmbH) and frozen. Total RNA was isolated using 1-bromo-3-chloropropane (Merck) and the Analytik Jena Kit (#845-KS-2040050), followed by treatment with DNase I RNase free (Thermo Scientific™) and RiboLock RNAse Inhibitor (Thermo Scientific™). First-strand cDNA synthesis was performed from 0,5 μg RNA using SuperScript™ III Reverse Transcriptase kit (Invitrogen). Real time PCR was carried out using Bio-Rad SsoAdvanced™ Universal SYBR® Green Supermix 2x (Bio-Rad) and CFX 96™ Real-Time System with C1000 Touch™ Thermal Cycler. Mean value of 3 housekeeping genes - actin, tubulin and laminin, were used as a reference, the primer sequences can be found in Supplementary Data 45.

## Sample preparation for proteomics analysis

For each sample, 700 synchronized L1 larvae were seeded with the expectation that at least 500 animals would be recovered at the indicated age, based on prior experience. Worms were washed five times with M9 buffer (three washes in 10 mL followed by two washes in 5 mL) to remove eggs and progeny, and were then collected directly into bead-beating tubes to minimize losses during transfer. Lysis buffer (1x) was 1 % SDS, 100 mM HEPES, pH 8, 100 mM DTT, 1 mM EDTA. Samples were bead-beaten in a Precellys bead-beater at 6000 rpm, 2×20 sec, 30 sec interval at 4 °C. Samples were centrifuged (1000 g, 3 min) and the supernatant pipetted into fresh 0.5 mL Eppendorf tubes®. Samples were then sonicated using a Bioruptor (60 sec ON/30 sec OFF, 10 cycles, high intensity at 20 °C) (Diagenode, Belgium), then heated to 95 °C for 10 min, before a repeat set of sonication cycles as before. Alkylation to block cysteines was carried out with iodoacetamide (15 mM final concentration, 30 min, dark, room temperature). Protein precipitation was carried out on an estimated 25 μg amount of each sample using 4x sample volume of ice-cold acetone, and samples were left at -20 °C overnight. The following day, samples were centrifuged (20800 g, 30 min, 4 °C), supernatant carefully removed, and protein pellets washed twice with ice cold 80 % acetone/20 % water (300 μL, 10 min centrifugation (as above). After removal of the second wash, pellets were air-dried before resuspension in the digestion buffer (3 M urea in 0.1 M HEPES, pH 8; LysC (1:100 enzyme: protein ratio), then incubated for 4 h at 37 °C. The samples were diluted 1:1 with Milli-Q® water (to reach 1.5 M urea) and incubated with trypsin (1:100 enzyme: protein) for 16 h at 37 °C. The digests were then acidified with 10 % trifluoroacetic acid and then desalted with Waters Oasis® HLB μElution Plate 30 μm (Waters Corporation, Milford, MA, USA) in the presence of a slow vacuum. In this process, the columns were conditioned with 3 × 100 μL solvent B (80 % acetonitrile; 0.05 % formic acid) and equilibrated with 3 × 100 μL solvent A (0.05 % formic acid in Milli-Q® water). The samples were loaded, washed 3 times with 100 μL solvent A, and

then eluted into PCR tubes with 50 μL solvent B. The eluates were dried down with the speed vacuum centrifuge prior to resuspension in MS Buffer (5 % acetonitrile, 95 % Milli-Q® water, with 0.1 % formic acid) for the MS analyses.

## Proteomics data aquisition in data dependent (DDA) and independent (DIA) modes

Data were acquired on a QE-HFX MS (Thermo), connected to an M-Class nanoAcquity (Waters). The outlet of the analytical column was coupled directly to the mass spectrometer using the Proxeon nanospray source. The trapping column was nanoAcquity Symmetry C18, 5 μm, 180 μm × 20 mm and the analytical column was nanoAcquity BEH C18, 1.7 μm, 75 μm × 250 mm. Solvent A was water, 0.1 % formic acid, and solvent B was acetonitrile, 0.1 % formic acid. -1 μg of each of the samples (reconstituted at 1 μg/ μL and spiked with HRM kit peptides (Biognosys AG, Switzerland)) were injected for LC-MS with a constant flow of solvent A, at 5 μL/min, in trapping mode. The trapping time was 6 minutes. Peptides were eluted via the analytical column with a constant flow of 0.3 μL/min. During the elution step, the percentage of solvent B increased in a non-linear fashion from 0 % to 40 % in 120 minutes. Total runtime was 145 minutes, including clean-up and column re-equilibration. The peptides were introduced into the mass spectrometer via a Pico-Tip Emitter 360 μm OD × 20 μm ID; 10 μm tip (New Objective) and a spray voltage of 2.2 kV was applied. The capillary temperature was set at 300 °C. The ion funnel RF was set to 40 %. DDA data were acquired on a subset of samples from all conditions, with the following settings: Full scan MS spectra with mass range 350-1650 m/z were acquired in profile mode in the Orbitrap with resolution of 120000. MS1 fill time was 20 ms, AGC target $3 \times 106$. Top N was used (=15) and the intensity threshold was $4 \times 104$. Normalized Collision Energy (NCE) with HCD was set to 27 % and a 1.6 Da window was used for quadrupole isolation. MS2 data were acquired in profile mode from 200–2000 m/z. MS2 fill time was 25 ms or an AGC target of $2 \times 105$. Only 2-5+ charge states were selected for MS/MS. Dynamic exclusion was 20 s, and the peptide match 'preferred' option was selected. For the DIA data, LC conditions remained unchanged. Full scan MS spectra with mass range 350–1650 m/z were acquired in profile mode in the Orbitrap with resolution of 120000.The default charge state was set to 3 +. The filling time was set at maximum of 60 ms with limitation of $3 \times 106$ ions. DIA scans were acquired with 34 mass window segments of differing widths across the MS1 mass range. HCD fragmentation (stepped normalized collision energy; 25.5, 27, 30 %) was applied and MS/MS spectra were acquired with a resolution of 30000 with a fixed first mass of 200 m/z after accumulation of $3 \times 106$ ions or after filling time of 40 ms (whichever occurred first). Data were acquired in profile mode. For data acquisition and processing of the raw data Xcalibur 4.0 (Thermo Scientific) was used in parallel with Tune version 2.9.

## Proteomics data analysis

For library creation, the DDA and DIA data were searched independently using the Pulsar search engine (Spectronaut™ (version 11.0.15038.17.27438); Biognosys AG, Zurich, Switzerland). Afterwards, the two libraries were merged to create a single DpD (DDA plus DIA library). In both cases, data were searched against a species specific (*C. elegans*) UniProt database (as for the digest check) alongside the database of common contaminants. The data were searched with trypsin/P specificity and the following modifications: Carbamidomethyl (C) (Fixed) and Oxidation (M)/ Acetyl (Protein N-term) (Variable). A maximum of 2 missed cleavages was allowed. The identifications satisfied an FDR (false discovery rate) of 1 % on both peptide and protein level. All other settings were the defaults from Biognosys.

The resulting library contained 92001 precursors, corresponding to 5338 protein groups using Spectronaut™ protein inference. DIA data were then uploaded and searched against this spectral library in

Spectronaut™. Relative quantification was performed in the software for each pairwise comparison using the replicates from each condition. Spectronaut™ ran a pairwise comparison at the peptide level and then summarized it at the protein level. Differences in protein abundances were statistically determined by Spectronaut™ software using the Student's t-test and multiple testing correction algorithm described by Storey (2002). Contrast table (candidates table) and protein matrix table were then exported for further data visualization analysis using in-house scripts with R-studio (version 1.0.153). The further used for analysis here Q value represents the false discovery rate for multiple comparisons.

## Proteomics data availability

The mass spectrometry proteomics data have been deposited to the ProteomeXchange Consortium (Deutsch et al., 2020) via the PRIDE (Perez-Riverol et al., 2019) partner repository available at (http://www.ebi.ac.uk/pride), with the dataset identifier PXD024180.

## Identification of relevant proteome changes

Relative quantification was performed in Spectronaut for each pairwise comparison among N2 and mito-mutants at AD1, AD5 and AD10 using the triplicate samples from each condition. The data (candidate table with log2 (fold changes) and Q values for each pairwise comparison, and raw intensities when applicable) were exported to Excel for further analyses presented as Venn diagrams, pie charts, bar charts and boxplots. Visualization was performed using Graph Prism 8.0.0.

The Venn diagrams compare lists of genes differentially expressed in N2 and mitomutants at AD1, AD5 and AD10. The online tool (http://bioinformatics.psb.ugent.be/webtools/Venn/) was used to create the diagrams from pairwise comparisons subsetted in respective Supplementary Data files. For pathway analysis of overlapping and non-overlapping protein lists, the WormCat 2.0 online tool (www.wormcat.com) was used. The annotation version applied was ORF_only_v2_nov-11-2021.csv, and all available reporting levels are presented in the corresponding Supplementary Data files (Supplementary Data 9–11, 15-16, 20–21, and 24–26). The protein lists, formatted for direct use in WormCat, are provided in Supplementary Data 8, 14, 19, and 23. The highlights (captions) in Fig. 1b–e reflect all available reporting levels, whereas the pie charts in Fig. 1f–h are based on Category 3 reporting, as presented in the Supplementary Data files and described in the respective figure legend. Complete WormCat output folders, including native graphics for all Category levels, are provided in the Supplementary Data 46 for all analyses shown in this study.

## Cell culture

BJ human skin fibroblasts (low to medium passage) were cultured in DMEM high-glucose medium containing L-glutamine and pyruvate, supplemented with 10% FBS (all reagents from Sigma). For cell-culture assays, cells were seeded in 96-well flat-bottom plates at a density of 20,000 cells per well in complete DMEM. After overnight incubation to allow for proper attachment, cells were treated with indicated supplements for 24 h. For treatments with metformin (with or without supplements), cells were washed with HBSS, and the medium was replaced with glucose-free, L-glutamine-free, and phenol red-free DMEM, supplemented with 10% FBS. All cell culture and treatments were conducted in a humidified incubator at 37 °C with 5% $CO_2$ (BBD6220, Thermo Scientific).

## Lactate Dehydrogenase (LDH) cytotoxicity assay

Cell death was quantified using the Pierce LDH Cytotoxicity Assay Kit (Thermo Scientific) and following procedures described in the kit. Following treatment, 50 µl of culture medium from each sample was transferred to a new flat-bottom 96-well plate and combined with 50 µl of reaction mixture. After a 30-minute incubation at room temperature, 50 µl of STOP Solution was added to each well. Absorbance was

measured at 490 nm, with background subtraction at 680 nm, using a plate-reading spectrophotometer (Infinite M200 Pro, Tecan). The percentage of cytotoxicity was calculated according to the manufacturer's instructions.

## Mitochondrial membrane potential assay

Drug-induced changes in mitochondrial membrane potential (MMP) were evaluated using the JC-1 dye (Thermo Fisher) following the manufacturer's protocol. Briefly, after treating cells in 96-well plates, JC-1 was added to each well to achieve a final concentration of 5 µg/ml (pipetting 10 µl of a 55 µg/ml JC-1 stock solution). The plate was incubated at 37 °C for 15 minutes, protected from light using aluminum foil. Following incubation, the JC-1-containing medium was removed, cells were washed once with HBSS, and 50 µl of JC-1-free growth medium was added to each well. Fluorescence intensities were measured using a plate-reading spectrophotometer (Infinite M200 Pro, Tecan). In healthy mitochondria, JC-1 forms red-fluorescent J-aggregates, while in cells with low MMP, JC-1 remains as green-fluorescent monomers. MMP loss was quantified by calculating the red-to-green fluorescence ratio of JC-1.

## Analysis of Human Transcriptomic Data

PEMT expression and age-related variations across tissues were analyzed using the GTEx dataset (v8) (https://www.nature.com/articles/ng.2653) and preprocessed data from Donertas et al. 2021 (https://www.nature.com/articles/s43587-021-00051-5). These data incorporated individuals with Hardy Scale scores 1 and 2, filtered out genes with a median TPM (transcript per million) below 1, applied log2 transformation, and quantile normalized the dataset after correcting for sex and death circumstances using a linear model. Samples deviating more than 3 standard deviations from the first four principal components were considered outliers and excluded. Preprocessing scripts are available in Github (https://github.com/mdonertas/aging_in_GTEx_v8). Age-related expression changes were calculated using Spearman's correlation for individuals for tissue-age pairs with more than five samples. Tissues with high PEMT expression were determined by computing mean TPM values, following the method employed in GTEx data portal but restricting data to same individuals used in our main analysis (i.e., excluding slow death and ventilator cases and outliers). All resources, including code, processed data, and results, can be accessed at (https://github.com/mdonertas/pc_synthesis_human_aging).

Code for PEMT analysis is at (https://github.com/mdonertas/pc_synthesis_human_aging). Code for GTEx data preprocessing is at (https://github.com/mdonertas/aging_in_GTEx_v8).

## Phospholipidomics

Worms were age-synchronized and exposed to EV, *pmt-1*, or *sams-1* RNAi from the L1 stage, followed by treatment with either vehicle or choline from the L4 stage and until day 2 of adulthood (AD2). For long-term choline supplementation, wild-type animals were treated with vehicle or choline from the L4 stage until measurement on AD6. Each replicate sample contained 800 worms, and in the case of long-term choline treatment, animals were separated from progeny by regular washing and transfer to fresh plates. For sample collection, worms were washed with M9 medium and shock-frozen in 500 µl medium. Lipids were extracted by sequential addition of methanol, chloroform, and saline (final ratio of 14:34:35:17) using 1-pentadecanoyl-2-oleoyl(d7)-sn-glycero-3-phosphocholine (PC(15:0_18:1-d7; Merck) and 1-pentadecanoyl-2-oleoyl(d7)-sn-glycero-3-phosphoethanolamine (PE(15:0_18:1-d7; Merck) as internal standards[3,79]. Using previously established settings[80], lipids were separated on a reversed-phase column (ACQUITY UPLC BEH C8; 1.7 µm, 2.1 × 100 mm; Waters) with an ACQUITY UPLC system (Waters). The LC system was coupled to a QTRAP 6500+ mass spectrometer (Sciex) equipped with an IonDrive

Turbo V ion source and a TurboIonSpray probe, which was operated by Analyst 1.7.1 (Sciex). PC, PE, LPC, and LPE were measured by multiple reaction monitoring in the negative ion mode, and the extracted chromatograms were processed using Analyst 1.6.3 (Sciex)[80,81].

Phospholipids were detected after fragmentation to one fatty acid anion (LPC, LPE) or two fatty acid anions (PC, PE), the latter with quantitation based on the mean of both transitions[79]. The absolute amounts of PC and PE were calculated by summing the signal intensities of all analyzed lipid species of the respective lipid class (including lysophospholipids) and normalizing them to the subclass-specific deuterated internal standard and the number of worms. For the quantitation of LPC and LPE, only the lysophospholipid fraction of the respective lipid class (PC or PE) was considered. PC/PE and LPC/LPE ratios were calculated from absolute amounts and are expressed as a percentage of non-treated empty vector controls (100%).

## UK Biobank human data analysis

The UK Biobank data was downloaded using Bash, following the guidelines provided by the UK Biobank. We accessed the data through the application number 129228. Lipid levels were measured in EDTA plasma using a high-throughput NMR-based metabolic biomarker profiling platform and they are described below along with their UK Biobank IDs.

- PC: Phosphatidylcholine level (measured in mmol/l) (UK Biobank ID: 23437)
- PUFA: Polyunsaturated fatty acid level (measured in mmol/l) (UK Biobank ID: 23446)
- MUFA: Monounsaturated fatty acid level (measured in mmol/l) (UK Biobank ID: 23447)
- SFA: Saturated fatty acid level (measured in mmol/l) (UK Biobank ID: 23448)
- TFA: Total fatty acid level (measured in mmol/l) (UK Biobank ID: 23442)
- PC/TFA Ratio: Ratio of phosphatidylcholine to total fatty acids calculated by PC/TFA.
- PUFA/TFA Ratio: Ratio of polyunsaturated fatty acids to total fatty acids. Units of measurement are percent.(UK Biobank ID: 23453)
- MUFA/TFA Ratio: Ratio of monounsaturated fatty acids to total fatty acids. Units of measurement are percent. (UK Biobank ID: 23454)
- SFA/TFA Ratio: Ratio of saturated fatty acids to total fatty acids. Units of measurement are percent. (UK Biobank ID: 23455)
- PUFA/MUFA Ratio: Ratio of polyunsaturated fatty acids to monounsaturated fatty acids. Units of measurement are ratio. (UK Biobank ID: 23458)

Charlson's comorbidity index (CCI) was calculated using the comorbidity package[82] in R using the ICD-10 disease codes. We included all individuals with available proteomics and metabolomics data, without applying any pre-selection criteria ($n = 30,278$). Subsequent analyses were conducted using the subset of samples containing the relevant measurements, with the specific sample sizes for each analysis provided in parentheses.

Group comparisons were done by the Wilcoxon rank-sum test and the correlations between lipids and health measures (Fig. 7h) were calculated using Spearman's correlation test. All the analysis of UK Biobank data is done in the R programming environment (v4.2) using the tidyverse package[83]. ggplot2 package was used for visualization[84]. As individual-level information cannot be shared due to data protection reasons, we provided LOESS predictions (using the default parameters of span = 0.75 and degree = 2 in the stats::loess() function in R) and density functions to describe population distributions, which should also be enough to replicate the manuscript figures with small differences.

## Statistics and reproducibility

Sample sizes were determined based on previous publications and field standards for each assay. Standard data exclusion criteria were applied for *C. elegans* survival assays (*e.g.*, gut ruptures and internal hatching were censored according to established protocols). Blinding was not performed, as most experiments were independently conducted by more than one researcher with consistent results. Randomization was achieved by selecting animals randomly from larger populations for each test. The results are expressed as mean ± SEM in most cases unless stated otherwise. Statistical analyses were performed in R, GraphPad Prism 8.0.0 (GraphPad Software Inc.) and Excel. Respective statistical tests are described in each figure legend and exact *p* values and *n* numbers are provided in the Source Data file for each experiment.

## Reporting summary

Further information on research design is available in the Nature Portfolio Reporting Summary linked to this article.

## Data availability

The mass spectrometry proteomics data have been deposited to the ProteomeXchange Consortium (Deutsch et al.) via the PRIDE (Perez-Riverol et al.) partner repository available at (http://www.ebi.ac.uk/pride), with the dataset identifier PXD024180. Alldata for the PEMT and UK Biobank analyses are available at (https://doi.org/10.5281/zenodo.18554287) and (https://github.com/donertas-group/mt-aging). The mass spectrometric lipidomics data generated in this study have been deposited in the Metabolomics Workbench database (an international repository for metabolomics data and metadata, metabolite standards, protocols, tutorials and training, and analysis tools[85]) under the accession code PR002921 (ST004627; https://doi.org/10.21228/M83K12). Source data are provided with this paper.

## Code availability

Code for the PEMT and UK Biobank analyses are available at (https://doi.org/10.5281/zenodo.18554287) and (https://github.com/donertas-group/mt-aging). Code for GTEx data preprocessing is available at (https://github.com/mdonertas/aging_in_GTEx_v8) and (https://doi.org/10.5281/zenodo.18554414).

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

## Acknowledgements

We thank Yvonne Woitzat and Lisa Adam for providing technical support for all experiments performed in the Ermolaeva laboratory. We thank the Proteomics Core Facility and especially Dr. Joanna Kirkpatrick, and SPARK Technology Transfer Core Facility at FLI for supporting this study. The FLI is a member of the Leibniz Association and is financially supported by the Federal Government of Germany and the State of Thuringia. TP was supported by the German Academic Exchange Services (Deutsche Akademische Austauschdienst, DAAD). ME and LE were supported by the EU-ESF Thüringer Aufbaubank funding (2019 FGR 0082). ME is funded by the ERC CoG LifeLongFit of the European Commission. ME is also funded by the Carl-Zeiss-Stiftung via IMPULS consortium and is a member of Cluster of Excellence Balance of the Microverse funded by the Deutsche Forschungsgemeinschaft (DFG). PA was supported by the DFG-funded Cluster of Excellence Balance of the Microverse. MD is funded by Carl-Zeiss-Stiftung (P2021-00-007). MD and UI were in part supported by DFG funded SFB1310. UI is supported by the Joachim Herz Foundation. Research activities of A.K. related to the subject of this article were funded in part by the Austrian Science Fund (FWF) (10.55776/P36299). LF is supported by Shenzhen University 2035 Program for Excellent Research (2024B004; LF) and the Program for Youzuzhikeyan of Shenzhen University (SZU2024YZZKY002; LF). C. elegans strains were obtained from the Caenorhabditis Genetics Center (CGC), which is funded by NIH. Parts of this research have been conducted using the UK Biobank Resource (application no.129228).

## Author contributions

M.E. conceptualized and designed the study; T.P., Y.L., P.C., I.V., P.A., L.E. and M.B. performed experiments; M.E., T.P., Y.L. and P.C. analyzed the data; M.D. designed and M.D. and U.I. performed human data analysis; A.K. designed and F.S. performed P.C. and P.E. extraction, measurement, and analysis; M.E., T.P., Y.L., P.C., U.I. and M.D. prepared figures; M.E., T.P., Y.L., P.C., U.I. and M.D. performed statistical analysis; L.F. co-supervised experiments contributed by Y.L. together with M.E.; M.E. wrote the manuscript and T.P., L.F., A.K. and M.D. reviewed the manuscript.

## Funding

## Competing interests

The authors declare no competing interest.
