## [Transparent Peer Review file · Nature Communications]

Aging-associated decline of phosphatidylcholine synthesis is a malleable trigger of natural mitochondrial aging.

Corresponding Author: Dr Maria Ermolaeva

Version 0:

Reviewer comments:

Reviewer #1

(Remarks to the Author)

Poliezhaieva et al. conducted a combined omics and genetic intervention approach on *C. elegans* to identify a key factor responsible for natural aging. The results described the involvement of S-adenosylmethionine (SAM) synthase in aging, particularly for mitochondria, and that methylation-dependent synthesis of phosphatidylcholine (PC) was involved downstream of the SAM process. RNAi modifications to worms indicated that alterations in SAM and PC metabolism during aging led to mitochondrial fragmentation. In addition, the contribution of PC synthesis to aging appeared widely conserved, as downregulated PEMT expression was observed in human aging. This study provides interesting insights, yet the lipid-related results may require a little more careful interpretation and additional evidence to increase the overall value of the work.

- It is still unclear how much methylation-dependent PC synthesis could contribute to mitochondrial fragmentation in aging. Measurements in the levels of PC, PE, or other phospholipids in knockdown worms or models with PC or choline supplementation may be required to better understand this point. The ratio of PC and PE could help interpret the involvement of the PC methylation pathway.

- How specific would PC be in causing mitochondrial fragmentation? While the authors discussed the roles of PA or PE in this regard, little information was available for PC. Related to the above point, it would be beneficial if the authors could provide quantitative information on lipid levels in their models to discuss this point better.

- How different would mitochondrial functionality be between young and old, and in SAM/PC-dependent fragmented mitochondria? While Figure 7 provided metabolic plasticity against metformin treatment, it might not be sufficient to discuss this thoroughly. Would it be possible to assess mitochondrial functionality in the absence of PMT or PEMT, such as by evaluating PMT knockdown worms or cell models? In addition, again, it would be better if the authors could see how much choline supplementation impacted PC levels in the cell models.

- Minor points:

I found some parts of the data presentation to be confusing. I was unsure what "Representative results of at least 3 independent experiments" meant, for example, in legends for Figures 4, 5, 6, and 7. Doesn't this refer to the average of all measurements?

The information on cell culture in the materials and methods section could be a bit more detailed. This would be helpful for readers to clearly understand what was done in this study, although the authors referenced their previous work.

Reviewer #2

(Remarks to the Author)

Poliezhaieva, et al. present a manuscript describing links between production of S-adenosylmethionine and mitochondrial function in *C. elegans*, with links to production of phosphatidylcholine. PC is an abundant phospholipid, it produced in mitochondria and has been shown to be important for mitochondrial function in other systems (import of complexes Schuler,

JBC 2017; reviewed in "The critical role of phosphatidylcholine and phosphatidylethanolamine metabolism in health and disease, BBA - Biomembranes, 2016). Links between SAM, PC, mitochondrial function and aging are of interest. However while the authors present much of their work as novel, much has been previously published, sometimes by multiple other groups. In addition, there are numerous errors in understanding the basic metabolic pathways in questions and results from GFP based reporters, dietary addition of metabolites and epistasis are over interpreted. Finally, in addition to failing to cite studies that contradict the novelty of their work, the authors also fail to attribute other studies correctly.

1. The authors state (line 13) "collectively our data revealed a novel previously unknown role of sams-1 in safeguarding mitochondrial safety". This assertion refers to the A) decline of sams-1 protein during aging, B) defects in mitochondria in sams-1 (RNAi) animals C) the induction of the multicopy hsp-6::GFP array in sams-1 animals. However, each of these observations has already been published, in some cases multiple times. The authors fail to cite these studies, which contradict the novelty of their study.

A) decline of SAMS-1 protein during aging

1. Proteomics by the Dillin/Yates labs showed that SAMS-1 protein decreased with age (Science, 2007).

<https://www.science.org/doi/full/10.1126/science.1139952#con7> (MS Figure 2A

2. Similar proteomics data is found in this study: (2015): <https://pubmed.ncbi.nlm.nih.gov/26390854/> and i

3. And this one (2015): <https://pubmed.ncbi.nlm.nih.gov/26390854/>. (MS figure 2A)

C) mitochondrial changes in sams-1 RNAi animals: Wei and Ruvkun published that sams-1 RNAi gave mitochondrial defects (PNAS, 2020). <https://www.pnas.org/doi/abs/10.1073/pnas.2008021117> (MS figure 3A). This is mentioned in the abstract of Wei and Ruvkun.

C) induction of hsp-6::GFP: The authors also essentially replicate a figure from this paper

(<https://www.pnas.org/action/downloadSupplement?doi=10.1073/pnas.1318262111&file=pnas.201318262SI.pdf>) showing

1) sams-1 RNAi increases hsp-6

2) sams-1 RNAi effects on hsp-6 can be rescued by choline and not ethanolamine (See MS figure S2 and Hou et al. Fig S6).

2. Metabolic pathways are incorrectly described, which limits confidence in the authors conclusions or understanding of the data.

A) Discussion pg 12 line 20: PMT-1 and PMT-2 do not convert phosphatidylethanolamine to phosphatidylcholine (Brendza, et al. 2007 J Biochem) and thus are not orthologs of PEMT in mammals (pg 13 In 6) which does make this conversion.

B) It is inaccurate to say that PC is a "precursor" to PA (pg 13 In 15). Phosphatidic acid is used in the synthesis of CDP DAG, which contributes to PC as well as other phospholipids as part of the Kennedy pathway. It can also be released when a specific phospholipase, PLD, cleaves PC. In addition, PE is only a "precursor" to PE (pg 20) in the methylation pathway, which is specific to only a few tissues in mammals. It is not a precursor in the Kennedy pathway, which produces the majority of PC.

C) Dietary rescue experiments are complex to interpret, as many metabolites have multiple fates and may be metabolized before the "rescued" step is reached. This is especially a concern in *C. elegans*, where bacterial diet is live and metabolically active.

a. SAM is labile and may break down into SAM or MTA which can inhibit SAM-utilizing enzymes, see Sun and Locasale, Cell Stress 2022. The authors treat *C. elegans* with SAM and measure HSP-6::GFP, however, there are multiple issues with the interpretation. SAM treatment has the same effect on body size as sams-1 RNAi, this suggest that SAM may have broken down and produce distinct effects.

i. SAM should be measured in control and SAM treated animals.

ii. SAM rescue of sams-1 phenotypes (lipid droplets, body size, reduced fertility, gene expression patterns) should be demonstrated. If SAM addition is having the same effects as sams-1 RNAi, as in the body size in Fig 4C, this would show that dietary addition of SAM has distinct metabolic and biological effects.

iii. Arguments about effects on chromatin from PC rescue in sams-1 vs pmt-1 RNAi are over interpreted. The hsp-6::GFP is a multicopy array and is only a GFP reporter. It may be subject to different chromatin effects. Also, RNAi or knockout of PMT enzymes has been shown to have feed back by increasing SAM and affecting histone methylation (Ye et al, Mol Cell 2017). Therefore this is a complex system which would require much more experimentation and validation before these conclusions could be made.

b. PC can not be taken up into cells and is first digested by phospholipases, therefore the PC rescues are likely to be essentially similar to directly adding choline.

c. In mammals, adding choline can produce SAM through the BHMT assays, therefore it is impossible to ascribe the effects in Figure 7E to PC.

D) Other issues:

a. GSEA is a distinct enrichment tool, weighting genes based on expression, thus WormCat is not a GSEA tool (pg 4, In 5).

b. Authors over interpret of lifespans effects with sams-1 RNAi and mitochondrial mutants. If SAMS-1 decreases in aging animals, loss of sams-1 extends lifespan extends life farther and sams-1 loss also effects mitochondrial morphology. Adding this defect to mitochondrial mutants may just make the worms sick, living shorter than wild type shows failure to thrive, and is not an ageing phenotype.

c. Vitellogenins are associated specifically with yolk droplets which are transported from the intestine to the germline in *C. elegans*. They are not general lipid droplets associated proteins, as implied by the authors. (pg 10). There is abundant literature of relationship of yolk secretion to aging.

d. Incorrectly attributed citations:

i. Pg 8, In 7: Reference 34 predates ref 29 for this observation.

ii. Pg 9, ln 12: reference 32 predates ref 29 for this observation.

Reviewer #3

(Remarks to the Author)

In this study, Poliezhaieva and colleagues employ proteomics to analyze the variations in protein expression between young and aged wild-type *C. elegans*, as well as between long-lived *clk-1(qm30)* and *isp-1(qm150)* mitochondrial mutants. They discover that S-adenosylmethionine synthase SAMS-1 is reduced in expression during normal aging. Knockdown of *sams-1* leads to changes in mitochondrial morphology and activates the mitochondrial unfolded protein response (mitoUPR). Furthermore, they reveal that the synthesis of phosphatidylcholine, which is dependent on S-adenosylmethionine (SAM), is crucial in modulating mitochondrial morphology, although it does not affect the mitochondrial unfolded protein response (mitoUPR). Overall, the manuscript presents some interesting observations that may be of interest to a broad readership. Nonetheless, there are several concerns that need to be resolved.

Major points:

- In Figure 2, it is shown that SAMS-1 protein expression level decreases in wild-type (WT) aged animals. This raises a question: why does further reducing *sams-1* expression extend lifespan (Fig. 2c)? The authors should address and discuss this point.
- The authors show that the lifespan of the long-lived mitochondrial mutants *clk-1(qm30)* and *isp-1(qm150)* is diminished with *sams-1* knockdown (KD), as illustrated in Figures 2d and e. The authors suggest these results confirm *sams-1*'s role in maintaining homeostasis and longevity under chronic mitochondrial impairments. However, *sams-1* KD also leads to significant developmental defects (Figure 3d), alters mitochondrial morphology, and activates mitoUPR (Figures 4b and c). It's plausible that the mitochondrial defects induced by *sams-1* KD exacerbate the existing mitochondrial deficiencies in *clk-1(qm30)* and *isp-1(qm150)* mutant worms. This could transform the mild mitochondrial defects in these mutants into more severe ones, thus shortening their lifespan. If this is true, it might imply that the effect observed is not directly related to *sams-1* expression levels in these mutants. Similarly, the knockdown of any other gene affecting mitochondrial function could potentially produce the same results.
- When the authors conducted a quantitative analysis of mitochondrial morphology, they used categories like 'very fragmented', 'fragmented', 'intermediate', and 'tubular'. However, it's unclear from the main text, figure legends, or methods section how these categories are defined. How do the authors distinguish between 'very fragmented' and 'fragmented'?
- SAM is metabolized by the phosphoethanolamine N-methyltransferases PMT-1 and PMT-2. The authors should present findings for *pmt-2* knockdown, similar to the results they have shown for *pmt-1*.
- The lifespan data for wild-type, *clk-1(qm30)*, and *isp-1(qm150)* under *pmt-1* knockdown conditions should be provided.
- Comparing Figures 4b and 6b, it is unclear why the supplementation of phosphocholine (pCho) results in a more effective rescue of mitochondrial morphology when *sams-1* is knocked down. Could the authors provide an explanation for this observation?
- When comparing Figures 6b and 7d, it's not evident why phosphocholine (pCho) supplementation is considerably less effective in rescuing mitochondrial morphology in aged animals. Could the authors clarify or offer an explanation for this observation? Furthermore, what impact does S-adenosylmethionine (SAM) supplementation have on aged animals as depicted in Figure 7d?
- The cell toxicity induced by Metformin is unrelated to aging. Therefore, the authors should refrain from claiming that they have investigated the same mechanism in human systems.

Minor points:

- The order of each panel in the figure needs to be numbered and arranged according to the sequence in which they are mentioned in the main text. For example, currently Figure 1E appears before the reference to Figure 1D, and there are similar instances later in the text.
- Scale bars seem to be absent in all the microscopy images.
- What is *mip-1* in Fig. 3c? This is not mentioned or explained in the text.
- In the main text, when referring to figure panels, uppercase letters are used, but the labels for each panel in the figures themselves are in lowercase letters.
- The labels 'C01', 'N01', 'I01' in Figure S1 are difficult to understand immediately. It is recommended that the authors use

gene names for labeling, or provide explanations for these abbreviations in the figure legends.

Reviewer #4

(Remarks to the Author)

In this manuscript, the authors explore an important question regarding how mitochondria naturally become dysfunctional with age, using *C. elegans*. Though an impressive longitudinal proteomic analysis of two well-characterised long-lived mutant *C. elegans* strains with perturbed mitochondrial function, the authors identify the phosphatidylcholine synthesis pathway as a protective pathway in the mutant *C. elegans* with mitochondrial dysfunction. The authors discover this pathway is decreased with age in WT *C. elegans*, and disruption of this pathway leads to a fragmented mitochondrial phenotype. Remarkably, dietary supplementation with choline and SAM was able to rescue the fragmented mitochondrial morphology. Supplementation with Choline was also found to be protective in a mammalian cell culture model of acute mitochondrial stress.

The comparative analysis of aged *C. elegans* vs. *C. elegans* with mitochondrial dysfunction that have developed protective adaptations is a very smart method of identifying cellular pathways that are lost with ageing. Overall, the data in this manuscript is high quality, but some additional experiments are required to robustly support the authors conclusions. In my opinion, the manuscript will be suitable for publication if the revisions suggested below are addressed.

1) Signs of UPR_{mt} activation were seen in the mutant *C. elegans* used for the proteomics analysis (Fig S1) as would be expected from the persistent mitochondrial stress in these strains. In contrast, minimal UPR_{mt} activation was seen in the *pmt-1* RNAi strain (Fig 5d), and as the authors described on pg8, the UPR_{mt} activation seen in the *sam-1* RNAi strain appears to be through a methylation dependent mechanism instead of through mitochondrial stress mediated activation. Collectively, this data is at odds with the mitochondrial fragmentation patterns observed in these strains (Fig 3b, Fig 5b) and would instead suggest that mitochondrial dysfunction is minimal in the absence of the PC synthesis pathway. Due to these conflicting observations, it is important for the authors to perform additional experiments specifically examining mitochondrial function to be able to robustly conclude that the age-dependent changes in PC metabolism are driving mitochondrial dysfunction. Such experiments could include Seahorse analysis of the respiratory capacity of mitochondria isolated from the different worm strains, BN-PAGE analysis and activity assays of respiratory complexes from isolated mitochondria, or JC-1 staining of mitochondria to measure membrane potential. These experiments should be performed in the worms with *sam-1* and *pmt-1* RNAi, and in the aged worms compared to young worms to demonstrate the effect of PC metabolism perturbation on mitochondrial function.

2) While the GTEX data and the cyto/mito-protective role of Choline supplementation during metformin treatment are very interesting, the GTEX data does not examine mitochondrial integrity which is the focus of this manuscript, and the metformin treatments do not replicate a phosphatidylcholine deficient scenario as is proposed to occur with ageing. As a result, the data included from the mammalian systems is providing correlative support of the main conclusions in the manuscript. I acknowledge that replicating ageing conditions in a mammalian system is extremely complex and difficult and I do not expect this to be done in this manuscript. However, the discussion regarding the mammalian data should be toned down to reflect the correlative nature of the data. Specifically:

Line 22 p2: the specific experiments performed relating to the human biological system should be specified (eg. editing that sentence to read 'In this study, we combined omics, genetics and functional analyses in *C. elegans*, with genetic analysis in humans and functional experiments in cell culture models...')

Lines 14 – 17 p14, sentence starting with 'Specifically, we hereby...': While changes to mitochondrial dynamics have been identified when the PC pathway is perturbed, it cannot be concluded from the data provided in this manuscript that the defect observed is specific to the mitochondrial fusion/fission processes. This line should be edited or removed as the data does not show that targeting mitochondrial fusion could be a therapeutic target during ageing.

3) The ontology analyses provided throughout the manuscript are very well done. To help viewers understand the significance of the ontology categories labelled in the main text figures (Fig 1b-h), data including the pValue and number of genes included in each category should be labelled next to the category name in the figures in addition to being included in the supplementary tables.

4) Representative images for the mitochondrial morphology analyses in Fig 4B, Fig 5b, Fig 6a and b, Fig 7d should be included in the main text or supplementary figures

5) Figures with N2 should also be labelled WT (i.e. WT (N2)), to make it easier for readers not as familiar with the *C. elegans* nomenclature to understand the figures

6) The rationale behind the inclusion of the control RNAi of *mip-1* should be referenced in the main text of the manuscript

7) Line 17 pg 6: A space is missing between 'sams-1' and 'KD'

Version 1:

Reviewer comments:

Reviewer #1

(Remarks to the Author)

The revised manuscript has addressed all of the points I raised previously. I have no further concerns.

Reviewer #2

(Remarks to the Author)

Poliezhhaieva et al. present a manuscript exploring connections between aging, levels enzymes contributing to the production of S-adenosylmethionine (SAM) to phosphatidylcholine (PC) and mitochondrial phenotypes in *C. elegans*. They also provide information from human samples in the UK Biobank. Low SAM is associated with models of lifespan extension across eukaryotes and SAM synthesis is one contributor to phosphatidylcholine levels. Their starting point in these studies were proteomics comparisons of aging *C. elegans* and two strains with mutations in mitochondrial proteins, which highlighted changes in the SAM synthase SAMS-1 as the animals age, along with changes in the enzymes which use SAM to produce PC in the methylation dependent pathway. They go on to investigate mitochondrial morphology and measure aspects of mitochondrial function in these animals and add back SAM or PC several of the assays, concluding that PC levels drop in normal aging and that levels can be rescued by diet.

While this subject area is of general interest and there are potential interesting observations, there are multiple issues with rigor and approach to published literature that reduce enthusiasm. First, key aspects of the data have been reported previously and either not cited or cited poorly. Second, there are issues with the analysis and interpretation of the data in multiple instances. Finally, the author's metabolic pathway descriptions are in some places overly simplified, reducing confidence in the interpretations. Alternative explanations for the decrease in PC in *C. elegans* during aging, such as the upregulation of PC consuming phospholipases that occurs with age are not considered. For example, this is noted in a similar proteomics study of aging *C. elegans* Copes, et al. 2015, that also finds decreases in SAMS-1 and SAM during *C. elegans* aging. Other pathways for producing PC (the Kennedy pathway) are the most dominant in mammals and the role that the rate limiting enzyme, *pcyt-1*, could have in this process is not considered.

1. The proteomics data is analyzed using WormCat, a *C. elegans* specific tool for pathway analysis. There are multiple issues with this analysis. First, the authors describe it as a "gene set enrichment analysis" (pg 75, line 16). Gene Set Enrichment (<https://www.gsea-msigdb.org/gsea/index.jsp>) is a distinct type of analysis that uses a ranked list of genes providing simultaneous evaluation of up and downregulated genes while WormCat uses enrichment statistics to evaluate all genes in a set without regard to expression level. Second, the authors do not supply information regarding the background list used for WormCat (whole genome or ORF). This is critical as proteomics data is sparse, and it is not appropriate to use whole transcriptome background lists for proteomics studies. Next, there are issues with the way the data is used to make the Venn Diagrams in Figure 1 f, g, and h. It is not clear which category level the authors are reporting, and it is not consistent to break down some categories (Metabolism) and not others. Even if the end conclusions are similar, the lack of reporting and rigor with this analysis reduces confidence in this study.

Some of these critiques were noted in the original review. While in the rebuttal, the authors state

Various tools for enrichment analysis are available for transcriptomic and proteomics studies, however, they should be used with precision and an understanding of the strengths and weaknesses in each approach. WormCat is still referred to as a "gene set enrichment analysis" (pg 75, line 16) in the present text.

2. One of the authors major rationales in the study is the change in SAMS-1 during aging *C. elegans* proteomes. This has already been observed in multiple studies. While a subset of these studies is mentioned in the discussion, this should be cited as a confirmatory result with the initial observation.

- Dong et al. show SAMS-1 decreases with age: <https://www.science.org/doi/full/10.1126/science.1139952#con7>.

- Copes et al. use quantitative proteomics to compare young adult and aging *C. elegans* and find that both SAMS-1 and SAM decrease.

-Walther et al. <https://www.sciencedirect.com/science/article/pii/S0092867415003207>

This was noted in the original review. Regarding the notion that "none of the above manuscripts specifically focused on this change", SAMS-1 is specifically investigated in the first two of these studies. Thus, the present study provides confirmatory and incremental advances in our knowledge of how the SAMS-1 protein changes during ageing.

- The decrease in SAMS-1 and PMT-1 is shown in Figure 1A of Dong et al.

- Copes et al. find the pathway that was "most changed with age was the biosynthesis of amino acids" including the S-adenosylmethionine pathway. This is mentioned in the abstract and in both the proteomics and metabolomics sections of the study and the decrease in SAM draw attention to the effects on metabolic function. While Walther et al. do not directly address the reduced levels of SAMS-1, Copes et al compare their data with Walther and note that SAMS-1 is decreased in both (see discussion). It is notable that the present study does not include such a comparison.

3. Both Reviewer's 2 and 3 noted questions about the combination of *sams-1*(RNAi) with effects on the lifespan of the mitochondrial mutants. The lifespan studies are very difficult to interpret without additional controls.

Although *clk-1* and *isp-1* are published to have longer lifespans than wild type animals, there are no wild type controls in Figure 2 c-e. It appears, especially for *isp-1*, that the mean lifespan is not that different from the wild type animals. If *sams-1*

has an effect on mitochondrial function, two mitochondrial problems might be enough to tip the balance from mito-hormesis to failure to thrive. The combination of effects fails to show the specificity claimed, because there are no additional controls or analysis of other mutants.

4. As mentioned in the previous review, fission of *sams-1* mitochondria have been previously published by Wei and Ruvkun. [https://www.pnas.org/doi/10.1073/pnas.2008021117?url_ver=Z39.88-](https://www.pnas.org/doi/10.1073/pnas.2008021117?url_ver=Z39.88-2003&rfr_id=ori%3Arid%3Aacrossref.org&rfr_dat=cr_pub++0pubmed#supplementary-materials)

The increase in mitochondrial fission is mentioned in the summary and is shown in the Figure S12 (Inactivation of *sams-1* increases mitochondrial fission), where effects on both *sams-1* RNAi on wild type animals with a muscle reporter of mitochondrial morphology and comparison to *drp-1* (*lof*) animals is shown.

Although Wei and Ruvkun's rationale stems from low SAM due to lysosomal changes, the context of *sams-1* knockdown in wild type animals causing mitochondrial fission is not sufficiently different to the present study to preclude discussion. It is not clear why the authors focus on figure 3A of Wei and Ruvkun, rather than the data directly with *sams-1* (Fig S12), as the reviewer states "MS Figure 3A" referring to the author's own manuscript, not the figure in the Wei and Ruvkun study.

5. In the previous review, there was a concern about the dietary addition of SAM. There are conflicting reports about the efficacy of adding SAM to media and the reviewer's request was to show that the phenotypes of the *sams-1* mutant or RNAi animals were rescued along with SAM levels. (See Sun and Locasale, Cell Stress 2022 for discussion of issues with SAM lability).

The authors were unsure why the request would be made to look at other phenotypes. (pg 9 of the rebuttal) If SAM were able to be taken up, then multiple phenotypes of the *sams-1* mutant or RNAi animals (lipid droplets, body size) would be rescued, not just the *hsp-6::GFP* expression. If SAM does not change body size, as the authors state in the rebuttal, then it appears even more likely that the SAM in the diet may be labile due to its instability and not taken up.

In Figure 5 B SAM is added and concluded to show changed in mitochondrial fragmentation. However, the magnitude of the change between *sams-1* with and without the SAM is quite low and measured by eye, rather than a quantitative assessment with a tool such as mitoMAPR (<https://www.life-science-alliance.org/content/7/11/e202402918>). While the authors report statistical significance, the fold effect of the added SAM is incremental at best.

6. Concerns remain about the over reliance on the *hsp-6::GFP* reporter as the only measure of the mitoUPR, especially when the authors note inconsistencies with their measures of mitochondrial phenotypes, potential rescue and this reporter.

SAM is used in multiple biosynthetic reactions and for regulatory modification of proteins and nucleic acids. The authors rebut that chromatin studies are not relevant to this study with PC, but the work by Ben Tu's lab (Ye Mol Cell 2017) cited above, show that changes in methylation dependent PC synthesis in yeast directly affect histone methylation and this was confirmed in *C. elegans* (Ding, et al. PLoS G, 2018). Thus, these processes are highly relevant to the present study and represent alternative hypotheses that should be considered.

Multiple copy arrays, such as the *hsp-6::GFP* reporter, are subject to regulation at the level of histone methylation in *C. elegans* and such reporters have been shown to be directly affected by SAM synthases (<https://www.sciencedirect.com/science/article/pii/S0092867412009439>). Thus, changes in activity of a single multi copy array especially when not accompanied by measurement of the endogenous gene are insufficient for the conclusions on the mitoUPR made in this study. While the author's note the alternative explanation, they do not use any distinct reporters or test endogenous genes, nor do they discuss the discrepancy with Hou et al, who show rescue of the reporter when choline is used to support PC levels.

The Walker et al study does not include the mitoUPR reporter, instead they use the *hsp-4::GFP* ER stress reporter.

7. Concerns remain about the author's explanation of metabolic/lipid synthesis pathways.

. – PC has multiple sources. Biosynthesis by the Kennedy pathway, which refers to the production of PC from choline, is the predominant pathway in mammalian cells (see

<https://pmc.ncbi.nlm.nih.gov/articles/PMC124149/#:~:text=Phosphatidylcholine is synthesized almost exclusively, pathway being cell type dependent>) and also a major contributor in *C. elegans* (see <https://academic.oup.com/genetics/article/207/2/413/5930737?login=false>).

-Dietary phospholipids are digested by phospholipases

(<https://www.sciencedirect.com/science/article/pii/S1878818120304357>), releasing lysophospholipids and then choline which are readily absorbed. In *C. elegans*, Copes et al. show that these phospholipases are highly expressed in aging animals and thus PC digestion would also be supported. Since choline has already been shown to rescue PC levels in *sams-1* (RNAi) animals (Ding, Cell Met 2015, <https://www.sciencedirect.com/science/article/pii/S1550413115003423>), it is likely this choline is providing the rescue, as has been previously shown, and not the PC.

- In the rebuttal, the author's mention two studies as "invalidating" the need to consider PC digestion. However, the papers do not invalidate this point. Bai et al show that PC in the yeast plasma membrane can be flipped by a translocase across one leaflet to the other. The PC is already in a membrane; the translocases is not acting on free PC nor is the PC added to an animal's digestive tract. Bai et al do add exogenous PC, however it is a non-physiological 7-nitrobenz-2-oxa-1,3-diazol-4-yl PC with a 6 carbon acyl chain to increase solubility for their structural and biochemical studies. The closest ortholog of this P4 ATPase in *C. elegans* is also predicted to reside inside the cell, not at the plasma membrane. The Sleight and Abanto study also provides very short chain NBD PC in liposomes to cultured cells at 2°C, which is non-physiological in comparison to PC added to *C. elegans* media.

-Thus, the simplest explanation of the PC rescue is that it is digested, with choline contributing to the rescue effect. Label

and tracing experiments would be necessary to demonstrate direct uptake, which are likely to be beyond the scope of the present study.

8. OCR assays with *sams-1* and *pmt-1* animals are difficult to interpret. The authors state that equal numbers of worms were added for control, *sams-1* and *pmt-1* animals. However, with the large difference in size, it would be expected that there are fewer mitochondria solely based on the fact that the animals are so much smaller. Thus, the reduction in OCR could be due to smaller animals having few mitochondria.

9. The authors have several figures showing rescue of PC by choline in the diet (Figure 6). This has already been shown by Ding, et al (see above comment). Lipidomic analysis is usually presented as normalized to total lipid, protein or relative to a labeled standard. It is not clear that the ratios add much in comparison to the unmodified data.

10. Concerns still remain about the author's statements regarding vitellogenins. In this study, the authors describe vitellogenins as "core components" of lipid droplets based on Espada, et al. one of their previous studies and in the rebuttal, the list two proteomics papers as proof.

Vitellogenins are yolk proteins, produced in the intestine and secreted into the germline (reviewed in <https://pmc.ncbi.nlm.nih.gov/articles/PMC6736625/>). To ignore this major role obscures their dominant function. There are abundant labeled markers and genetic studies showing accumulation of VIT proteins in intestinal endosomes (easily distinguishable organelles from lipid droplets), exocytosis and uptake into the germline by receptor mediated endocytosis (work by Barth Grant and others).

VITs are also among the most abundant proteins in *C. elegans*. If present as minor components of lipid droplet proteomes, they could be contamination if not independently verified. Zhang, et al., mentioned in the rebuttal, contains an immunoblot comparing VIT-2 in comparison to other lipid droplet proteins, however the levels are lower in the LDs than other compartments and they conclude that "Western blotting analysis showed that VIT-2 was only partially associated with LDs and the main signal was in total membrane fraction (Fig. 2C)". Thus, Zhang, et al does not support a major role for VIT-2 in LDs.

11. Concerns remain about the argument that choline supplementation is protective in mammalian cells. Mammals can convert choline to SAM via BHMT, therefore SAM could act through multiple mechanisms, and it is not a PC-specific treatment.

12. Concerns remain about the description of PEMT as a "functional analog" of *C. elegans* PMT-1/2. While the authors changed the language from ortholog (as there is no sequence similarity and differences in enzyme function), this statement is an oversimplification. The authors also corrected the conversion steps; in *C. elegans*, PMT-1 and PMT-2 convert phosphoethanolamine to phosphocholine and in mammals the conversion occurs after the acyl chains have been added from PE to PC.

However, these are not analogous reactions physiologically. In both cases, the methyl groups on PC are obtained from SAM rather than choline, but in mammals, PEMT acts tissue specifically and methylates PE so that PEMT-produce PC has different characteristics that PC produced by the Kennedy pathway. In *C. elegans*, since the methylation occurs before acylation, both are likely to contribute to bulk synthesis and as SAMS-1 has a wide distribution pattern, is likely to act across the animal. Due to the distinct natures of these enzymes, substrate kinetics and other regulatory mechanisms are not likely to be conserved. Although most PC synthesis does occur in the ER, as the authors note in their rebuttal, PEMT has been purified in mitochondrial fractions (<https://www.sciencedirect.com/science/article/pii/S0005273613003799>), providing a potential mitochondrial source of PC.

13. Although statements regarding PC being a precursor to phosphatidic acid were changed to say that PC could make PA through action of phospholipases there are still concerns regarding the author's incomplete descriptions these complex metabolic pathways.

While PC can be acted on in a signaling pathway to produce PA, PA can be made by 3 other pathways and it is also a precursor for all glycerolphospholipids. This makes it also a precursor to PC, providing the acyl chains after conversion to diacylglycerol by DAK kinases. Linking its levels solely to PLD ignores its major biosynthetic pools, and alternative sources of this lipid and over simplifies this complex pathway which affect confidence in the conclusions. (<https://www.sciencedirect.com/science/article/pii/S1388198119300460?via=ihub>).

14. The authors also look at PC levels in the UK biobank data. However, it is not clear what tissue this is in. Since only the liver used PEMT as contributor to PC synthesis, it is not clear this is relevant to SAM-dependent processes. Furthermore, if its plasma, the PC is likely to be a component of lipoprotein particles such as VLDL and HDL, therefore, there could be multiple indirect ways this metabolite could change.

Reviewer #3

(Remarks to the Author)

The authors have addressed my previous concerns in the revised manuscript.

Reviewer #4

(Remarks to the Author)

The authors have addressed all of my concerns, and I support publication of this manuscript in Nature Communications. I applaud the authors for the comprehensive revisions they have undertaken - all the additional experiments and text discussion now included have generated an excellent manuscript.

Version 2:

Reviewer comments:

Reviewer #2

(Remarks to the Author)

Poliezhaieva et al present a manuscript that begins with a proteomics analysis of mitochondrial mutants as they age, which show decreases in mitochondrial proteins. The authors then go on to select *sams-1* for further study and investigate changes in mitochondrial morphology as *sams-1* animals age. Next, they use dietary add back experiments to investigate how those might mitigate the physiological changes. Finally, the authors investigate metabolites in plasma from the UK BioBank samples. If the primary hypothesis is that SAM-dependent changes in PC synthesis are a driver of aging, the connection to the human data limited, as only a few tissues use SAM for a portion of its PC production in mammals. In their rebuttal, the authors point out that Kennedy pathway proteins are also present in their proteomics, but did not reach statistical significance. Thus, the relevance to human physiology is limited.

Several key aspects of this work (decline of SAMS-1 protein during aging and effects of *sams-1* RNAi on mitochondrial morphology) have been previously studied in multiple publications. Although the authors now cite these studies, the present work provides an incremental increase. For example, all of the *sams-1* data in figure 3 has been previously published. Additional issues with consistency in dietary update of metabolites, category analysis and functional characterization of protein function (vitellogenins as representative lipid droplet proteins) continue to reduce enthusiasm.

Figure 1:

In Figure 1, the authors show a proteomics comparison of wild type and mitochondrial mutants as they age. The authors then use WormCat, a *C. elegans* specific category enrichment program to identify functional groups within the changed proteins. They report changes in mitochondrial proteins, however the WormCat analysis used the whole genome annotation list, rather than the ORF set, which would be the appropriate background list for proteomics analysis (Higgins, Genetics 2022). WormCat contains three levels of analysis, and although it is not identified on the figure, in the legend or in the supplemental file, the authors only show most granular level, Category 3. They do provide protein lists in the supplement, and while they require reformatting to be run in WormCat, it is notable that checking the 499 genes with the correct background list for the data in figure 1C (converted from table S12) does not return any category 3 data.

The authors also appear to add up the category 3 data to generate the venn diagram in figure 1F to approximate category 2 level data. This does not allow correct enrichment statistics, as the other gene/category sets from category 2 are not present in the statistical analysis.

The most important aspects for rigorous presentation of proteomics data with any enrichment tool are use of an appropriate background list (Zhao and Rhee, Trends in Genetics 2023) and transparency of the data used in the search. Thus, background lists designed for RNA seq studies are not appropriate for proteomics studies. The gene lists provided in the supplemental data require reformatting (CEL_ numbers and gene names to gene ID or Wormbase ID and additional curation to update gene names or resolve in-line duplicates), thus utility is limited.

Figure 2:

The raw data for each repeat of lifespan data would be important to show that figure 2c, d, and e were conducted as part of the same experiment, along with the independent replicates.

Figure 3:

The data for *sams-1* in figure 3 has been reported in multiple studies (Wei and Ruvkun, PNAS 2020; Chen et al. Aging Cell 2024), thus the advance is incremental. As noted in the previous review, *hsp-6::GFP* is a multicopy reporter. The authors proteomics data show that as *sams-1* decreases, the endogenous *hsp-6* protein decreases (Table S4 Individual changes). Thus, evaluation of the endogenous RNA would also be a critical measure of mitoUPR induction, along with other genes induced in this process.

Figure 4:

Regarding evaluation of mitochondrial morphology (relevant to rescue experiments in figures 4-6):

The authors use visual scoring to evaluate mitochondrial morphology, rather than a computational approach. The studies they cite in the rebuttal are from 2014 and 2017. While this may be sufficient for large changes as in Figure 3A, the effect size from the addition of SAM is quite small. A more rigorous approach to these experiments (see Kim, et al. J Vis Exp 2025, Valera-Alberni, et al. Life Science Alliance 2024) would employ quantitation of mitochondrial networks with Fiji using a program such as MitoMapr (Zhang et al Elife 2019).

As mentioned in the previous critique, SAM is likely to break down and is unreliable addition to diet or cell culture (Sun and Locasale, Cell Stress 2021). Therefore, it is critical that the authors demonstrate SAM has direct effects in rescue experiments. In figure 4F and figure S2A, the effect of added SAM is quite low to moderate. Given that quantitation is by visual scoring in 4F and that no quantitation is provided in S2A, it appears that the ability of dietary SAM to rescue

mitochondrial morphology is limited at best. See recent suggestions for best practices on cell biology regarding the importance of effect size (Morgan, JCB, 2025).

There are inconsistencies compared to the treatment of SAM in the hsp-6::GFP experiments (Fig4C, FigS2b) where it appears have a more dramatic change on the GFP levels that the minor changes in the mitochondrial morphology. However, since HSP-6 levels decrease along with SAMS-1 in the aging studies and the endogenous hsp-6 mRNA data is not supplied for the day 1 animals, this data does not support the notion that dietary SAM rescues the mitochondrial phenotypes.

As noted in the previous critique, loss of sams-1 results in lipid droplet accumulation, small body size and reduction in brood size which are rescued by dietary choline. In figure S2, it does not appear the SAM alters the body size, thus limiting the rationale for rescue by dietary SAM. Finally, SAM may be routinely measured by LCMS in *C. elegans* most recently in Choi et al. Nat Comm 2023 and Tharpa et al. NPG Aging 2023.

Figure 5 -6

Figure 5-6 explore the links between the mitochondrial phenotypes and PC by dietary addition of PC. As noted in previous critiques, PC entering an animal though the diet is digested into choline and lysophosphatidic acid. Given this, the simplest explanation for the rescue in Fig5A, B is the addition of choline. As pointed out in the previous critique, P4 ATPases can "flip" PC from the outer leaflet to the inner leaflet of the plasma membrane, however, this would require that PC enter the plasma membrane directly. Members of P4 family of ATPases in *C. elegans*, as pointed out in the previous critique, are also predicted to be on intercellular membranes. Thus, given the likelihood that PC is digested and choline provides the rescue, even with the added discussion in the text, this data appears to provide only an incremental increase in knowledge without more rigorous tracing studies.

Seahorse assays measuring OCR are sensitive to the number animals (Haroon and Vermulst, Bio Protoc 2019). Thus, while normalization by number of animals is appropriate when they are the same size, the much smaller body size of sams-1 and pmt-2 animals means that fewer mitochondria are being added to each well. Rigorous experimental design would require normalizing to body size or protein concentration. See Gioran and Chondrogianni, Redox Biology 2925, Guidelines for the measurement of oxygen consumption rate in *Caenorhabditis elegans*).

Figure 6

There was also concern in the previous critique regarding the authors explanation of metabolic/lipid synthesis pathways regarding the source of PC in most mammalian tissues as the Kennedy pathway (with liver being the predominant exception and minor roles in adipose tissues), which limits the ability to make inferences between the SAM-PC methylation connection in human aging.

In their response, the authors state that some Kennedy pathway components are present in their proteomics, but they are not statistically significant, thus not a rigorous component of the data. Later in the rebuttal, they respond that in figure 6A, knockdown of sams-1 or pmt-1 reduce PC in the entire animal as evidence that whole animal PC levels can be affected by methylation-dependent PC production. In *C. elegans*, SAM synthases are widely expressed, thus this does not refute the relative importance of PC production pathways in mammals. Finally although the authors have acknowledged that adding choline can have multiple fates in culture cells, the lack of specificity limits interpretations from their cell culture study.

Although the authors now cite Ding et al, which shows PC is restored in sams-1 animals fed choline, since this has been previously established much of the data in figure 6 represents a previously published or incremental advance.

Other: Regarding authors assignment of vitellogenin as lipid droplet markers. As noted in the previous critique, vitellogenins in *C. elegans* are part of yolk particles, which are distinct from lipid droplets. Chen et al. (PMID: 37123874) use a highly rigorous approach, two-photon fluorescence to show lipid droplet and vit-2::GFP localization and Zhai et al. (PMID: 36199214) use immunogold to localize vit-2 and other vitellogenins to the electron dense yolk bodies, and show increases in aging animals. Thus, vitellogenins are quite relevant to post reproductive elements. Although VIT-2 may have come down as a contaminant or minor component biochemical preparations with lipid droplets, multiple imaging studies provide a clear distinction.

For assessment of lipid droplet components that have been rigorously used in the field, DHS-3::GFP is widely available.

Reviewer #5

(Remarks to the Author)

Nat Comm 466648_2 Review

Poliezhaeva et al. "Aging-associated decline..."

I was asked to review this manuscript and to specifically determine if comments raised by Reviewer 2 had been adequately addressed by the authors in their rebuttal and revised manuscript. The manuscript asks a fundamental question: what are the root causes of age-related mitochondrial dysfunction? I found the manuscript to be well-written, ambitious and interesting. It shows that expression of proteins (e.g. SAMS-1 and others in the 1CC pathway) important for the synthesis of phosphatidylcholine (PC) declines with age in *C. elegans* and that this contributes to the morphological/functional decline of

mitochondria. Importantly, the mitochondrial dysfunction observed in mutants with defects in PC synthesis or in aging wild-type worms could be suppressed by providing PC or choline as dietary supplement. Lipidomics analysis confirms the expected impact of mutations/dietary supplements on PC levels. The results are extended to human cells where interesting correlations were observed, including declining PC levels in old age and diabetics.

Most comments by Reviewer 2 are judicious and speak of a deep expertise in the subject. However, I also find that the author's rebuttal and revised manuscript address adequately (though sometimes imperfectly) the reviewer's comments and therefore view the manuscript positively.

Like the reviewer, I would expect smaller worms to have fewer mitochondria and reduced OCR. Nevertheless, I find the presented OCR data to be valuable. For example, there is no denying the potent effect of PC supplementation in Fig. 5e. Also like the reviewer, I find it difficult to interpret the vitellogenin data (Fig. S8A) and feel that neatly presenting them as core components of lipid droplets is an oversimplification and a little misleading. But I can live with that, though I would prefer that the authors mention that vitellogenins are main components of yolk granules and that they may accumulate in the pseudocoelomic cavity in aging/mutant worms.

Reviewer 2 raises many other important points, including concerns about the way WormCat was used and results reported, variable/low efficacy of SAM supplements, excessive reliance on the multi-copy hsp-6 GFP reporter, quantification of mitochondrial morphology, etc. However, I also find that the authors have adequately addressed these comments and improved the manuscript as a result.

The authors find that PC supplements can suppress mitochondrial dysfunction. As Reviewer 2 points out, it is possible that the mechanism for this depends specifically on restoring choline/SAM levels through digestion of PC. However, I find it more plausible that it is the restoration of adequate PC levels that is key and most consistent with all the data. Even if PC is digested into lysoPC, FFA and choline before uptake into the intestinal cells, these are then building blocks to generate new PC. The novelty of this paper is that it brings together a set of experimental observations, some of which confirms data previously published by others, supporting the hypothesis that reduced PC levels associated with aging contribute to declining mitochondria quality and that sustaining PC levels can prevent mitochondrial dysfunction in mutants or during aging. I am sufficiently convinced that the data supports these conclusions.

Version 3:

Reviewer comments:

Reviewer #2

(Remarks to the Author)

The authors have provided the base data for WormCat for some of the figures, but continue to use the tool in non-standard ways, such as summing category 3 data to produce "67%" (Fig1C and F). For example, the WB IDs in S12a refer to the unique proteins in Figure 1C, but not the overlap, which must be looked up. Adding the Category 3's together to reformulate a larger category (Metabolism) separates these categories from the background lists essential to the statistics used to calculate enrichment in the Fisher test.

Response to reviewers.

We would like to thank the reviewers for their insightful comments, which helped us to improve our manuscript and enhance the clarity of data presentation. Responses to the specific comments can be seen below. The revised parts of the manuscript text are marked with the grey highlight.

Reviewer #1 (Remarks to the Author):

Comment: Poliezhaieva et al. conducted a combined omics and genetic intervention approach on *C. elegans* to identify a key factor responsible for natural aging. The results described the involvement of S-adenosylmethionine (SAM) synthase in aging, particularly for mitochondria, and that methylation-dependent synthesis of phosphatidylcholine (PC) was involved downstream of the SAM process. RNAi modifications to worms indicated that alterations in SAM and PC metabolism during aging led to mitochondrial fragmentation. In addition, the contribution of PC synthesis to aging appeared widely conserved, as downregulated PEMT expression was observed in human aging. This study provides interesting insights, yet the lipid-related results may require a little more careful interpretation and additional evidence to increase the overall value of the work.

Response: We thank the reviewer for their positive assessment of our work.

Comment: It is still unclear how much methylation-dependent PC synthesis could contribute to mitochondrial fragmentation in aging. Measurements in the levels of PC, PE, or other phospholipids in knockdown worms or models with PC or choline supplementation may be required to better understand this point. The ratio of PC and PE could help interpret the involvement of the PC methylation pathway.

Response: We thank the reviewer for this suggestion. As suggested, we measured levels of PC and PE lipids in young control, *sams-1* knock-down (KD) and *pmt-1* KD nematodes with and without choline supplementation by mass spectrometry. We also measured PC and PE levels in post-reproductive (adulthood day 6) nematodes with and without choline supplementation. Choline was chosen over PC in most of our new supplementation experiments due to its higher solubility in water, allowing us to avoid organic solvents as vehicle. Of note, in previous work we already

demonstrated that PC and PE levels, and especially the LPC and LPE levels decline with age in WT *C. elegans* grown on the OP50 *E. coli* diet¹. Our new data demonstrate reduced PC/PE and LPC/LPE ratios in young *sams-1* and *pmt-1* knockdown animals (Figure 6a, b and Table S24), consistent with a pronounced inhibitory effect of these genetic manipulations on methylation-dependent phosphatidylcholine synthesis. The decline of LPC levels is particularly interesting due to the role of LPC in membrane curvature that contributes to membrane fusion². Notably, the relative loss of PC and LPC could be reversed by choline supplementation in both KD backgrounds (Figure 6a, b and Table S24). Choline supplementation elevated PC but not PE levels during aging (Figure S8b, c and Table S24) as well as PC/PE and LPC/LPE ratios in aged worms (Figures 6d and Table S24). This new lipidomics analysis (i) confirms that methylation-dependent phosphatidylcholine synthesis decreases with age, (ii) reveals that *sams-1* and *pmt-1* knockdowns mimic age-related declines in PC and LPC levels, and (iii) demonstrates that these deficits can be rescued by choline supplementation, both in aged animals and those with knockdowns. The new data is discussed in the revised manuscript on page 11, lines 3-22 and page 12, lines 10-21.

Comment: How specific would PC be in causing mitochondrial fragmentation? While the authors discussed the roles of PA or PE in this regard, little information was available for PC. Related to the above point, it would be beneficial if the authors could provide quantitative information on lipid levels in their models to discuss this point better.

Response: As described in the above response, PC but not PE levels were reduced both during aging and in *sams-1* and *pmt-1* KD animals, as well as the levels of LPC – a derivative of PC implicated in membrane curvature² (Figures 6a, b and d). At the same time, choline supplementation predominantly restored PC and LPC levels—but not PE and LPE levels (Figures 6a, b, and d)—and also improved age-related mitochondrial morphology (Figures 6e and S10a), suggesting a link between mitochondrial integrity and PC/LPC levels. Previous reports linked the relative abundance of PC and LPC to membrane curvature and flexibility, which are required for membrane fusion³. We thus revised our model to propose that PC and LPC may contribute directly to mitochondrial network integrity by regulating the flexibility/fluidity and fusion of mitochondrial membranes, and we have re-written relevant parts of the text to improve clarity (page 19, line 23 – page 20, line13). In this context, it is noteworthy that our new analysis of UK Biobank human

data shows that metabolic health and resilience in aging—both linked to mitochondrial efficacy—positively correlate with the abundance of lipid species that promote membrane flexibility (such as PCs and PUFAs), whereas lipid species associated with membrane rigidity (MUFAs and SFAs) are negatively correlated (Figures 7d–h and S15a–c). We now discuss these data and hypotheses in the revised manuscript (page 16, line 5 – page 18, line 5).

Comment: How different would mitochondrial functionality be between young and old, and in SAM/PC-dependent fragmented mitochondria? While Figure 7 provided metabolic plasticity against metformin treatment, it might not be sufficient to discuss this thoroughly. Would it be possible to assess mitochondrial functionality in the absence of PMT or PEMT, such as by evaluating PMT knockdown worms or cell models? In addition, again, it would be better if the authors could see how much choline supplementation impacted PC levels in the cell models.

Response: We thank the referee for highlighting the need of a more direct assessment of the mitochondrial function in the context of differential SAM/PC availability. To this end, we now performed whole animal Seahorse experiments in young control, *sams-1* KD and *pmt-1* KD worms with and without PC supplementation (tested in the *pmt-1* KD background), as well as in post-reproductive nematodes with and without ectopic choline. These assays enabled a direct comparison of mitochondrial function with previously acquired morphological data (Figures 4b, S2a, 5a–b, S5, 6e, S10a) as well as newly generated lipidomics profiles (Figures 6a, b, d and S8b–c). The new OCR measurements clearly show that (i) mitochondrial function is reduced in both knockdown backgrounds compared to controls (Figure 5d), and (ii) it also declines with age, as indicated by lower baseline values in Figure 6f relative to those in Figures 5d and 5e. These functional declines coincide with reduced PC levels and an increased proportion of fragmented mitochondria in the affected animals. Notably, PC supplementation restored OCR in *pmt-1*-deficient worms (Figure 5e), consistent with PC's role in promoting mitochondrial fusion. Similarly, choline supplementation improved OCR levels in aged worms (Figure 6f). Unlike the consistent effects seen with choline and PC, SAM supplementation produced highly variable results across assays (Figure S11), likely owing to its known low stability, as elaborated on page 15, lines 1-15.

•Minor points:

Comment: I found some parts of the data presentation to be confusing. I was unsure what “Representative results of at least 3 independent experiments” meant, for example, in legends for Figures 4, 5, 6, and 7. Doesn’t this refer to the average of all measurements?

Response: This indicates that a minimum of three independent experiments were conducted, yielding comparable results, with one representative experiment shown in the figure. In response to this comment, we have clarified this information in all figure legends.

Comment: The information on cell culture in the materials and methods section could be a bit more detailed. This would be helpful for readers to clearly understand what was done in this study, although the authors referenced their previous work.

Response: We have expanded the section as suggested (page 7, line 19 – page 8, line 20 of the methods file).

Reviewer #2 (Remarks to the Author):

Comment: Poliezhaieva, et al. present a manuscript describing links between production of S-adenosylmethionine and mitochondrial function in *C. elegans*, with links to production of phosphatidylcholine. PC is an abundant phospholipid, it produced in mitochondria and has been shown to be important for mitochondrial function in other systems (import of complexes Schuler, JBC 2017; reviewed in” The critical role of phosphatidylcholine and phosphatidylethanolamine metabolism in health and disease, BBA - Biomembranes, 2016). Links between SAM, PC, mitochondrial function and aging are of interest. However, while the authors present much of their work as novel, much has been previously published, sometimes by multiple other groups. In addition, there are numerous errors in understanding the basic metabolic pathways in questions and results from GFP based reporters, dietary addition of metabolites and epistasis are over interpreted. Finally, in addition to failing to cite studies that contradict the novelty of their work, the authors also fail to attribute other studies correctly.

Response: We appreciate the reviewer’s acknowledgment of the originality of our key findings, as reflected in the statement that “Links between SAM, PC, mitochondrial function, and aging are of interest.” However, the assertion that our observations have been “previously published, sometimes by multiple other groups,” as well as the implication that we may have misunderstood or misinterpreted our own data, is incorrect. These points are thoroughly addressed and refuted in our detailed response below. Additionally, the reviewer’s statement that PC is synthesized in mitochondria is inaccurate; it is well established that PC is primarily synthesized in the endoplasmic reticulum and subsequently transported to mitochondria.

Comment: The authors state (line 13) “collectively our data revealed a novel previously unknown role of *sams-1* in safeguarding mitochondrial safety”. This assertion refers to the A) decline of *sams-1* protein during aging, B) defects in mitochondria in *sams-1* (RNAi) animals C) the induction of the multipcopy *hsp-6::GFP* array in *sams-1* animals. However, each of these observations has already been published, in some cases multiple times. The authors fail to cite these studies, which contradict the novelty of their study.

Response: The sentence referenced by the reviewer (page 8, lines 1–2 of the revised manuscript) differs from the version in the earlier submission, which read: “Collectively, our data revealed a novel, previously unknown role of *sams-1* in safeguarding mitochondrial INTEGRITY.” This difference is important because the correct sentence specifically addresses the role of *sams-1* in maintaining mitochondrial network integrity, as demonstrated using the dedicated mitochondrial morphology reporter *myo-3p::gfpmit*. This reporter is distinct from the multi copy *hsp-6p::GFP* reporter mentioned by the reviewer, which was not used in this context. The relevant findings are presented in Figures 3a and 3b. Points A, B, and C raised by the reviewer are not related to this figure or its associated conclusions. To avoid any potential misunderstanding, we have revised the wording in the manuscript accordingly (page 8, lines 1–2).

Comments suggesting that aspects of our findings have been previously reported:

A) decline of SAMS-1 protein during aging

1. Proteomics by the Dillin/Yates labs showed that SAMS-1 protein decreased with age (Science, 2007). <https://www.science.org/doi/full/10.1126/science.1139952#con7> (MS Figure 2A)

2. Similar proteomics data is found in this study: (2015):

<https://pubmed.ncbi.nlm.nih.gov/26390854/>

3. And this one (2015): <https://pubmed.ncbi.nlm.nih.gov/26390854/>. (MS figure 2A)

Response: If SAMS-1 protein expression genuinely declines with age, this expression change can retrospectively be found in many published proteomics datasets of *C. elegans* aging, which is what the referee reports here. However, none of the above manuscripts specifically focused on this change, and especially none of them investigated this change in connection with mitochondrial aging. Moreover, none of the above studies did their analysis in such a way as to discover that SAMS-1, PMT-1 and PMT-2 proteins are the 3 strongest downregulated proteins during advanced aging in the nematode, drawing specific attention to methylation-dependent phosphatidylcholine biosynthesis and its role in mitochondrial and metabolic aging. As such, the above studies do not interfere with the novelty of our work, and on the contrary, provide an independent validation of some of our proteomics findings.

B) mitochondrial changes in *sams-1* RNAi animals: Wei and Ruvkun published that *sams-1* RNAi gave mitochondrial defects (PNAS, 2020).

<https://www.pnas.org/doi/abs/10.1073/pnas.2008021117> (MS figure 3A). This is mentioned in the abstract of Wei and Ruvkun.

Response: The reviewer suggests that “mitochondrial defects” resulting from *sams-1* RNAi are demonstrated in Figure 3A of Wei and Ruvkun, PNAS, 2020. However, upon reviewing this figure, we found that it does not include any data on *sams-1* RNAi. Instead, the figure shows that RNAi against *hlh-30*, *lmp-1*, and *lmp-2*, which are involved in lysosomal function, can reverse the abnormal mitochondrial fusion caused by *drp-1* RNAi exposure. Overall, this study is not related to aging and focuses on the interplay between the lysosome and vitamin B12 metabolism in regulating mitochondrial dynamics, which is distinct from the focus of our study. The abstract does mention that inactivation of *sams-1* “also suppresses the *drp-1* fission defect”, and some *sams-1* data are presented in the supplementary figures. However, this effect relates to *sams-1*’s role in

vitamin B12 utilization through the folate-methionine cycle and is unrelated to our findings. While there is no obvious direct overlap between that study and ours, we now included the citation of this work because it connects *sams-1* and mitochondrial function, albeit in a different context.

C) induction of *hsp-6::GFP*: The authors also essentially replicate a figure from this paper (<https://www.pnas.org/action/downloadSupplement?doi=10.1073/pnas.1318262111&file=pnas.201318262SI.pdf>) showing

1) *sams-1* RNAi increases hsp-6

2) *sams-1* RNAi effects on hsp-6 can be rescued by choline and not ethanolamine (See MS figure S2 and Hou et al. Fig S6).

Response: Contrary to the impression of referee #2, our manuscript does not present the upregulation of *hsp-6p::GFP* reporter expression by *sams-1* RNAi as a novel or original discovery. Furthermore, the key conclusions of our study do not rely on this observation being novel. The *hsp-6p::GFP* reporter is used throughout the study as a supplementary assay alongside the mitochondrial morphology tests with the *myo-3p::gfpmit* strain, which serve as our primary tool for discovery in most cases. Using both reporters in parallel enables us to distinguish between the specific mitochondrial effects of the interventions and the different gene knockdowns. An example of this can be seen in figures 5b and S4 where we used both reporters to show that phosphatidylcholine (PC) supplementation rescues mitochondrial morphology defects of *sams-1* KD animals (Fig 5b) but not their high levels of UPR^{mt} activity (Fig S4a, b). The upregulation of *hsp-6p::GFP* expression due to *sams-1* deficiency does not need to be novel in itself to support our original discovery of the specific role of phosphatidylcholine (PC) in maintaining mitochondrial network integrity, without impacting proteostasis. In response to this comment, we made it explicitly clear in the revised manuscript that the induction of UPR^{mt} by *sams-1* inactivation was already described prior to our study (page 7, line 23).

Comment: Metabolic pathways are incorrectly described, which limits confidence in the authors conclusions or understanding of the data.

Response: We respectfully disagree with this assessment. This statement is inaccurate, as shown in the following point-by-point response.

A) Discussion pg 12 line 20: PMT-1 and PMT-2 do not convert phosphatidylethanolamine to phosphatidylcholine (Brendza, et al. 2007 J Biochem) and thus are not orthologs of PEMT in mammals (pg 13 ln 6) which does make this conversion.

Response: In fact, the functional analogy between the PMT and PEMT pathways in contributing to phosphatidylcholine (PC) synthesis through SAM-dependent methylation reactions is well-established in the field (see Ehmke M et al., Genes Nutr, 2014, first paragraph of the Discussion, or the scheme in Figure 2B of Walker AK et al., Cell, 2011). To avoid any misunderstanding, we have revised both Figure S3a and its legend, as well as rephrased the original sentence in question to clarify that we intended to convey functional analogy (page 15, lines 19-20).

B) It is inaccurate to say that PC is a “precursor” to PA (pg 13 ln 15). Phosphatidic acid is used in the synthesis of CDP DAG, which contributes to PC as well as other phospholipids as part of the Kennedy pathway. It can also be released when a specific phospholipase, PLD, cleaves PC. In addition, PE is only a “precursor” to PE (pg 20) in the methylation pathway, which is specific to only a few tissues in mammals. It is not a precursor in the Kennedy pathway, which produces the majority of PC.

Response: Quoting from Bills and Knowles, Biomolecules, 2022: “Phosphatidic acid (PA) is a signaling lipid that is enzymatically produced from phosphatidylcholine (PC), lysophosphatidic acid, or diacylglycerol”, which confirms that PC is a direct precursor of PA in one of its biosynthetic pathways. When stating that “PE is only a precursor to PE (pg 20) in the methylation pathway,” the reviewer likely refers to the fact that PE serves as a precursor to PC only in the methylation-based biosynthesis route, and not in the Kennedy pathway. This is correct, as explicitly highlighted in the revised version of Figure S3a, and we have not suggested otherwise. While the Kennedy pathway, involving choline kinase, is generally considered more prevalent across tissues, our new lipidomics analysis clearly demonstrates that disrupting the methylation-dependent biosynthetic route alone is enough to reduce organismal levels of PC in *sams-1* or *pmt-1* knockdown animals (Figure 6a, b) and during aging (Figure S8b, c). This supports our overall findings and mechanistic model.

Comment: Dietary rescue experiments are complex to interpret, as many metabolites have multiple fates and may be metabolized before the “rescued” step is reached. This is especially a concern in *C. elegans*, where bacterial diet is live and metabolically active.

a. SAM is labile and may break down into SAM or MTA which can inhibit SAM-utilizing enzymes, see Sun and Locasale, Cell Stress 2022. The authors treat *C. elegans* with SAM and measure HSP-6::*GFP*, however, there are multiple issues with the interpretation. SAM treatment has the same effect on body size as *sams-1* RNAi, this suggest that SAM may have broken down and produce distinct effects.

i. SAM should be measured in control and SAM treated animals.

ii. SAM rescue of *sams-1* phenotypes (lipid droplets, body size, reduced fertility, gene expression patterns) should be demonstrated. If SAM addition is having the same effects as *sams-1* RNAi, as in the body size in Fig 4C, this would show that dietary addition of SAM has distinct metabolic and biological effects.

Response: We are unsure why the referee suggests that SAM treatment has the same effect on body size as *sams-1* RNAi, because the manuscript does not present data supporting this. The figure referenced by the reviewer (currently Figure S2b) actually assesses *hsp-6>::GFP* expression under various conditions, rather than body size. However, during the revision, we tested the effect of SAM on the body size of WT animals and found no impact (Figure S11d). In contrast, *sams-1* RNAi reduced body size, as previously shown (Figure S11d), which addresses and resolves the concerns raised above. Given that bacterial metabolism of the supplements may be a concern in some cases, we also tested whether the effect of SAM on mitochondrial morphology (the primary focus of this study) depends on the presence of live bacteria. Our results showed that it does not (Figure S6b), further resolving the referee's concerns.

Comment: Arguments about effects on chromatin from PC rescue in *sams-1* vs *pmt-1* RNAi are over interpreted. The *hsp-6>::GFP* is a multicopy array and is only a GFP reporter. It may be subject to different chromatin effects. Also, RNAi or knockout of PMT enzymes has been shown to have feed back by increasing SAM and affecting histone methylation (Ye et al, Mol Cell 2017).

Therefore this is a complex system which would require much more experimentation and validation before these conclusions could be made.

Response: In our manuscript, which focuses on the role of methylation-dependent PC synthesis in mitochondrial network integrity during aging, we did not investigate the chromatin effects of any of the KDs and/or supplements, as this is not directly relevant to our study and falls outside its scope. We are unsure what is specifically meant by the above comment, as it does not reference a particular figure or section of the text. However, in one instance, we do mention that previous studies (Merkwirth C et al., Cell, 2016, and Walker AK et al., Cell, 2011) suggest a plausible link between SAM levels and mitochondrial UPR activity through changes in histone methylation. (page 9, lines 8-10). This is simply a reference to previous publications provided for context, and this topic is not explored further due to its lack of relevance to our study's key objectives.

Comment: PC can not be taken up into cells and is first digested by phospholipases, therefore the PC rescues are likely to be essentially similar to directly adding choline.

c. In mammals, adding choline can produce SAM through the BHMT assays, therefore it is impossible to ascribe the effects in Figure 7E to PC.

Response: Contrary to the above statement, PC can be transported across the plasma membrane through ATP-dependent flipping without being digested by phospholipases (Sleight and Abanto, Journal of Cell Science, 1989; Bai L et al, eLife, 2020), invalidating the foundation of the above concerns.

Comment: GSEA is a distinct enrichment tool, weighting genes based on expression, thus WormCat is not a GSEA tool (pg 4, ln 5).

Response: WormCat is a pathway analysis tool, which is the type of analysis we intended to conduct. The definition of the analysis has been amended for clarity (page 4, lines 17-18).

Comment: Authors over interpret of lifespans effects with *sams-1* RNAi and mitochondrial mutants. If SAMS-1 decreases in aging animals, loss of *sams-1* extends lifespan extends life farther and *sams-1* loss also effects mitochondrial morphology. Adding this defect to mitochondrial mutants may just make the worms sick, living shorter than wild type shows failure to thrive, and is not an ageing phenotype.

Response: We interpreted the fact that *sams-1* RNAi shortens the lifespan of mitochondrial mutants, in contrast to its effects in wild-type animals, as an indication that *sams-1* inactivation negatively impacts mitochondria, thereby exacerbating the baseline impairment seen in the mutants. Based on this observation, we examined the mitochondria in KD animals and found morphological defects, supporting a new function of *sams-1* in preserving mitochondrial integrity. We view this as a logical and evidence-driven conclusion, not an overinterpretation. Furthermore, we now provide supporting lifespan data for two additional genes in the methylation-dependent phosphatidylcholine (PC) synthesis pathway - *pmt-1* and *pmt-2* (Figure 4g), which corroborate the findings observed with *sams-1* RNAi. To prevent any potential misunderstanding, we also revisited and revised the relevant text (page 6, lines 20-22; page 7, lines 3-8).

Comment: Vitellogenins are associated specifically with yolk droplets which are transported from the intestine to the germline in *C. elegans*. They are not general lipid droplets associated proteins, as implied by the authors. (pg 10). There is abundant literature of relationship of yolk secretion to aging.

Response: Contrary to the above statement, vitellogenins were identified as core components of *C. elegans* lipid droplets in two independent mass spectrometry studies (Zhang et al, Moll Cell Proteom, 2012; Vrablik TL et al, Biochim Biophys Acta, 2015).

Comment: Incorrectly attributed citations:

- i. Pg 8, ln 7: Reference 34 predates ref 29 for this observation.
- ii. Pg 9, ln 12: reference 32 predates ref 29 for this observation.

Response: In their comment, the reviewer is pointing out that the references are not presented in an ideal chronological order. The order of references was amended as suggested during the revision of the text.

Reviewer #3 (Remarks to the Author):

Comment: In this study, Poliezhaieva and colleagues employ proteomics to analyze the variations in protein expression between young and aged wild-type *C. elegans*, as well as between long-lived *clk-1(qm30)* and *isp-1(qm150)* mitochondrial mutants. They discover that S-adenosylmethionine synthase SAMS-1 is reduced in expression during normal aging. Knockdown of *sams-1* leads to changes in mitochondrial morphology and activates the mitochondrial unfolded protein response (mitoUPR). Furthermore, they reveal that the synthesis of phosphatidylcholine, which is dependent on S-adenosylmethionine (SAM), is crucial in modulating mitochondrial morphology, although it does not affect the mitochondrial unfolded protein response (mitoUPR). Overall, the manuscript presents some interesting observations that may be of interest to a broad readership. Nonetheless, there are several concerns that need to be resolved.

Response: We thank the referee for their positive assessment of our work and for their insightful comments.

Major points:

Comment: In Figure 2, it is shown that SAMS-1 protein expression level decreases in wild-type (WT) aged animals. This raises a question: why does further reducing *sams-1* expression extend lifespan (Fig. 2c)? The authors should address and discuss this point.

Response: The life-span prolonging effect of the life-long *sams-1* gene inactivation has been demonstrated by other studies^{4,5} and is thus reproducible. In our view, the difference is in the timing of *sams-1* inactivation, e.g. in Figure 2c and in previous publications it is initiated in early life, leading to a pro-longevity stress resilience response. At the same time, the natural decrease in SAMS-1 levels reaches significance only in late life, when animals can no longer mount effective adaptive stress responses. We have previously observed the same dynamics by using metformin as

a mitochondrial and metabolic stressor – the same concentration of the drug was positive for the young and negative for the old *C. elegans*¹. We have added a discussion of this point to the manuscript (page 7, lines 3-8).

Comment: The authors show that the lifespan of the long-lived mitochondrial mutants *clk-1(qm30)* and *isp-1(qm150)* is diminished with *sams-1* knockdown (KD), as illustrated in Figures 2d and e. The authors suggest these results confirm *sams-1*'s role in maintaining homeostasis and longevity under chronic mitochondrial impairments. However, *sams-1* KD also leads to significant developmental defects (Figure 3d), alters mitochondrial morphology, and activates mitoUPR (Figures 4b and c). It's plausible that the mitochondrial defects induced by *sams-1* KD exacerbate the existing mitochondrial deficiencies in *clk-1(qm30)* and *isp-1(qm150)* mutant worms. This could transform the mild mitochondrial defects in these mutants into more severe ones, thus shortening their lifespan. If this is true, it might imply that the effect observed is not directly related to *sams-1* expression levels in these mutants. Similarly, the knockdown of any other gene affecting mitochondrial function could potentially produce the same results.

Response: Our reasoning followed the same logic as suggested by the reviewer: we observed that *sams-1* knockdown shortens the lifespan of mitochondrial mutants but not wild-type animals. This suggested that *sams-1* knockdown exacerbates existing mitochondrial dysfunction, implying that SAMS-1 may play a role in maintaining mitochondrial homeostasis. We then tested this hypothesis by directly measuring mitochondrial morphology and stress levels in *sams-1* knockdown worms (Figure 3a–d), which revealed a previously unrecognized role for SAMS-1 in promoting mitochondrial fusion. In response to this comment, we rephrased the sub-title on page 6, lines 5-6 as well as text on page 7, lines 3-8 to improve clarity. We agree that knockdown of any gene supporting mitochondrial function would most likely reduce the lifespan of *clk-1(qm30)* and *isp-1(qm150)* worms. However, this does not diminish our specific discovery of a novel mitochondrial role for *sams-1*.

Comment: When the authors conducted a quantitative analysis of mitochondrial morphology, they used categories like 'very fragmented', 'fragmented', 'intermediate', and 'tubular'. However, it's

unclear from the main text, figure legends, or methods section how these categories are defined. How do the authors distinguish between 'very fragmented' and 'fragmented'?

Response: The precise quantification procedure is described in detail in our cited previous paper¹. In response to this comment, we amended the methods section with a more detailed description of the procedure (page 3, line 25 – page 4, line 2 of the methods file).

Comment: SAM is metabolized by the phosphoethanolamine N-methyltransferases PMT-1 and PMT-2. The authors should present findings for *pmt-2* knockdown, similar to the results they have shown for *pmt-1*.

Response: In response to this comment, we now demonstrate that *pmt-2* knockdown (i) increases the lifespan of WT animals but impedes the longevity of *clk-1(qm30)* and *isp-1(qm150)* mutants (Figure 4g), (ii) causes mitochondrial fragmentation in young wild-type animals (Figure S7a, b), (iii) impairs mitochondrial oxygen consumption rate (Figure S7d), and (iv) reduces body size (Figure S7c), mirroring the effects seen with *pmt-1* and *sams-1* knockdowns in all cases.

Comment: The lifespan data for wild-type, *clk-1(qm30)*, and *isp-1(qm150)* under *pmt-1* knockdown conditions should be provided.

Response: The data is now provided for both *pmt-1* and *pmt-2* KDs (Figure 4g), showing effects comparable to *sams-1* KD, in line with our mechanistic model.

Comment: Comparing Figures 4b and 6b, it is unclear why the supplementation of phosphocholine (PC) results in a more effective rescue of mitochondrial morphology when *sams-1* is knocked down. Could the authors provide an explanation for this observation?

Response: Our combined results from lipidomics, Seahorse assays, and mitochondrial morphology analyses consistently show that choline and phosphatidylcholine (PC) provide greater rescue effects than SAM across multiple genotypes and assays (Figures 5a, b, and e; 6a and b; S5a, b; and S11a–e). We further demonstrate that the variable effects of SAM are not due to its metabolism by

the bacterial diet of the nematodes, as similar outcomes were observed when SAM was combined with both live and heat-killed bacteria (Figure S6b). Because SAM is known to be highly unstable⁶, we attribute its variable efficacy to the chemical instability and lowered bioavailability, discussing this point in detail on (page 15, lines 1-15).

Comment: When comparing Figures 6b and 7d, it's not evident why phosphocholine (PC) supplementation is considerably less effective in rescuing mitochondrial morphology in aged animals. Could the authors clarify or offer an explanation for this observation? Furthermore, what impact does S-adenosylmethionine (SAM) supplementation have on aged animals as depicted in Figure 7d?

Response: While mitochondrial defects in young *sams-1* and *pmt-1* knockdown animals are primarily due to impaired phosphatidylcholine (PC) synthesis, the mitochondrial alterations that occur during normal aging are likely more complex and involve disruptions in multiple molecular pathways. This likely explains the comparatively smaller effect of choline supplementation in aging animals. We have incorporated a discussion of this important aspect into the revised text (page 13, lines 7-11). Similar to choline provision, SAM supplementation improved mitochondrial network integrity in old animals as shown in the new Figure S11e.

Comment: The cell toxicity induced by Metformin is unrelated to aging. Therefore, the authors should refrain from claiming that they have investigated the same mechanism in human systems.

Response: We agree with the reviewer that this experiment is not directly related to aging. It however demonstrates the ability of choline supplementation to support mitochondrial function under stress in human cells. We revised the relevant text to communicate the message more precisely in this context (page 14, lines 11-14).

Minor points:

Comment: The order of each panel in the figure needs to be numbered and arranged according to the sequence in which they are mentioned in the main text. For example, currently Figure 1E appears before the reference to Figure 1D, and there are similar instances later in the text.

Response: The panel labelling in Figure 1 was amended to eliminate this discrepancy.

Comment: Scale bars seem to be absent in all the microscopy images.

Response: All images and all respective figure legends were amended accordingly.

Comment: What is *mip-1* in Fig. 3c? This is not mentioned or explained in the text.

Response: We amended the text with a brief explanation and reference (page 7, lines 20-22).

Comment: In the main text, when referring to figure panels, uppercase letters are used, but the labels for each panel in the figures themselves are in lowercase letters.

Response: This has been unified throughout the manuscript.

Comment: The labels 'C01', 'N01', 'I01' in Figure S1 are difficult to understand immediately. It is recommended that the authors use gene names for labelling, or provide explanations for these abbreviations in the figure legends.

Response: We have updated the labelling and revised the corresponding figure legends accordingly.

Reviewer #4 (Remarks to the Author):

Comment: In this manuscript, the authors explore an important question regarding how mitochondria naturally become dysfunctional with age, using *C. elegans*. Though an impressive longitudinal proteomic analysis of two well-characterised long-lived mutant *C. elegans* strains

with perturbed mitochondrial function, the authors identify the phosphatidylcholine synthesis pathway as a protective pathway in the mutant *C. elegans* with mitochondrial dysfunction. The authors discover this pathway is decreased with age in WT *C. elegans*, and disruption of this pathway leads to a fragmented mitochondrial phenotype. Remarkably, dietary supplementation with choline and SAM was able to rescue the fragmented mitochondrial morphology. Supplementation with Choline was also found to be protective in a mammalian cell culture model of acute mitochondrial stress.

The comparative analysis of aged *C. elegans* vs. *C. elegans* with mitochondrial dysfunction that have developed protective adaptations is a very smart method of identifying cellular pathways that are lost with ageing. Overall, the data in this manuscript is high quality, but some additional experiments are required to robustly support the authors conclusions. In my opinion, the manuscript will be suitable for publication if the revisions suggested below are addressed.

Response: We thank the reviewer for their positive assessment of our study and the insightful comments.

Comment: Signs of UPR^{mt} activation were seen in the mutant *C. elegans* used for the proteomics analysis (Fig S1) as would be expected from the persistent mitochondrial stress in these strains. In contrast, minimal UPR^{mt} activation was seen in the *pmt-1* RNAi strain (Fig 5d), and as the authors described on pg8, the UPR^{mt} activation seen in the *sams-1* RNAi strain appears to be through a methylation dependent mechanism instead of through mitochondrial stress mediated activation. Collectively, this data is at odds with the mitochondrial fragmentation patterns observed in these strains (Fig 3b, Fig 5b) and would instead suggest that mitochondrial dysfunction is minimal in the absence of the PC synthesis pathway.

Response: Given that UPR^{mt} is mainly a response to the disruption of mitochondrial proteostasis⁷, not every type of mitochondrial dysfunction is associated with elevated UPR^{mt} activity. For instance, in previous work we found metformin treatment to strongly interfere with mitochondrial respiration without activating UPR^{mt}. However, the reviewer is right in suggesting that this point requires clarification, and we added the relevant discussion to the text (page 9, lines 5-10). Ultimately, as suggested in the below comment, we used *in vivo* respirometry assays to

demonstrate that knockdowns of *sams-1*, *pmt-1* and *pmt-2* impair mitochondrial function comparably (Figure 5d and S7d) although only *sams-1* KD strongly activates UPR^{mt}.

Comment: Due to these conflicting observations, it is important for the authors to perform additional experiments specifically examining mitochondrial function to be able to robustly conclude that the age-dependent changes in PC metabolism are driving mitochondrial dysfunction. Such experiments could include seahorse analysis of the respiratory capacity of mitochondria isolated from the different worm strains, BN-PAGE analysis and activity assays of respiratory complexes from isolated mitochondria, or JC-1 staining of mitochondria to measure membrane potential. These experiments should be performed in the worms with *sams-1* and *pmt-1* RNAi, and in the aged worms compared to young worms to demonstrate the effect of PC metabolism perturbation on mitochondrial function.

Response: We thank the reviewer for this helpful suggestion. As proposed, we performed whole animal Seahorse experiments in (i) young control as well as *sams-1*, *pmt-1* and *pmt-2* knock-down animals (Figures 5d and S7d), (ii) *pmt-1* KD animals with and without PC supplementation (Figure 5e) and (iii) in post-reproductive *C. elegans* with and without choline supplementation (Figure 6f). In the latter case, we used choline because its water solubility allowed us to avoid long-term exposure of the animals to organic solvents such as ethanol, which are required as a vehicle for PC. These assays showed that all tested knockdowns, as well as aging, caused a marked reduction in OCR, while PC or choline supplementation effectively rescued this defect in the *pmt-1* knockdown and aging contexts respectively. These comprehensive new tests, along with additional lipidomics measurements (Figures 6a, b, and d; S8b–c), provide strong evidence linking alterations in methylation-dependent phosphatidylcholine (PC) synthesis to mitochondrial dysfunction through changes in mitochondrial network integrity.

Comment: While the GTEX data and the cyto/mito-protective role of Choline supplementation during metformin treatment are very interesting, the GTEX data does not examine mitochondrial integrity which is the focus of this manuscript, and the metformin treatments do not replicate a phosphatidylcholine deficient scenario as is proposed to occur with ageing. As a result, the data

included from the mammalian systems is providing correlative support of the main conclusions in the manuscript. I acknowledge that replicating ageing conditions in a mammalian system is extremely complex and difficult and I do not expect this to be done in this manuscript. However, the discussion regarding the mammalian data should be toned down to reflect the correlative nature of the data.

Specifically: Line 22 p2: the specific experiments performed relating to the human biological system should be specified (e.g. editing that sentence to read ‘In this study, we combined omics, genetics and functional analyses in *C. elegans*, with genetic analysis in humans and functional experiments in cell culture models...’)

Response: We amended the text as suggested (page 3, lines 3-5). Furthermore, to reinforce the human aspect of our study, we examined metabolomics data from the UK Biobank human cohort. Mitochondrial fission and, particularly, fusion depend on membrane curvature and fluidity. Our analysis showed marked alterations in the levels of phosphatidylcholine (PC) and other lipids (MUFA, PUFA, and SFA) that are critical for membrane curvature and fluidity, both during aging and in individuals with metabolic disorders like obesity and diabetes (Figures 7c–h; S14a-b and S15a-c). Furthermore, higher levels of phosphatidylcholine (PC) were associated with improved metabolism and better survival prognosis during human aging (Figure 7h). While correlative and descriptive, these new analyses support the model that aging-related changes in PC levels may contribute to mitochondrial and metabolic impairments across species (page 16, line 5 – page 18, line 5).

Comment: Lines 14 – 17 p14, sentence starting with ‘Specifically, we hereby...’: While changes to mitochondrial dynamics have been identified when the PC pathway is perturbed, it cannot be concluded from the data provided in this manuscript that the defect observed is specific to the mitochondrial fusion/fission processes. This line should be edited or removed as the data does not show that targeting mitochondrial fusion could be a therapeutic target during ageing.

Response: We have amended the text as suggested (page 21, lines 13-15).

Comment: The ontology analyses provided throughout the manuscript are very well done. To help

viewers understand the significance of the ontology categories labelled in the main text figures (Fig 1b-h), data including the pValue and number of genes included in each category should be labelled next to the category name in the figures in addition to being included in the supplementary tables.

Response: We understand the reviewer's reasoning, but we find it challenging to apply in our specific case. For instance, in Figure 1b, when focusing on pathways shared between old and middle age, the term “lipid metabolism” encompasses three Gene Ontology (GO) terms: “Metabolism: lipid: binding”, “Metabolism: lipid: beta oxidation”, and “Metabolism: lipid: fatty acid” (Table S9). Each of these terms has its own gene enrichment (RGS/AC ratio) and *p*-value. The same applies to the term “Ribosome” which includes “Ribosome: subunit” and “Ribosome: biogenesis”. In this case, we believe that providing general indications in the figure, along with specific terms and numbers in the table, is the most effective approach.

Comment: Representative images for the mitochondrial morphology analyses in Fig 4B, Fig 5b, Fig 6a and b, Fig 7d should be included in the main text or supplementary figures.

Response: We now included the images in Figures S2a, S5, S10a.

Comment: Figures with N2 should also be labelled WT (i.e. WT (N2)), to make it easier for readers not as familiar with the *C. elegans* nomenclature to understand the figures.

Response: The figures were revised by replacing N2 with WT, as we opted not to use WT(N2) to avoid cluttering the figure labels. N2 is introduced as WT in Figure 1a and its corresponding legend.

Comment: The rationale behind the inclusion of the control RNAi of *mip-1* should be referenced in the main text of the manuscript.

Response: The rationale was added as requested (page 7, lines 20-22).

Comment: Line 17 pg 6: A space is missing between 'sams-1' and 'KD'.

Response: This has been corrected.

References:

- 1 Espada, L. *et al.* Loss of metabolic plasticity underlies metformin toxicity in aged *Caenorhabditis elegans*. *Nat Metab* **2**, 1316-1331 (2020). <https://doi.org:10.1038/s42255-020-00307-1>
- 2 Fuller, N. & Rand, R. P. The influence of lysolipids on the spontaneous curvature and bending elasticity of phospholipid membranes. *Biophys J* **81**, 243-254 (2001). [https://doi.org:10.1016/S0006-3495\(01\)75695-0](https://doi.org:10.1016/S0006-3495(01)75695-0)
- 3 Ohki, S. & Arnold, K. Experimental evidence to support a theory of lipid membrane fusion. *Colloids Surf B Biointerfaces* **63**, 276-281 (2008). <https://doi.org:10.1016/j.colsurfb.2007.12.010>
- 4 Hansen, M., Hsu, A. L., Dillin, A. & Kenyon, C. New genes tied to endocrine, metabolic, and dietary regulation of lifespan from a *Caenorhabditis elegans* genomic RNAi screen. *PLoS Genet* **1**, 119-128 (2005). <https://doi.org:10.1371/journal.pgen.0010017>
- 5 Lim, C. Y. *et al.* SAMS-1 coordinates HLH-30/TFEB and PHA-4/FOXO activities through histone methylation to mediate dietary restriction-induced autophagy and longevity. *Autophagy* **19**, 224-240 (2023). <https://doi.org:10.1080/15548627.2022.2068267>
- 6 Morana, A. *et al.* Stabilization of S-adenosyl-L-methionine promoted by trehalose. *Biochim Biophys Acta* **1573**, 105-108 (2002). [https://doi.org:10.1016/s0304-4165\(02\)00333-1](https://doi.org:10.1016/s0304-4165(02)00333-1)
- 7 Uoselis, L. *et al.* Temporal landscape of mitochondrial proteostasis governed by the UPR(mt). *Sci Adv* **9**, eadh8228 (2023). <https://doi.org:10.1126/sciadv.adh8228>

We appreciate the positive feedback from reviewers 1, 3, and 4, who, based on our previous revision, indicated that they have no further questions. Below, we present a point-by-point response to the remaining comments from reviewer 2. All revised sections in the text files are highlighted in grey. These revisions include changes made in direct response to reviewer 2's comments, as well as additional modifications we introduced independently to enhance the clarity of data presentation and discussion.

Reviewer #2 (Remarks to the Author):

Poliezhaieva et al. present a manuscript exploring connections between aging, levels enzymes contributing to the production of S-adenosylmethionine (SAM) to phosphatidylcholine (PC) and mitochondrial phenotypes in *C. elegans*. They also provide information from human samples in the UK Biobank. Low SAM is associated with models of lifespan extension across eukaryotes and SAM synthesis is one contributor to phosphatidylcholine levels. Their starting point in these studies were proteomics comparisons of aging *C. elegans* and two strains with mutations in mitochondrial proteins, which highlighted changes in the SAM synthase SAMS-1 as the animals age, along with changes in the enzymes which use SAM to produce PC in the methylation dependent pathway. They go on to investigate mitochondrial morphology and measure aspects of mitochondrial function in these animals and add back SAM or PC several of the assays, concluding that PC levels drop in normal aging and that levels can be rescued by diet.

While this subject area is of general interest and there are potential interesting observations, there are multiple issues with rigor and approach to published literature that reduce enthusiasm. First, key aspects of the data have been reported previously and either not cited or cited poorly. Second, there are issues with the analysis and interpretation of the data in multiple instances. Finally, the author's metabolic pathway descriptions are in some places overly simplified, reducing confidence in the interpretations. Alternative explanations for the decrease in PC in *C. elegans* during aging, such as the upregulation of PC consuming phospholipases that occurs with age are not considered. For example, this is noted in a similar proteomics study of aging *C. elegans* Copes, et al. 2015, that also finds decreases in SAMS-1 and SAM during *C. elegans* aging. Other pathways for producing PC (the Kennedy pathway) are the most dominant in mammals and the role that the rate limiting enzyme, *pcyt-1*, could have in this process is not considered.

Response: We respectfully but firmly disagree with the assertion that our interpretation of the literature and experimental data lacks rigor or depth. We believe these concerns are unfounded, as demonstrated in our detailed point-by-point response below, which addresses each issue with clarity and supporting evidence. The age-related decline in phosphatidylcholine (PC) levels coincides with a pronounced downregulation of the entire SAM-dependent PC synthesis pathway in the nematode. Moreover, this phenotype is robustly recapitulated in young animals following knockdown of key enzymes in the pathway, namely SAMS-1 and PMT-1. Together, these findings strongly support the conclusion that impaired SAM-dependent PC synthesis is a primary driver of this aging-related change. Nonetheless, in response to this comment, we have revised the manuscript to explicitly acknowledge that other mechanisms - such as increased PC turnover via phospholipase activity, may also contribute (page 20, lines 5-6). Contrary to the reviewer's suggestion, we had previously addressed the age-associated changes in the Kennedy pathway in Figure S8d, Table S25, and the corresponding section of the main text. However, upon receiving this comment, we realized that our discussion of this aspect may not have been sufficiently prominent. We have therefore revised the figure, table, and text to better highlight this point and ensure its improved visibility (page 12, line 22 – page 13, line 4). In brief, the Kennedy pathway components *cka-1*, *cka-2*, and *pcyt-1* were detected in our proteomics analysis (Tables S2 and S25), but none of them met the threshold for the most prominent age-associated changes reported in Table S23. Additionally, some of the individual changes were not statistically significant, as indicated by the respective Q values (Table S25). While there is a trend toward downregulation with age, the magnitude of this trend is modest compared to the progressive and coordinated decline observed for SAMS-1, PMT-1, and PMT-2 (Figure S8d vs. Figures 2a and b, 6c and S8a). Finally, the fact that choline supplementation remains effective in counteracting late-life mitochondrial disruption (Figure 6e-f) suggests that the Kennedy pathway retains its functionality during aging, in contrast to the methylation-dependent route of PC synthesis.

Comment: The proteomics data is analyzed using WormCat, a *C. elegans* specific tool for pathway analysis. There are multiple issues with this analysis. First, the authors describe it as a “gene set enrichment analysis” (pg 75, line 16). Gene Set Enrichment (<https://www.gsea-msigdb.org/gsea/index.jsp>) is a distinct type of analysis that uses a ranked list of genes providing

simultaneous evaluation of up and downregulated genes while WormCat uses enrichment statistics to evaluate all genes in a set without regard to expression level. Second, the authors do not supply information regarding the background list used for WormCat (whole genome or ORF). This is critical as proteomics data is sparse, and it is not appropriate to use whole transcriptome background lists for proteomics studies. Next, there are issues with the way the data is used to make the Venn Diagrams in Figure 1 f, g, and h. It is not clear which category level the authors are reporting, and it is not consistent to break down some categories (Metabolism) and not others. Even if the end conclusions are similar, the lack of reporting and rigor with this analysis reduces confidence in this study.

Some of these critiques were noted in the original review. While in the rebuttal, the authors state Various tools for enrichment analysis are available for transcriptomic and proteomics studies, however, they should be used with precision and an understanding of the strengths and weaknesses in each approach. WormCat is still referred to as a “gene set enrichment analysis” (pg 75, line 16) in the present text.

Response: In the previous revision, we sought to correct the unfortunate mistake of naming WormCat a GSEA tool. However, we regret that one instance remained in the Materials and Methods section, as the reviewer helpfully pointed out. We appreciate this observation, and the sentence has now been corrected (page 7, lines 27-28 of the Methods file). In response to this comment, we have clarified both the reference background used for the WormCat analysis and the category level reported in the Methods section (page 7, lines 28-30 of the Methods file). While only the relevant portions of the data (i.e., metabolic pathways) are visually highlighted in the figure (Figure 1 f-h), the full dataset is provided in the corresponding supplementary tables (Tables S13, S17 and S20). Additionally, the raw input used for the WormCat analysis is included in Tables S7, S12, S16 and S19 enabling any reader to reproduce the analysis from scratch to validate our findings. We therefore believe that the data and analyses have been reported fully and transparently.

Comment: One of the authors major rationales in the study is the change in SAMS-1 during aging *C. elegans* proteomes. This has already been observed in multiple studies. While a subset of these

studies is mentioned in the discussion, this should be cited as a confirmatory result with the initial observation.

- Dong et al. show SAMS-1 decreases with age:

<https://www.science.org/doi/full/10.1126/science.1139952#con7>.

- Copes et al. use quantitative proteomics to compare young adult and aging *C. elegans* and find that both SAMS-1 and SAM decrease.

-Walther et al. <https://www.sciencedirect.com/science/article/pii/S0092867415003207>

This was noted in the original review. Regarding the notion that “none of the above manuscripts specifically focused on this change”, SAMS-1 is specifically investigated in the first two of these studies. Thus, the present study provides confirmatory and incremental advances in our knowledge of how the SAMS-1 protein changes during ageing.

- The decrease in SAMS-1 and PMT-1 is shown in Figure 1A of Dong et al.

- Copes et al. find the pathway that was “most changed with age was the biosynthesis of amino acids” including the S-adenosylmethionine pathway. This is mentioned in the abstract and in both the proteomics and metabolomics sections of the study and the decrease in SAM draw attention to the effects on metabolic function. While Walther et al. do not directly address the reduced levels of SAMS-1, Copes et al compare their data with Walther and note that SAMS-1 is decreased in both (see discussion). It is notable that the present study does not include such a comparison.

Response: Our primary rationale is clearly outlined on page 6, lines 7-9 of the main text, and reads as follows: "*To further explore pathways that mediate adaptive responses to mitochondrial dysfunction, we next searched for individual proteins whose expression is strongly and progressively altered during advanced aging in WT animals, but whose changes are attenuated in the aging mito-mutants.*" The comparison between wild-type aging and aging in the mito-mutants represents the central aspect of our rationale. None of the studies cited by the referee adopt a similar rationale or use a comparable approach, which limits their direct relevance and comparability to our work. As noted in our previous response to this reviewer, we acknowledge earlier reports of *sams-1* downregulation during normal aging, and we actually find this consistency with our own findings reassuring. With respect to the sentence from our previous response highlighted by the

referee (e.g., “none of the above manuscripts specifically focused on this change”), our intention was to convey that none of the manuscripts experimentally validated the cellular consequences of age-associated decline in SAMS-1 levels, which remains accurate. In response to this comment, we have amended our list of references to ensure that relevant prior studies reporting age-related changes in SAMS-1 at the protein level are included (page 6, lines 14-16 and page 19, lines 2-3).

Comment: Both Reviewer’s 2 and 3 noted questions about the combination of sams-1(RNAi) with effects on the lifespan of the mitochondrial mutants. The lifespan studies are very difficult to interpret without additional controls.

Although clk-1 and isp-1 are published to have longer lifespans than wild type animals, there are no wild type controls in Figure 2 c-e. It appears, especially for isp-1, that the mean lifespan is not that different from the wild type animals. If sams-1 has an effect on mitochondrial function, two mitochondrial problems might be enough to tip the balance from mito-hormesis to failure to thrive. The combination of effects fails to show the specificity claimed, because there are no additional controls or analysis of other mutants.

Response: The wild-type controls, which were conducted in parallel, are indeed shown in the figure (panel c), thereby addressing the referee’s concern. To clarify this, we have updated the figure legend to explicitly state that panels c–e present results from the same experiment separated into different panels for convenience (page 2, lines 14-16 of the Figure legends file).

Comment: As mentioned in the previous review, fission of sams-1 mitochondria have been previously published by Wei and Ruvkun.

https://www.pnas.org/doi/10.1073/pnas.2008021117?url_ver=Z39.88-

[2003&rfr_id=ori%3Arid%3Ahttps://doi.org/10.1073/pnas.2008021117&rfr_dat=cr_pub++0pubmed#supplementary-materials](https://www.pnas.org/doi/10.1073/pnas.2008021117?url_ver=Z39.88-2003&rfr_id=ori%3Arid%3Ahttps://doi.org/10.1073/pnas.2008021117&rfr_dat=cr_pub++0pubmed#supplementary-materials)

The increase in mitochondrial fission is mentioned in the summary and is shown in the Figure S12 (Inactivation of sams-1 increases mitochondrial fission), where effects on both sams-1 RNAi on wild type animals with a muscle reporter of mitochondrial morphology and comparison to drp-1(lof) animals is shown.

Although Wei and Ruvkun's rationale stems from low SAM due to lysosomal changes, the context of *sams-1* knockdown in wild type animals causing mitochondrial fission is not sufficiently different to the present study to preclude discussion. It is not clear why the authors focus on figure 3A of Wei and Ruvkun, rather than the data directly with *sams-1* (Fig S12), as the reviewer states "MS Figure 3A" referring to the author's own manuscript, not the figure in the Wei and Ruvkun study.

Response: As noted in our previous response to a related comment - and as the reviewer has now reiterated, the context of Wei and Ruvkun's work (i.e., lysosomal changes) differs significantly from ours (i.e., aging) limiting direct comparability. Nonetheless, as already stated in the previous point-by-point response, we agree that their findings warrant discussion and citation. Accordingly, both were included already during the previous revision (page 19, lines 10-11).

Comment: In the previous review, there was a concern about the dietary addition of SAM. There are conflicting reports about the efficacy of adding SAM to media and the reviewer's request was to show that the phenotypes of the *sams-1* mutant or RNAi animals were rescued along with SAM levels. (See Sun and Locasale, Cell Stress 2022 for discussion of issues with SAM lability).

The authors were unsure why the request would be made to look at other phenotypes. (pg 9 of the rebuttal)

If SAM were able to be taken up, then multiple phenotypes of the *sams-1* mutant or RNAi animals (lipid droplets, body size) would be rescued, not just the *hsp-6::GFP* expression. If SAM does not change body size, as the authors state in the rebuttal, then it appears even more likely that the SAM in the diet may be labile due to its instability and not taken up.

Response: In the previous revision, we devoted an entire figure (Figure S11) and a substantial portion of the text (page 15, lines 7-21) to evaluating the efficacy of SAM supplementation, using a range of assays - from lipidomics and *in vivo* respirometry to body size and mitochondrial reporter assays. Our conclusion aligns with the reviewer's comment and prior literature: the effects of SAM supplementation are variable, likely due to its known low stability. This is an unfortunate limitation when working with SAM, which cannot be easily resolved, and it restricts the strength

of conclusions we can draw from the SAM supplementation studies, as we have transparently discussed and stated in the manuscript.

Comment: In Figure 5 B SAM is added and concluded to show changed in mitochondrial fragmentation. However, the magnitude of the change between *sams-1* with and without the SAM is quite low and measured by eye, rather than a quantitative assessment with a tool such as mitoMAPR (<https://www.life-science-alliance.org/content/7/11/e202402918>). While the authors report statistical significance, the fold effect of the added SAM is incremental at best.

Response: The method we used to quantify mitochondrial morphology is based on high-resolution microscopy scoring and is widely recognized and accepted in the field (Regmi SG et al, Aging, 2014; Weir HJ et al, Cell Metabolism, 2017 and others). Although the effect size may appear modest - a perception that is indeed subjective, it aligns with the above discussed low bioavailability of SAM, a point also recognized by the reviewer. Crucially, the observed differences are statistically significant, supporting their valid reporting and the conclusions derived from them.

Comment: Concerns remain about the over reliance on the *hsp-6::GFP* reporter as the only measure of the mitoUPR, especially when the authors note inconsistencies with their measures of mitochondrial phenotypes, potential rescue and this reporter.

Response: Because the reviewer did not cite specific figures or sections of the text that possibly contain inconsistencies, it is difficult to comment on this point in detail. However, we note that, in general, we neither observed nor reported any meaningful inconsistencies in our measurements of mitochondrial phenotypes.

Comment: SAM is used in multiple biosynthetic reactions and for regulatory modification of proteins and nucleic acids. The authors rebut that chromatin studies are not relevant to this study with PC, but the work by Ben Tu's lab (Ye Mol Cell 2017) cited above, show that changes in methylation dependent PC synthesis in yeast directly affect histone methylation and this was

confirmed in *C. elegans* (Ding, et al. PLoS G, 2018). Thus, these processes are highly relevant to the present study and represent alternative hypotheses that should be considered.

Response: The complete rescue of mitochondrial morphology defects in *sams-1* knockdown animals through PC boosting clearly demonstrates that impaired SAM-dependent PC synthesis is the underlying cause of these defects. In this context, investigating the impact of SAM on unrelated cellular pathways is not meaningful and falls outside the scope of our study.

Comment: Multiply copy arrays, such as the *hsp-6::GFP* reporter, are subject to regulation at the level of histone methylation in *C. elegans* and such reporters have been shown to be directly affected by SAM synthases

(<https://www.sciencedirect.com/science/article/pii/S0092867412009439>). Thus, changes in activity of a single multi copy array especially when not accompanied by measurement of the endogenous gene are insufficient for the conclusions on the mitoUPR made in this study. While the author's note the alternative explanation, they do not use any distinct reporters or test endogenous genes, nor do they discuss the discrepancy with Hou et al, who show rescue of the reporter when choline is used to support PC levels.

The Walker et al study does not include the mitoUPR reporter, instead they use the *hsp-4::GFP* ER stress reporter.

Response: We would like to respectfully note that, for some of the cited works - such as "Hou et al" and "Walker et al", the information provided was insufficient for us to identify the specific studies with certainty. Consequently, we are unable to assess the comparability of our study with these references. Another referenced study

(<https://www.sciencedirect.com/science/article/pii/S0092867412009439>) explored the role of SAM-dependent histone methylation in silencing of gene arrays; however, it did not investigate the specific *hsp-6p::GFP* reporter used in our study and is therefore not directly applicable to our work. Despite its potential limitations, the *hsp-6p::GFP* reporter is a long-standing and well-established tool in the field for measuring mitochondrial UPR activity *in vivo* and in real time (Bennett CF et al, Nat Communications, 2014; Haeussler and Conradt, Methods Mol Biol, 2022 and others). Nevertheless, histone methylation could influence the expression of this reporter as part of a broader effect on UPR MT and *hsp-6* promoter activity, as noted in the text (page 9, lines

10–12). We also clearly indicate that this interpretation is derived solely from published literature. Since the *hsp-6::GFP* reporter and the mitochondrial UPR are not central to our study and represent only secondary observations, we do not believe that additional, more elaborate experiments in this direction are warranted.

Comment: Concerns remain about the author’s explanation of metabolic/lipid synthesis pathways.

– PC has multiple sources. Biosynthesis by the Kennedy pathway, which refers to the production of PC from choline, is the predominant pathway in mammalian cells (see <https://pmc.ncbi.nlm.nih.gov/articles/PMC124149/#:~:text=Phosphatidylcholine is synthesized almost exclusively,pathway being cell type dependent>) and also a major contributor in *C. elegans* (see <https://academic.oup.com/genetics/article/207/2/413/5930737?login=false>).

Response: As demonstrated by the lipidomics data in Figure 6a, knockdown of either *sams-1* or *pmt-1*, which disrupts the SAM-dependent PC synthesis pathway, is sufficient to reduce whole-organism PC levels. This highlights the key functional contribution of this PC synthesis route and supports our proposed model.

Comment: Dietary phospholipids are digested by phospholipases (<https://www.sciencedirect.com/science/article/pii/S1878818120304357>), releasing lysophospholipids and then choline which are readily absorbed. In *C. elegans*, Copes et al. show that these phospholipases are highly expressed in aging animals and thus PC digestion would also be supported. Since choline has already been shown to rescue PC levels in *sams-1*(RNAi) animals (Ding, Cell Met 2015, <https://www.sciencedirect.com/science/article/pii/S1550413115003423>), it is likely this choline is providing the rescue, as has been previously shown, and not the PC.

Response: We would like to respectfully point out that the very study cited by the reviewer, Copes et al. (2015), reports that the abundance of phospholipase D3 - an ER membrane protein responsible for converting PC into choline and phosphatidic acid, declines with age. This finding contrasts with the reviewer’s interpretation and instead supports the notion that the conversion of PC to choline may become less likely during aging. While it is true that PC can be initially broken

down into LPC and FFA during intake, it is largely re-synthesized inside cells by lysophosphatidylcholine acyltransferases (LPCATs). Thus, both PC supplementation and choline supplementation represent valid strategies to increase organismal PC levels, with choline serving as a precursor for PC biosynthesis.

Comment: In the rebuttal, the author's mention two studies as "invalidating" the need to consider PC digestion. However, the papers do not invalidate this point. Bai et al show that PC in the yeast plasma membrane can be flipped by a translocase across one leaflet to the other. The PC is already in a membrane; the translocase is not acting on free PC nor is the PC added to an animal's digestive tract. Bai et al do add exogenous PC, however it is a non-physiological 7-nitrobenz-2-oxa-1,3-diazol-4-yl PC with a 6 carbon acyl chain to increase solubility for their structural and biochemical studies. The closest ortholog of this P4 ATPase in *C. elegans* is also predicted to reside inside the cell, not at the plasma membrane. The Sleight and Abanto study also provides very short chain NBD PC in liposomes to cultured cells at 2°C, which is non-physiological in comparison to PC added to *C. elegans* media.

Response: In our hereby referenced comment, we solely addressed the reviewer's incorrect suggestion that PC can enter cells only in a digested form, which is not accurate. We did not, and do not, dispute the well-established fact that PC can be converted to LPC and FFA during uptake and subsequently re-synthesized inside the cell. To resolve this misunderstanding, we have now revised the manuscript to clarify this point (page 21, lines 11-14).

Comment: Thus, the simplest explanation of the PC rescue is that it is digested, with choline contributing to the rescue effect. Label and tracing experiments would be necessary to demonstrate direct uptake, which are likely to be beyond the scope of the present study.

Response: Contrary to this suggestion, we hope to have demonstrated in the above responses that PC likely provides rescue either by being digested into LPC and FFA during intake and subsequently resynthesized within the cell, or by entering the cells directly through flipping. We agree with the reviewer that labelling and tracing experiments are beyond the scope of the current study.

Comment: OCR assays with *sams-1* and *pmt-1* animals are difficult to interpret. The authors state that equal numbers of worms were added for control, *sams-1* and *pmt-1* animals. However, with the large difference in size, it would be expected that there are fewer mitochondria solely based on the fact that the animals are so much smaller. Thus, the reduction in OCR could be due to smaller animals having few mitochondria.

Response: Normalization to animal number is appropriate in *C. elegans* whole animal OCR assays, allowing assessment of respiratory capacity per individual worm. While we cannot rule out that smaller animals may contain fewer mitochondria, it is unlikely that the observed differences are driven solely by size. For example, *pmt-1* knockdown worms are larger than *sams-1* knockdown worms (Figure 5c) but exhibit similar or even lower OCR capacity (Figure 5d). Moreover, the literature does not provide consistent evidence linking smaller body size to reduced mitochondrial counts in *C. elegans*. Instead, there is abundant evidence identifying mitochondrial dysfunction as a cause of smaller body size, which is consistent with our observations (Curran SP et al, JBC, 2004; Yang W and Hekimi S, Aging Cell, 2010 and others).

Comment: The authors have several figures showing rescue of PC by choline in the diet (Figure 6). This has already been shown by Ding, et al (see above comment). Lipidomic analysis is usually presented as normalized to total lipid, protein or relative to a labeled standard. It is not clear that the ratios add much in comparison to the unmodified data.

Response: The rescue of PC levels by dietary choline is not presented as a novel finding in our study but rather used as an established tool. We have added a reference to Ding et al. as prior validation of this approach (page 10, lines 8-9). All lipidomics data presented are normalized to both the internal standard and the number of worms, as clearly stated in the respective figure legends and the Methods section and the complete raw data is presented in Table S24. The lipid ratios (e.g., PC/PE and LPC/LPE) were included in response to a specific request from Reviewer 1. We agree with Reviewer 1 that these ratios are valuable for highlighting specific alterations in methylation-dependent PC synthesis, as opposed to detecting a general decline of phospholipid levels.

Comment: Concerns still remain about the author's statements regarding vitellogenins. In this study, the authors describe vitellogenins as “core components” of lipid droplets based on Espada, et al. one of their previous studies and in the rebuttal, the list two proteomics papers as proof.

Vitellogenins are yolk proteins, produced in the intestine and secreted into the germline (reviewed in <https://pmc.ncbi.nlm.nih.gov/articles/PMC6736625/>). To ignore this major role obscures their dominant function. There are abundant labeled markers and genetic studies showing accumulation of VIT proteins in intestinal endosomes (easily distinguishable organelles from lipid droplets), exocytosis and uptake into the germline by receptor mediated endocytosis (work by Barth Grant and others).

VITs are also among the most abundant proteins in *C. elegans*. If present as minor components of lipid droplet proteomes, they could be contamination if not independently verified. Zhang, et al., mentioned in the rebuttal, contains an immunoblot comparing VIT-2 in comparison to other lipid droplet proteins, however the levels are lower in the LDs than other compartments and they conclude that “Western blotting analysis showed that VIT-2 was only partially associated with LDs and the main signal was in total membrane fraction (Fig. 2C)”. Thus, Zhang, et al does not support a major role for VIT-2 in LDs.

Response: The statement that vitellogenins are key protein components of lipid droplets is supported by the two cited proteomics studies, not Espada et al. (2020). Espada et al. was cited as our previous work employing a specific proteomics method to quantify VIT protein levels and correlate them with neutral lipid staining using Oil Red O. In our earlier response, we addressed the explicit reviewer's claim that VITs are not part of lipid droplets, which is incorrect. However, we do not dispute that VITs are components of yolk droplets - a function that is less relevant in post-reproductive animals, the stage at which our measurements were conducted. VIT-2 is just one of six vitellogenin proteins in *C. elegans* and does not reflect the distribution of the entire protein family. Additionally, its suggested highest abundance in the membrane fractions does not align with its role as a yolk protein either.

Comment: Concerns remain about the argument that choline supplementation is protective in mammalian cells. Mammals can convert choline to SAM via BHMT, therefore SAM could act through multiple mechanisms, and it is not a PC-specific treatment.

Response: While choline supplementation can ultimately support SAM synthesis within the cell, it is more likely utilized primarily as a precursor for PC synthesis, as hinted in the study by Ding et al., which the reviewer previously cited. The combination of multiple lines of evidence presented in this study strongly supports a primary role of PC in the observed effects, including the outcomes of choline exposure. However, in response to this comment, we have added a sentence to the discussion acknowledging that contributions from other choline metabolites cannot be fully excluded (page 19, lines 18-21).

Comment: Concerns remain about the description of PEMT as a “functional analog” of *C. elegans* PMT-1/2. While the authors changed the language from ortholog (as there is no sequence similarity and differences in enzyme function), this statement is an oversimplification. The authors also corrected the conversion steps; in *C. elegans*, PMT-1 and PMT-2 convert phosphoethanolamine to phosphocholine and in mammals the conversion occurs after the acyl chains have been added from PE to PC.

However, these are not analogous reactions physiologically. In both cases, the methyl groups on PC are obtained from SAM rather than choline, but in mammals, PEMT acts tissue specifically and methylates PE so that PEMT-produce PC has different characteristics that PC produced by the Kennedy pathway. In *C. elegans*, since the methylation occurs before acylation, both are likely to contribute to bulk synthesis and as SAMS-1 has a wide distribution pattern, is likely to act across the animal. Due to the distinct natures of these enzymes, substrate kinetics and other regulatory mechanisms are not likely to be conserved. Although most PC synthesis does occur in the ER, as the authors note in their rebuttal, PEMT has been purified in mitochondrial fractions (<https://www.sciencedirect.com/science/article/pii/S0005273613003799>), providing a potential mitochondrial source of PC.

Response: We acknowledge the outlined differences in methylation-dependent PC synthesis between mammals and *C. elegans*, which we have explicitly clarified in Figure S3a. However, in

both systems, PC production is expected to be limited by reduced SAM availability during aging, supporting the analogy we draw. Additionally, our human metabolomics analysis confirms age-related changes in PC levels and associated physiological alterations that can be interpreted as consequences of mitochondrial impairment.

Comment: Although statements regarding PC being a precursor to phosphatidic acid were changed to say that PC could make PA through action of phospholipases there are still concerns regarding the author's incomplete descriptions these complex metabolic pathways.

While PC can be acted on in a signaling pathway to produce PA, PA can be made by 3 other pathways and it is also a precursor for all glycerolphospholipids. This makes it also a precursor to PC, providing the acyl chains after conversion to diacylglycerol by DAK kinases. Linking its levels solely to PLD ignores its major biosynthetic pools, and alternative sources of this lipid and over simplifies this complex pathway which affect confidence in the conclusions. (<https://www.sciencedirect.com/science/article/pii/S1388198119300460?via=ihub>).

Response: The reviewer is right to point out that PA synthesis is more complex and bidirectional, with PA serving both as a precursor to PC and vice versa. However, we do not believe our manuscript contains any incorrect statements. We accurately noted that PC can contribute to changes in mitochondrial morphology by serving as a precursor to PA. However, our primary hypothesis remains that PC and LPC directly influence mitochondrial dynamics through their effects on membrane curvature and fluidity.

Comment: The authors also look at PC levels in the UK biobank data. However, it is not clear what tissue this is in. Since only the liver used PEMT as contributor to PC synthesis, it is not clear this is relevant to SAM-dependent processes. Furthermore, if its plasma, the PC is likely to be a component of lipoprotein particles such as VLDL and HDL, therefore, there could be multiple indirect ways this metabolite could change.

Response: As now clearly indicated in the text (page 16, lines 12–14 of the Main text and page 10, line 23 of the Methods file), the human measurements were obtained from EDTA plasma samples

and represent systemic levels. While it is true that lipoproteins transport plasma PC along with other phospholipids and neutral lipids to various tissues, we do not see how this physiological fact undermines any of our findings.

We greatly appreciate the positive and constructive feedback from Reviewer 5, as well as their kind interest in our findings. Below, we provide a point-by-point response to the remaining comments from Reviewers 2 and 5. All revised sections in the manuscript are highlighted in grey.

Reviewer #2 (Remarks to the Author):

Comment: Poliezhaieva et al present a manuscript that begins with a proteomics analysis of mitochondrial mutants as they age, which show decreases in mitochondrial proteins.

Response: While our global analysis of whole proteomes revealed extensive age-related changes in metabolic pathways, including those involving mitochondria, the directionality of these changes was not investigated or used to draw conclusions.

Comment: The authors then go on to select *sams-1* for further study and investigate changes in mitochondrial morphology as *sams-1* animals age. Next, they use dietary add back experiments to investigate how those might mitigate the physiological changes. Finally, the authors investigate metabolites in plasma from the UK BioBank samples. If the primary hypothesis is that SAM-dependent changes in PC synthesis are a driver of aging, the connection to the human data limited, as only a few tissues use SAM for a portion of its PC production in mammals. In their rebuttal, the authors point out that Kennedy pathway proteins are also present in their proteomics, but did not reach statistical significance. Thus, the relevance to human physiology is limited.

Response: We respectfully disagree with this statement. Our gene expression data (Figures 7a-b, S12, and S14) indicate that the SAM-dependent pathway is active in several human tissues and that its activity declines with age in these tissues. Notably, these include adipose tissue and liver, which are key organs involved in the systemic circulation of lipids. Our findings are also supported by previous literature. For example, in the liver, the phosphatidylethanolamine N-methyltransferase pathway is known to account for a highly significant proportion of total phosphatidylcholine synthesis (van der Veen et al., 2017, BBA Biomembranes), which we consider to be physiologically highly relevant. Therefore, our findings likely have significant implications for human physiology.

Comment: Several key aspects of this work (decline of SAMS-1 protein during aging and effects of *sams-1* RNAi on mitochondrial morphology) have been previously studied in multiple publications. Although the authors now cite these studies, the present works provides an incremental increase. For example, all of the *sams-1* data in figure 3 has been previously published. Additional issues with consistency in dietary update of metabolites, category analysis and

functional characterization of protein function (vitellogenins as representative lipid droplet proteins) continue to reduce enthusiasm.

Response: We respectfully disagree with the reviewer's assessment of the novelty of our work and the robustness of our experimental design and analyses. These aspects have already been discussed in previous revision rounds and are further clarified in the detailed responses below.

Comment: Figure 1:

In Figure 1, the authors show a proteomics comparison of wild type and mitochondrial mutants as they age. The authors then use WormCat, a *C. elegans* specific category enrichment program to identify functional groups within the changed proteins. They report changes in mitochondrial proteins, however the WormCat analysis used the whole genome annotation list, rather than the ORF set, which would be the appropriate background list for proteomics analysis (Higgins, Genetics 2022).

Response: In response to this comment, we repeated the analysis shown in Figure 1 and in Tables S8-10, S13-14, S17-18, and S20-22 using the ORF background. The updated analysis yielded all of the same conclusions as the original one, and the revised versions of the respective figure and tables now show the new analysis performed with the ORF set.

Comment: WormCat contains three levels of analysis, and although it is not identified on the figure, in the legend or in the supplemental file, the authors only show most granular level, Category 3. They do provide protein lists in the supplement, and while they require reformatting to be run in WormCat, it is notable that checking the 499 genes with the correct background list for the data in figure 1C (converted from table S12) does not return any category 3 data.

Response: In response to this comment, Tables S8-10, S13-14, S17-18, and S20-22 now include all available levels of analysis.

Comment: The authors also appear to add up the category 3 data to generate the venn diagram in figure 1F to approximate category 2 level data. This does not allow correct enrichment statistics, as the other gene/category sets from category 2 are not present in the statistical analysis.

Response: The pie charts in Figure 1f-h do not attempt to infer Category 2 data from Category 3 data. Instead, they present Category 3 data partitioned into metabolic and non-metabolic pathways. In response to this comment, we have updated the descriptions of the analysis in Tables S13, S17, and S20, as well as in the Methods section and figure legend, to clarify this point and prevent potential misunderstandings by readers.

Comment: The most important aspects for rigorous presentation of proteomics data with any enrichment tool are use of an appropriate background list (Zhao and Rhee, Trends in Genetics 2023) and transparency of the data used in the search. Thus, background lists designed for RNA seq studies are not appropriate for proteomics studies. The gene lists provided in the supplemental data require reformatting (CEL_ numbers and gene names to gene ID or Wormbase ID and additional curation to update gene names or resolve in-line duplicates), thus utility is limited.

Response: In response to this comment, we now provide reformatted lists for all relevant datasets in Tables S7a, S12a, S16a, and S19a, suitable for immediate use in WormCat.

Comment: Figure 2:

The raw data for each repeat of lifespan data would be important to show that figure 2c, d, and e were conducted as part of the same experiment, along with the independent replicates.

Response: As clearly noted in the figure legend, the survival measurements presented in Figure 2c-e originate from the same experiment, with the corresponding statistical source data available in the supplementary Source Data file.

Comment: Figure 3:

The data for *sams-1* in figure 3 has been reported in multiple studies (Wei and Ruvkun, PNAS 2020; Chen et al. Aging Cell 2024), thus the advance is incremental. As noted in the previous review, *hsp-6::GFP* is a multicopy reporter. The authors proteomics data show that as *sams-1* decreases, the endogenous *hsp-6* protein decreases (Table S4 Individual changes). Thus, evaluation of the endogenous RNA would also be a critical measure of mitoUPR induction, along with other genes induced in this process.

Response: Because changes in proteostasis during aging are not the focus of our study, we consider the proposed additional experiments to be outside its scope.

Comment: Figure 4:

Regarding evaluation of mitochondrial morphology (relevant to rescue experiments in figures 4-6):

The authors use visual scoring to evaluate mitochondrial morphology, rather than a computational approach. The studies they cite in the rebuttal are from 2014 and 2017. While this may be sufficient for large changes as in Figure 3A, the effect size from the addition of SAM is quite small. A more rigorous approach to these experiments (see Kim, et al. J Vis Exp 2025, Valera-Alberni, et al. Life

Science Alliance 2024) would employ quantitation of mitochondrial networks with FIJI using a program such as MitoMapr (Zhang et al Elife 2019).

Response: We have already addressed this comment in detail in the previous round of revisions. Additionally, we would like to emphasize that, while automated tools for assessing mitochondrial morphology offer certain advantages, they are also prone to artifacts such as false object recognition - particularly when analyzing complex 3D environments like large body muscle cells, as in our study. Therefore, we stand by the assessment approach we selected.

Comment: As mentioned in the previous critique, SAM is likely to break down and is unreliable addition to diet or cell culture (Sun and Locasale, Cell Stress 2021). Therefore, it is critical that the authors demonstrate SAM has direct effects in rescue experiments. In figure 4F and figure S2A, the effect of added SAM is quite low to moderate. Given that quantitation is by visual scoring in 4F and that no quantitation is provided in S2A, it appears that the ability of dietary SAM to rescue mitochondrial morphology is limited at best. See recent suggestions for best practices on cell biology regarding the importance of effect size (Morgan, JCB, 2025).

Response: As explained in detail in the previous round of revisions, we acknowledge the inherent limitations of SAM supplementation driven by its low stability and have dedicated an entire figure (Figure S11) as well as a substantial portion of the text (page 15, line 11 – page 16, line 2) to discussing and addressing these points.

Comment: There are inconsistencies compared to the treatment of SAM in the *hsp-6::GFP* experiments (Fig4C, FigS2b) where it appears have a more dramatic change on the GFP levels that the minor changes in the mitochondrial morphology. However, since HSP-6 levels decrease along with SAMS-1 in the aging studies and the endogenous *hsp-6* mRNA data is not supplied for the day 1 animals, this data does not support the notion that dietary SAM rescues the mitochondrial phenotypes.

Response: Mitochondrial UPR, as measured by the *hsp-6::GFP* reporter, and mitochondrial morphology represent two distinct aspects of mitochondrial biology that do not always change in parallel. Therefore, it is not reasonable to expect both to respond with the same intensity to every intervention or supplement. Nevertheless, both parameters show highly significant responses to SAM supplementation in Figures 4b-c and S2a-b, demonstrating the clear rescuing potential of SAM in this context.

Comment: As noted in the previous critique, loss of *sams-1* results in lipid droplet accumulation, small body size and reduction in brood size which are rescued by dietary choline. In figure S2, it does not appear the SAM alters the body size, thus limiting the rationale for rescue by dietary

SAM. Finally, SAM may be routinely measured by LCMS in *C. elegans* most recently in Choi et al. Nat Comm 2023 and Tharpa et al. NPG Aging 2023.

Response: This comment reiterates the limitations of SAM supplementation due to its low stability, which we have already addressed above. Although SAM does not rescue the reduced body size of *sams-1*-deficient animals in Figure S2b, it effectively rescues the aberrant activity of UPR MT in this figure as well as and mitochondrial morphology in Figure S2a (independent experiment), demonstrating that SAM is functional, albeit incompletely - likely a consequence of its well-known limited stability, as extensively discussed in our manuscript.

Comment: Figure 5 -6

Figure 5-6 explore the links between the mitochondrial phenotypes and PC by dietary addition of PC. As noted in previous critiques, PC entering an animal through the diet is digested into choline and lysophosphatidic acid. Given this, the simplest explanation for the rescue in Fig5A, B is the addition of choline. As pointed out in the previous critique, P4 ATPases can “flip” PC from the outer leaflet to the inner leaflet of the plasma membrane, however, this would require that PC enter the plasma membrane directly. Members of P4 family of ATPases in *C. elegans*, as pointed out in the previous critique, are also predicted to be on intercellular membranes. Thus, given the likelihood that PC is digested and choline provides the rescue, even with the added discussion in the text, this data appears to provide only an incremental increase in knowledge without more rigorous tracing studies.

Response: Even if dietary PC is digested into choline and lysophosphatidic acid during intestinal absorption, these molecules are most likely to serve as substrates for de novo PC synthesis within the cell. The data presented in this study - including molecular, functional, and lipidomic analyses, strongly support a key role for PC, rather than choline, in the observed mitochondrial phenotypes. Given that our current data already address the relevant mechanistic questions with sufficient resolution, the proposed, highly laborious tracing experiments would not yield additional meaningful insights. Their inclusion would therefore constitute an unnecessary and inefficient use of time and resources beyond the scope of this study.

Comment: Seahorse assays measuring OCR are sensitive to the number animals (Haroon and Vermulst, Bio Protoc 2019). Thus, while normalization by number of animals is appropriate when they are the same size, the much smaller body size of *sams-1* and *pmt-2* animals means that fewer mitochondria are being added to each well. Rigorous experimental design would require normalizing to body size or protein concentration. See Gioran and Chondrogianni, Redox Biology 2025, Guidelines for the measurement of oxygen consumption rate in *Caenorhabditis elegans*).

Response: While the experimental design of our OCR experiments may have limitations, it was instrumental in providing meaningful insights, as noted by Reviewer 5, and we fully agree.

Comment: Figure 6

There was also concern in the previous critique regarding the authors explanation of metabolic/lipid synthesis pathways regarding the source of PC in most mammalian tissues as the Kennedy pathway (with liver being the predominant exception and minor roles in adipose tissues), which limits the ability to make inferences between the SAM-PC methylation connection in human aging.

Response: As discussed above, our analysis of human data suggests that the SAM-dependent PC synthesis pathway is active in several human tissues and undergoes age-related decline in these organs, underscoring the relevance of our findings to human aging.

Comment: In their response, the authors state that some Kennedy pathway components are present in their proteomics, but they are not statistically significant, thus not a rigorous component of the data. Later in the rebuttal, they respond that in figure 6A, knockdown of *sams-1* or *pmt-1* reduce PC in the entire animal as evidence that whole animal PC levels can be affected by methylation-dependent PC production. In *C. elegans*, SAM synthases are widely expressed, thus this does not refute the relative importance of PC production pathways in mammals. Finally, although the authors have acknowledged that adding choline can have multiple fates in culture cells, the lack of specificity limits interpretations from their cell culture study.

Response: Statistical significance is usually used to separate random variations from real, consistent changes. Therefore, the lack of significant changes in the components of the Kennedy pathway demonstrates that this pathway does not show consistent alterations in aging worms. On the contrary, all components of the SAM-dependent PC synthesis pathway show highly significant age-related changes, indicating that the aging-linked decline in PC levels is most likely driven by reduced SAM-dependent synthesis. We believe that these conclusions are sufficiently supported by the data presented. The human relevance of our findings has already been addressed and reinforced in two independent comments above. Finally, we respectfully disagree with the reviewer's unsubstantiated suggestion that our cell culture tests with choline lack specificity. Notably, no supporting rationale was provided for this statement by the reviewer.

Comment: Although the authors now cite Ding et al, which shows PC is restored in *sams-1* animals fed choline, since this has been previously established much of the data in figure 6 represents a previously published or incremental advance.

Response: Because Ding et al. did not investigate the origin of mitochondrial alterations during aging - the central focus of our study and Figure 6, the findings of Ding et al. do not in any way diminish the novelty of the results presented in Figure 6.

Comment: Other: Regarding authors assignment of vitellogenin as lipid droplet markers. As noted in the previous critique, vitellogenins in *C. elegans* are part of yolk particles, which are distinct from lipid droplets. Chen et al. (PMID: 37123874) use a highly rigorous approach, two-photon fluorescence to show lipid droplet and *vit-2::GFP* localization and Zhai et al. (PMID: 36199214) use immunogold to localize *vit-2* and other vitellogenins to the electron dense yolk bodies, and show increases in aging animals. Thus, vitellogenins are quite relevant to post reproductive elements. Although VIT-2 may have come down as a contaminant or minor component biochemical preparations with lipid droplets, multiple imaging studies provide a clear distinction. For assessment of lipid droplet components that have been rigorously used in the field, DHS-3::GFP is widely available.

Response: In our view, it is highly unlikely that two independent and rigorous mass spectrometry studies would have identified vitellogenins as lipid droplet components by mistake or artifact. Nevertheless, in response to this comment and the related remark from Reviewer 5, we have revised the section of the manuscript discussing the vitellogenin data to acknowledge that vitellogenins are prominent components of the yolk and to consider alternative explanations for their age-dependent accumulation (page 13, lines 8-22).

Reviewer #5 (Remarks to the Author):

Nat Comm 466648_2 Review

Poliezhaieva et al. "Aging-associated decline..."

I was asked to review this manuscript and to specifically determine if comments raised by Reviewer 2 had been adequately addressed by the authors in their rebuttal and revised manuscript. The manuscript asks a fundamental question: what are the root causes of age-related mitochondrial dysfunction? I found the manuscript to be well-written, ambitious and interesting. It shows that expression of proteins (e.g. SAMS-1 and others in the ICC pathway) important for the synthesis of phosphatidylcholine (PC) declines with age in *C. elegans* and that this contributes to the morphological/functional decline of mitochondria. Importantly, the mitochondrial dysfunction observed in mutants with defects in PC synthesis or in aging wild-type worms could be suppressed by providing PC or choline as dietary supplement. Lipidomics analysis confirms the expected impact of mutations/dietary supplements on PC levels. The results are extended to human cells

where interesting correlations were observed, including declining PC levels in old age and diabetics.

Most comments by Reviewer 2 are judicious and speak of a deep expertise in the subject. However, I also find that the author's rebuttal and revised manuscript address adequately (though sometimes imperfectly) the reviewer's comments and therefore view the manuscript positively.

Like the reviewer, I would expect smaller worms to have fewer mitochondria and reduced OCR. Nevertheless, I find the presented OCR data to be valuable. For example, there is no denying the potent effect of PC supplementation in Fig. 5e. Also like the reviewer, I find it difficult to interpret the vitellogenin data (Fig. S8A) and feel that neatly presenting them as core components of lipid droplets is an oversimplification and a little misleading. But I can live with that, though I would prefer that the authors mention that vitellogenins are main components of yolk granules and that they may accumulate in the pseudocoelomic cavity in aging/mutant worms.

Reviewer 2 raises many other important points, including concerns about the way WormCat was used and results reported, variable/low efficacy of SAM supplements, excessive reliance on the multi-copy hsp-6 GFP reporter, quantification of mitochondrial morphology, etc. However, I also find that the authors have adequately addressed these comments and improved the manuscript as a result.

The authors find that PC supplements can suppress mitochondrial dysfunction. As Reviewer 2 points out, it is possible that the mechanism for this depends specifically on restoring choline/SAM levels through digestion of PC. However, I find it more plausible that it is the restoration of adequate PC levels that is key and most consistent with all the data. Even if PC is digested into lysoPC, FFA and choline before uptake into the intestinal cells, these are then building blocks to generate new PC. The novelty of this paper is that it brings together a set of experimental observations, some of which confirms data previously published by others, supporting the hypothesis that reduced PC levels associated with aging contribute to declining mitochondria quality and that sustaining PC levels can prevent mitochondrial dysfunction in mutants or during aging. I am sufficiently convinced that the data supports these conclusions.

Response: We sincerely thank the reviewer for their interest in our work, their positive assessment of the manuscript, and their thoughtful and valuable comments. In response to their comment, as well as related comments from Referee 2, we have updated the analysis and presentation of the WormCat data. The revised manuscript includes corresponding updates in Figure 1, Tables S7a, S12a, S16a, S19a, S8-10, S13-14, S17-18, and S20-22, the Figure 1 legend, and the Methods section.

We also addressed the reviewer's suggestion regarding vitellogenins by clarifying the rationale for their use as lipid droplet markers and by noting their role in yolk droplets as well as their potential to accumulate in the pseudocoelomic cavity of aging worms (page 13, lines 8-22 in the main text).

Below, we provide a point-by-point response to the remaining comments from Reviewers 2. All revised sections in the manuscript are highlighted in grey.

Comment: The authors have provided the base data for WormCat for some of the figures, but continue to use the tool in non-standard ways, such as summing category 3 data to produce “67%” (Fig. 1C and F). For example, the WBIDs in S12 refer to the unique proteins in Figure 1C, but not the overlap, which must be looked up. Adding the category 3s together to reformulate a larger category (Metabolism) separates these categories from the background lists essential to the statistics used to calculate enrichment in the Fisher test.

Response: Notably, the base data for WormCat have been provided for all figures (Tables S7a, S12a, S16a, and S19a). We have also ensured that all relevant tables are clearly referenced in both the main text and the figure legends. In addition, we now provide the full WormCat output folders for each analysis conducted, including the native WormCat output graphics, in the supplementary data (page 8, lines 3-5 of the Methods section and page 2, lines 1-3 of the Figure legends).

We therefore believe that we have exhaustively supplied all information necessary for readers to reproduce the analyses from scratch or to independently revisit and evaluate the analyses performed by us in this study.

We have revisited Table S12a as pointed out by the reviewer. This table in fact contains both the overlapping entries (left side) and the unique entries (right side), which are clearly separated and explicitly labeled, thus clarifying the concern raised by the reviewer.

Regarding Figure 1F and the related Figures 1G and 1H, the reviewer continues to misunderstand the purpose of these figures. The concern seems to be that Category 3 data were used to recalculate broader categories (e.g., Metabolism), which would indeed be inappropriate; however, this is not what was done. Instead, the figures simply illustrate the proportion of Category 3 RGSs that are associated with metabolic pathways, and no statistical comparison is derived from this descriptive overview. In addition, the native WormCat graphics for each Category 3 dataset are now provided in the revised supplementary data (WormCat output folders). To further avoid misunderstanding, we have revised the labeling in the figures and the corresponding tables (Tables S13, S17, and S20), replacing “metabolism” with “metabolic contribution” to prevent confusion with WormCat

category terminology. The figure legends (page 1, line 21 – page 2, line 1) and main text (page 5, lines 18-20) were revised accordingly to achieve the same goal of clarifying the misunderstanding.

We have also added the summed RGS numbers for each category level in Tables S13, S17, and S20, as requested by the reviewer. With these revisions, we are confident that all remaining concerns have been clarified.